# Exploring Data-Driven Models for Compound Flood Forecasting: A comprehensive benchmark

## Abstract

Compound flood forecasting remains challenging due to complex interactions between meteorological, hydrological, and oceanographic factors, a challenge intensified by climate change. Traditional physics-based methods, such as the Hydrologic Engineering Center's River Analysis System, are often time-inefficient and non-executable due to lack of geographic data. While machine learning shows promise over traditional physics-based methods in both accuracy and efficiency, the lack of comprehensive datasets has hindered systematic evaluation of data-driven approaches. To address this gap, we introduce SF$^2$Bench, a comprehensive benchmark for compound flood forecasting using real-world data from South Florida. Our benchmark integrates four critical factors, tide, rainfall, groundwater, and human management activities, enabling systematic comparison of forecasting methods. We evaluate six modeling paradigms: Multilayer Perceptrons, Convolutional Neural Networks, Recurrent Neural Networks, Graph Neural Networks, Transformers, and Large Language Models. Through extensive experiments, we analyze the impact of key features, temporal dependencies, and spatial relationships on forecasting performance. The results across approaches highlight each method's distinct capabilities in capturing compound flood dynamics. By providing this benchmark with code and data, we aim to accelerate progress in flood forecasting through collaborative research between machine learning and environmental science communities. The code and data will be available once the paper is accepted.

## 1 Introduction

Floods are among the most common and hazardous natural events, causing environmental damage (Yin et al., 2023a), catastrophic loss of life (Jonkman & Vrijling, 2008), and property damage (Brody et al., 2007). Compared with single-driver flood events, such as fluvial floods and pluvial floods (Green et al., 2024), compound floods, occurring when two or more distinct flood drivers coincide in space or time (Sebastian, 2022), pose greater challenges for prediction and prevention, making it an important research topic in environmental science. Recent research indicates a rise in both the frequency and scale of compound floods due to global climate change (Wahl et al., 2015; Wing et al., 2022; Hirabayashi et al., 2013). Therefore, understanding the underlying causes of compound floods is both critical and urgent. Accurate and explainable compound flood models can support decision-making in water management, minimizing damage to human life and infrastructure.

Classical physics-based methods predict the water stage by solving complex partial differential equations (PDE) (Paniconi & Putti, 2015; Yin et al., 2023b), such as the Hydrologic Engineering Center's River Analysis System (HEC-RAS) (Brunner, 1997). Despite their accuracy and explainability, the extensive data requirements of physics-based methods, including high-resolution terrain data, reservoir characteristics, canal networks, and river geometries (Sampson et al., 2015; Zang et al., 2021), limit their widespread applications. The rapid development of machine learning (ML) has led to the application of data-centric methods, which utilize deep learning (DL) models for flood prediction and prevention. Researchers employ Convolutional Neural Networks (CNNs) (LeCun et al., 1998), Long Short-Term Memory networks (LSTMs) (Hochreiter & Schmidhuber, 1997), Graph Neural Networks (GNNs) (Kipf & Welling, 2016), and Transformers (Vaswani et al., 2017) to uncover the underlying principles of compound flood.

However, the lack of comprehensive benchmarks has significantly hindered systematic evaluation and comparison of data-driven approaches for compound flood forecasting. Existing methods (Adikari et al., 2021; Ruma et al., 2023; Miau & Hung, 2020; Shi et al., 2024; 2023; Liu et al., 2024c) largely focus on temporal causality, often underestimating the complex interplay of factors. Most prominently, compound floods have garnered increasing attention due to their capacity to analyze multiple influencing factors (Bevacqua et al., 2019; Wahl et al., 2015; Xu et al., 2023; Olbert et al., 2023; Kirschstein & Sun, 2024). Nevertheless, existing datasets (Kabir et al., 2020; Ruma et al., 2023; Shi et al., 2024) often contain limited factors and geographical scope, preventing systematic analysis and fair comparison across different modeling approaches. For example, LamaH-CE (Klingler et al., 2021) provides hydrological and topological data for the Danube River basin but lacks other important factors, such as rainfall and human management activities.

Building a comprehensive benchmark for compound flood analysis is challenging due to diverse drivers (e.g., meteorological factors, tides) and the need for long-term records. A suitable dataset must capture this complexity while supporting robust model evaluation. However, prior studies often focus on limited regions and factors, such as Haikou City in (Xu et al., 2023), limiting their representativeness and generalizability to other coastal areas.

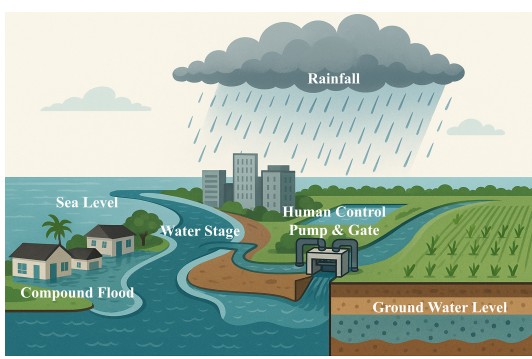

Figure 1: The schematic diagram of compound flood. The key factors include rainfall, sea, groundwater, and human control. The figure is assisted by OpenAI SORA.

In this paper, we introduce SF$^2$Bench, a benchmark for compound flood forecasting built on a curated dataset from South Florida, an ideal testbed for such analysis. South Florida exemplifies vulnerable coastal regions worldwide, facing frequent and severe compound flooding driven by interacting factors such as hurricanes and urban development. Its complex waterway system, low-lying topography, and porous geology, together with flood drivers including sea level, rainfall, river discharge, groundwater, storm surge, and waves (Jane et al., 2020), make it a comprehensive natural laboratory for studying compound flood dynamics. Our benchmark leverages this rich environment by incorporating multiple critical factors: water level, sea level, groundwater level, rainfall, and human management activities on hydraulic structures (gates and pumps), as illustrated in Figure 1. We compiled time series data from 2,452 monitoring stations across multiple counties, spanning from 1985 to 2024, providing unprecedented temporal and spatial coverage for compound flood analysis. This comprehensive dataset serves as a representative example for other coastal areas facing similar compound flooding challenges, offering insights that can inform flood management strategies in comparable regions worldwide.

To establish a rigorous evaluation framework, we benchmark a wide range of state-of-the-art forecasting methods using our dataset, including Multilayer Perceptrons (MLPs), Recurrent Neural Networks (RNNs), CNNs, GNNs, Transformers, and Large Language Models (LLM)-based approaches. Our systematic evaluation reveals that MLPs and Transformers exhibit advantages in terms of MAE and MSE metrics, while MLPs and GNNs demonstrate superior performance in extreme flood events. The benchmark results highlight the varying degrees of effectiveness of each method in capturing the complex temporal and spatial dependencies inherent in compound flooding, providing crucial insights for method selection and development. Furthermore, we conduct comprehensive experiments to demonstrate the individual and combined effectiveness of different factors across various model architectures, offering guidance for feature selection in compound flood modeling. Finally, we discuss potential strategies for improving flood forecasting performance by leveraging both spatial and temporal information, establishing a foundation for future research in this critical area.

## 2 RELATED WORK

**Flood Dataset**. Monitoring floods presents a significant challenge due to their unpredictable nature and potentially devastating consequences. Existing flood datasets can be broadly categorized into satellite image datasets (Rahnemoonfar et al., 2021; Dottori et al., 2022; Montello et al., 2022; Papagiannaki et al., 2022; Xu et al., 2025; Bonafilia et al., 2020) and time series monitoring datasets (Xu

et al., 2023; Kabir et al., 2020; Adikari et al., 2021; Ruma et al., 2023; Klingler et al., 2021; Fowler et al., 2021; Chagas et al., 2020). Satellite image datasets utilize remote sensing to capture surface water extent (Amitrano et al., 2024). While effective in delineating flood-affected areas, this type of dataset (Xu et al., 2025; Bonafilia et al., 2020) often lacks crucial temporal dynamics and information on the underlying hydrological and meteorological factors that drive flood formation, limiting its utility for in-depth modeling. Time series monitoring datasets, on the other hand, typically utilize fixed monitoring stations to record hydrological-related data such as soil moisture, water level, and temperature. A prominent example within this category is the CAMELS-x family of datasets (Addor et al., 2017; Alvarez-Garreton et al., 2018; Coxon et al., 2020; Chagas et al., 2020; Fowler et al., 2021). For instance, CAMELS-BR (Chagas et al., 2020) encompasses data from 3,679 gauges across Brazil. LamaH-CE (Klingler et al., 2021) provides daily and hourly time series data from 882 gauges, including runoff, meteorological variables, and catchment attributes. In (Kirschstein & Sun, 2024), LamaH-CE is used as a benchmark for flood forecasting, primarily focusing on temporal and spatial aspects. However, these datasets primarily focus on general hydrological modeling. Analyzing compound floods, as highlighted in (Jane et al., 2020), necessitates detailed data on rainfall, water levels, and groundwater, which are often limited in existing time series datasets.

**Machine Learning for Forecasting**. The task of forecasting time series data presents inherent complexities and high dimensionality. Recent advancements in time series forecasting have been significantly propelled by a data-centric approach (Zha et al., 2025), underscoring the critical role of extensive, high-quality data in training robust models. Deep learning methodologies, with their powerful representation learning capabilities, have shown considerable promise in this domain. Based on their architectural designs, deep learning methods applied to time series forecasting can be categorized as follows: **MLP-based models** (Chen et al., 2023; Zeng et al., 2023; Wang et al., 2024b; Lin et al., 2024b;a) leverage the capabilities of multilayer perceptrons for analyzing temporal sequences. **RNN-based methods** (Lai et al., 2018; Salinas et al., 2020; Wang et al., 2018; Qin et al., 2017; Jhin et al., 2024) are widely adopted in time series forecasting due to their inherent ability to model temporal dependencies within sequential data. **CNN-based methods** (Wang et al., 2024a; Cheng et al., 2024; Luo & Wang, 2024; Wu et al., 2023; Wang et al., 2023) employ convolutional operations to extract hierarchical features from time series data, enabling effective learning of underlying patterns and trends. **GNN-based methods** (Wu et al., 2020; 2019; Cao et al., 2020; Liu et al., 2022b; Yi et al., 2023; Cai et al., 2023) utilize graph structures to model intricate relationships between different time series variables, enhancing forecasting accuracy. **Transformer-based methods** (Wang et al., 2024c; Liu et al., 2024a; Nie et al., 2023; Zhou et al., 2022; Liu et al., 2022a; Wu et al., 2021; Zhou et al., 2021a) have demonstrated remarkable performance in capturing long-range dependencies and complex temporal dynamics within time series data. **LLM-based methods** (Jin et al., 2024; Pan et al., 2024; Zhou et al., 2023; Gruver et al., 2023) explore the application of prompting and reprogramming techniques to align time series data with text embeddings for forecasting tasks.

## 3 THE SF$^2$BENCH DATASET

SF$^2$Bench comprises a time series data collection from 2,452 monitoring stations across a 67,349 km$^2$ area in South Florida, sourced from the South Florida Water Management District (SFWMD) [1]. The dataset spans the period from 1985 to 2024 and is divided into 8 temporal splits. This dataset incorporates key factors that play critical roles in compound floods (Jane et al., 2020), including water level [2], rainfall, groundwater level, and human control data for pumps and gates. Notably, sea level data is inherently included within the water level at certain monitoring stations due to their direct connection to the sea. It is specifically collected for benchmarking data-driven forecasting approaches in the context of compound flood analysis. In Table 1, we provide a comparison with other datasets. Compared to CAMELS-x (Addor et al., 2017; Alvarez-Garreton et al., 2018; Coxon et al., 2020; Chagas et al., 2020; Fowler et al., 2021), SF$^2$Bench focuses on a region particularly susceptible to flooding, making it more relevant for compound flood analysis. In comparison to BangladeshFlood (Ruma et al., 2023), SF$^2$Bench offers a more comprehensive set of driving factors relevant to compound flooding, considering both temporal and spatial dimensions.

---

[1]https://www.sfwmd.gov/

[2]Water stage and water level are used interchangeably.

Table 1: **Comparison with other datasets for flood forecasting.** N/A indicates Not Available. * marks datasets with Other Attributions. Climatic indices capture climate statistics (e.g., potential evapotranspiration). Land cover attributes describe surface materials (e.g., woodland ratio). Soil attributes include properties such as porosity and depth, while geological attributes refer to subsurface features (e.g., geologic class, porosity). Anthropogenic influences cover human activities (e.g., water abstraction, discharges). Other catchment attributes include location, area, and topography.

| Dataset | Time Span | Interval | Type | Gauges | Area(km$^2$) | Public | Other Attributions |
|---|---|---|---|---|---|---|---|
| DarlingFlood | 1900-2018 | Daily | Flow | 12 | $3.5 \times 10^4$ | No | Rainfall |
| SekongFlood | 1981-2013 | Daily | Flow | 8 | $2.8 \times 10^4$ | No | Rainfall |
| BangladeshFlood | 1979-2013 | Daily | Stage | 24 | $1.5 \times 10^5$ | No | N/A |
| Qi River | 1979-2020 | Hour | Flow | 7 | $7.1 \times 10^3$ | No | Rainfall |
| Tunxi basins | 1981-2007 | Hour | Flow | 12 | N/A | No | Rainfall |
| CAMELS* | 1989-2009 | Daily | Flow | 671 | $1.0 \times 10^4$ | Yes | Climatic Indices |
| CAMELS-CL* | 1913-2018 | Daily | Flow | 516 | N/A | Yes | Land Cover Attributes |
| CAMELS-GB* | 1970-2015 | Daily | Flow | 671 | $2.1 \times 10^5$ | Yes | Soil Attributes |
| CAMELS-BR* | 1925-2024 | Daily | Flow | 4,025 | N/A | Yes | Geological Attributes |
| CAMELS-AUS* | 1951-2014 | Daily | Flow | 107 | $6.9 \times 10^5$ | Yes | Anthropogenic Influences |
| LamaH-CE* | 1951-2014 | Daily & Hour | Flow | 859 | $1.7 \times 10^5$ | Yes | Other Catchment Attributes |
| SF$^2$Bench | 1985-2024 | Hour | Stage | 2,452 | $6.7 \times 10^4$ | Yes | Rainfall, Groundwater, Human Control |

## 3.1 PREPROCESSING

**Data Collection**. The data for SF$^2$Bench was collected from DBHYDRO [3], an environmental database maintained by the SFWMD that stores a wide range of hydrologic, meteorologic, hydrogeologic, and water quality data. We initially collected data from 3731 monitoring stations and subsequently screened them based on data availability and relevance to flood analysis, resulting in a final set of 2,452 valid stations. This included 993 water stage monitoring stations, 349 rainwater monitoring stations, 582 groundwater level monitoring stations, 99 pump stations, and 429 gates.

The different types of data and their physical meanings are summarized in Table 2. To capture fine-grained temporal dynamics, all raw data is collected at their native "breakpoint" frequency, resulting in a high (up to second-level) temporal resolution for each recorded value. However, this breakpoint frequency collection method results in inconsistent data across stations, with some recording data at much higher frequencies than others. In addition, some monitoring stations have missing data for certain periods, while others provide data only for a limited duration, such as one year.

**Data Processing**. To provide an AI-ready dataset, we introduce data processing to unify the format. As previously mentioned, the data collection ranges vary across different monitoring stations, making it challenging to standardize all data to a uniform length. To balance the number of time series and temporal length, we divided the data into eight splits. Each time-series data is sampled with an hourly interval for all splits. During processing, for each interval, we first compute the mean of the available data points within that interval. Subsequently, we address missing values using interpolation methods. According to the characteristics of the data, we have two in-

Table 2: The summary of the data type in SF$^2$Bench.

| Type | Stations | Unit | Description |
|---|---|---|---|
| Water | 993 | Feet | Water Stage |
| Groundwater | 582 | Feet | Stage of Groundwater |
| Rainfall | 349 | Inches | Rainfall |
| Pump | 99 | RPM | Rotational Speed |
| Gate | 429 | Feet | Opening Level |

terpolation methods: linear interpolation and zero data filling. For water stage and groundwater level, we regard them as continuous variables and apply linear interpolation to fill missing values. For the other variables, rainfall and control data (pumps and gates), we treat them as discrete events and fill missing values with zero. The detailed information is presented in Appendix Table 7.

## 3.2 QUALITATIVE ANALYSIS

To provide an intuitive insight of SF$^2$Bench, we provide visualization results from spatial and temporal aspects. More detailed information are provided in the Appendix A.

---

[3] https://apps.sfwmd.gov/dbhydroInsights/%23/homepage

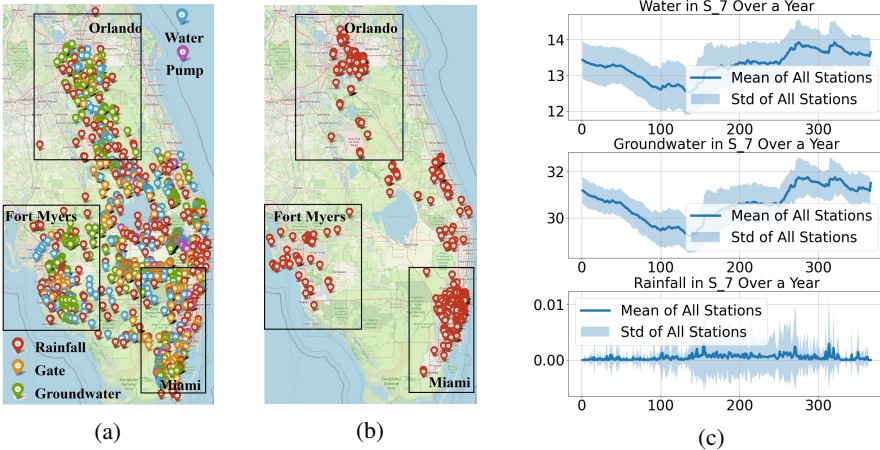

Figure 2: (a) Monitor stations location distribution. (b) Flood observation location distribution. (c) Temporal patterns of key features over a year. In (a) and (b), we highlight three interest parts: Orlando, Fort Myers, and Miami. In (c), the x-axis is the number of days in one year.

**Spatial Distribution**. Figure 2a illustrates the spatial distribution of all monitoring stations, which are primarily located around the intricate river system of South Florida, radiating outwards from Lake Okeechobee. The geographically staggered distribution of hydrological, groundwater, and rainwater monitoring stations enables more effective spatial analysis. For example, the water level at a specific location is likely correlated with rainfall in the surrounding area and the local groundwater level (representing the soil's water storage capacity). We also highlight the observed flood locations from 2020 to 2023 and in 2008 in Figure 2b provided by SFWMD (Office of Resilience, South Florida Water Management District, 2025), which are predominantly concentrated in urban areas.

**Temporal Pattern Visualization**. Figure 2c illustrates the average annual temporal patterns across all monitoring stations, using data from split S7 as a representative example. From this pattern, we observe a strong correlation between groundwater level and water level data. The water level generally rises from approximately the 150th to the 300th day of the year, followed by a gradual decrease. According to the National Weather Service (NWS) [4], Florida's climate is characterized by distinct dry and rainy seasons, with the latter typically spanning from May to October. This aligns well with the observed data patterns in our dataset. Taking rainfall as a reference, we note that increases in water level tend to correspond with rainfall events, such as the rainfall peak after the 300th day and the subsequent rise in water level. In addition, human control activities on hydraulic structures (e.g., pumps and gates) appear to influence water level changes in response to rainfall. We can infer that human intervention has played a role in mitigating potential flooding.

## 4 FORECASTING BENCHMARKS

### 4.1 PROBLEM DEFINITION

Given water stage time series data $\boldsymbol{X} \in \mathbb{R}^{N \times L}$ from $N$ water monitoring stations, the forecasting task is to predict the water stage values for the next $T$ time steps, denoted as $\boldsymbol{Y} \in \mathbb{R}^{N \times T}$, using a fixed look-back window of length $L$. We also have access to additional time series information, including groundwater levels $\boldsymbol{X}_w \in \mathbb{R}^{N_w \times L}$ (from $N_w$ stations), rainfall $\boldsymbol{X}_r \in \mathbb{R}^{N_r \times L}$ (from $N_r$ stations), pump control data $\boldsymbol{X}_p \in \mathbb{R}^{N_p \times L}$ (from $N_p$ stations), gate control data $\boldsymbol{X}_g \in \mathbb{R}^{N_g \times L}$ (from $N_g$ stations), and location information for the monitoring stations. For the primary benchmark experiments aimed at fair comparison across different models, we mainly consider the water stage data as the supervised data. However, we acknowledge that incorporating the additional information (groundwater levels, rainfall, pump and gate control data, and location) has the potential to further enhance forecasting performance.

---

[4]https://www.weather.gov/tbw/TBWTstmClimoQuickReference

Table 3: Benchmark of average MAE&MSE results on three interest areas across 8 splits. The best and second results are shown in bold font and underlined, respectively.

| Metric | T | MLP | | | CNN | | | Transformer | |
|---|---|---|---|---|---|---|---|---|---|
| | | MLP | TSMixer | NLinear | TCN | ModernTCN | TimesNet | iTransformer | PatchTST |
| ↓MAE | 1D | 0.0788 | 0.0928 | 0.0817 | 0.1792 | 0.0798 | 0.0983 | 0.0756 | **0.0741** |
| | 3D | 0.1351 | 0.1442 | 0.1373 | 0.2281 | 0.1441 | 0.1528 | **0.1314** | 0.1316 |
| | 5D | 0.1764 | 0.1816 | 0.1769 | 0.2697 | 0.1891 | 0.1927 | 0.1722 | **0.1719** |
| | 7D | 0.2063 | 0.2126 | 0.2082 | 0.3088 | 0.2248 | 0.2267 | **0.2041** | 0.2044 |
| | Avg. | 0.1492 | 0.1578 | 0.1510 | 0.2465 | 0.1594 | 0.1676 | 0.1458 | **0.1455** |
| ↓MSE | 1D | 0.0521 | 0.0648 | 0.0556 | 0.3722 | 0.0681 | 0.0704 | **0.0253** | 0.0531 |
| | 3D | **0.1029** | 0.1132 | 0.1105 | 0.4647 | 0.2443 | 0.1358 | 0.1112 | 0.1132 |
| | 5D | **0.1432** | 0.1566 | 0.1547 | 0.4173 | 0.2808 | 0.1829 | 0.1583 | 0.1566 |
| | 7D | **0.1707** | 0.1903 | 0.1841 | 0.5386 | 0.2723 | 0.2241 | 0.1911 | 0.1903 |
| | Avg. | **0.1172** | 0.1283 | 0.1262 | 0.4482 | 0.2164 | 0.1533 | 0.1284 | 0.1283 |

| Metric | T | RNN | | | GNN | | | LLM | |
|---|---|---|---|---|---|---|---|---|---|
| | | LSTM | DeepAR | DilatedRNN | GCN | FourierGNN | StemGNN | GPT4TS | AutoTimes |
| ↓MAE | 1D | 0.1182 | 0.1178 | 0.0919 | 0.1696 | 0.0921 | 0.1332 | 0.1256 | 0.0846 |
| | 3D | 0.1821 | 0.1837 | 0.1573 | 0.2006 | 0.1503 | 0.2181 | 0.1521 | 0.1362 |
| | 5D | 0.2232 | 0.2247 | 0.2022 | 0.2504 | 0.1930 | 0.3153 | 0.1911 | 0.1752 |
| | 7D | 0.2576 | 0.2596 | 0.2374 | 0.2799 | 0.2280 | 0.3570 | 0.2247 | 0.2062 |
| | Avg. | 0.1953 | 0.1964 | 0.1722 | 0.2251 | 0.1658 | 0.2559 | 0.1734 | 0.1505 |
| ↓MSE | 1D | 0.1339 | 0.1230 | 0.1033 | 7.2299 | 0.0768 | 0.1632 | 0.0966 | 0.0584 |
| | 3D | 0.1985 | 0.1954 | 0.1672 | 1.5645 | 0.1416 | 0.2543 | 0.1410 | 0.1125 |
| | 5D | 0.2348 | 0.2389 | 0.2341 | 2.1341 | 0.2071 | 0.4189 | 0.1847 | 0.1522 |
| | 7D | 0.2873 | 0.2691 | 0.2591 | 1.0374 | 0.2125 | 0.4716 | 0.2245 | 0.1823 |
| | Avg. | 0.2136 | 0.2066 | 0.1909 | 2.9915 | 0.1595 | 0.3270 | 0.1617 | 0.1263 |

Table 4: Benchmark of average SEDI results on three interest areas across 8 splits. The best and second results are shown in bold font and underlined, respectively.

| Metric | MLP | | | CNN | | | Transformer | |
|---|---|---|---|---|---|---|---|---|
| | MLP | TSMixer | NLinear | TCN | ModernTCN | TimesNet | iTransformer | PatchTST |
| ↑SEDI(10%) | **0.6897** | 0.6144 | 0.6278 | 0.5311 | 0.6067 | 0.5829 | 0.6286 | 0.6296 |
| ↑SEDI(5%) | **0.5834** | 0.4942 | 0.5111 | 0.3706 | 0.4846 | 0.4589 | 0.5079 | 0.5086 |
| ↑SEDI(1%) | **0.3666** | 0.2480 | 0.2767 | 0.1387 | 0.2512 | 0.2222 | 0.2690 | 0.2685 |

| Metric | RNN | | | GNN | | | LLM | |
|---|---|---|---|---|---|---|---|---|
| | LSTM | DeepAR | DilatedRNN | GCN | FourierGNN | StemGNN | GPT4TS | AutoTimes |
| ↑SEDI(10%) | 0.6097 | 0.5989 | 0.6164 | 0.6179 | 0.6623 | 0.5507 | 0.5581 | 0.6239 |
| ↑SEDI(5%) | 0.4702 | 0.4643 | 0.5014 | 0.5138 | 0.5511 | 0.4270 | 0.4640 | 0.5015 |
| ↑SEDI(1%) | 0.1680 | 0.1478 | 0.2499 | 0.2828 | 0.3217 | 0.1690 | 0.2268 | 0.2568 |

## 4.2 METRICS

We follow standard time series forecasting practices by using the Mean Absolute Error (MAE) and Mean Squared Error (MSE) as our primary evaluation metrics. To better assess the performance of our models in real-world applications, particularly for extreme flood events, we also employ the Symmetric Extremal Dependence Index (SEDI) (Han et al., 2024a; Xu et al., 2024), as suggested by (Han et al., 2024b). By selecting quantile thresholds (e.g., the 95th and 5th percentiles of the observed values), SEDI classifies each time stamp as belonging to either a normal or an extreme case and then calculates the hit rate of this classification. A higher SEDI value indicates better performance in predicting extreme events. The formulation of SEDI is as follows:

$$\text{SEDI}(p) = \frac{|\hat{\boldsymbol{Y}} < V_{1-\frac{p}{2}} \& \boldsymbol{Y} < V_{1-\frac{p}{2}}| + |\hat{\boldsymbol{Y}} > V_{\frac{p}{2}} \& \boldsymbol{Y} > V_{\frac{p}{2}}|}{|\boldsymbol{Y} < V_{1-\frac{p}{2}}| + |\boldsymbol{Y} > V_{\frac{p}{2}}|}, \tag{1}$$

where $|\cdot|$ means the number of the true values satisfying the condition, $\hat{\boldsymbol{Y}}$ is the forecasting results, $p$ is the quantile of the threshold, and $V_{1-\frac{p}{2}}, V_{\frac{p}{2}}$ are the lower and upper threshold of top and worst $\frac{p}{2}$ percent, respectively.

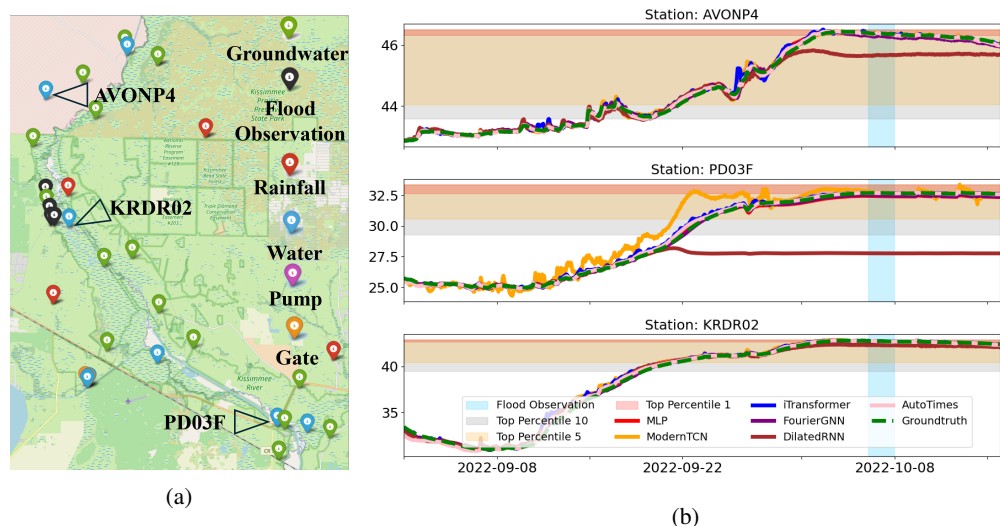

(a)

(b)

Figure 3: (a) The interest area of the case study. (b) The comparison among different methods. We mark the flood observation on the x-axis and the top percentile 10/5/1 with colors.

## 4.3 METHODS

We benchmark six categories of time series forecasting architectures: MLP, CNN, RNN, GNN, Transformer, and LLM. We select two advanced methods as representative examples for each of these categories. Additionally, for the first four classical architectures, we also implement a basic foundational architecture. The specific advanced methods we evaluate are: MLP: NLinear (Zeng et al., 2023), TSMixer (Chen et al., 2023), CNN: ModernTCN (Luo & Wang, 2024), TimesNet (Wu et al., 2023), RNN: DeepAR (Salinas et al., 2020), DilatedRNN (Chang et al., 2017), GNN: FourierGNN (Yi et al., 2023), StemGNN (Cao et al., 2020), Transformer: PatchTST (Nie et al., 2023), iTransformer (Liu et al., 2024a), LLM: GPT4TS (Zhou et al., 2023), AutoTimes (Liu et al., 2024b). We follow the source code of NeuralForecast [5] for the implementation of these approaches. The summary of these methods can be found in Appendix B.2.

## 4.4 EXPERIMENT SETUP

Due to the memory and training time limitations associated with GPUs, applying some methods, particularly LLM-based approaches (Pan et al., 2024; Jin et al., 2024; Zhou et al., 2023; Liu et al., 2024b), to the entire dataset is challenging. To facilitate a fair comparison, we conduct our experiments using two setups: evaluation on three specific areas of interest and evaluation on the entire dataset. The three areas were selected based on the flood observation data presented in Figure 2. Figure 2b visualizes these flood

Table 5: Average results of basic methods on the whole dataset. The best and second results are shown in bold font and underlined, respectively.

| Metric | MLP | LSTM | TCN | GCN |
|---|---|---|---|---|
| ↓MAE | **0.2328** | 0.3302 | 0.3491 | 0.3053 |
| ↓MSE | **0.3902** | 2.2772 | 0.6807 | 1.0673 |
| ↑SEDI(10%) | **0.6853** | 0.5751 | 0.5167 | 0.6137 |
| ↑SEDI(5%) | **0.5970** | 0.4516 | 0.3904 | 0.5213 |
| ↑SEDI(1%) | **0.4107** | 0.1828 | 0.1842 | 0.3134 |

locations alongside the selected areas. We report the performance of all evaluated methods within these areas of interest. For the entire dataset, we only report the results of the basic foundational techniques. For each data split, the last year's data is used for testing, the data of the second-to-last year is used for validation, and the remaining preceding data is used for training. In the benchmark, we maintain a consistent lookback window of two days, and we evaluate prediction windows of one, three, five, and seven days. The detailed experimental setup, including software and hardware platforms, is provided in the Appendix B.

---

[5]https://github.com/Nixtla/neuralforecast

Table 6: Input factors ablation study on S6. All, G, R, and C represent all factors, groundwater, rainfall, and human control(pump and gate). The best and second results are shown in bold font and underlined, respectively.

| Metric | method | w/ All | w/o G | w/o R | w/o C | w/o GR | w/o RC | w/o WC | w/o WRC |
|--------|--------|--------|-------|-------|--------|--------|--------|--------|---------|
| ↓MAE | iTransformer | 0.1406 | 0.1407 | 0.1406 | 0.1405 | 0.1411 | 0.1410 | **0.1402** | 0.1411 |
| | PatchTST | **0.1376** | 0.1391 | 0.1378 | 0.1377 | 0.1398 | 0.1396 | 0.1385 | 0.1421 |
| | TSMixer | 0.1596 | 0.1419 | 0.2523 | 0.2111 | **0.1418** | 0.2807 | 0.1423 | 0.1422 |
| | NLinear | 0.1546 | 0.1577 | 0.1483 | 0.1540 | 0.1485 | 0.1450 | 0.1579 | **0.1435** |
| | TimesNet | 0.1642 | 0.1599 | 0.1662 | 0.1650 | 0.1592 | 0.1656 | **0.1578** | 0.1580 |
| ↓MSE | iTransformer | 0.0953 | 0.0960 | **0.0949** | 0.0954 | 0.0957 | 0.0956 | **0.0949** | 0.0962 |
| | PatchTST | 0.0917 | 0.0932 | 0.0927 | **0.0916** | 0.0950 | 0.0938 | 0.0923 | 0.0971 |
| | TSMixer | 0.1080 | 0.0946 | 0.4061 | 0.2698 | 0.0946 | 0.6697 | 0.0943 | **0.0941** |
| | NLinear | 0.0992 | 0.1011 | 0.0966 | 0.0984 | 0.0973 | 0.0954 | 0.1006 | **0.0947** |
| | TimesNet | 0.1188 | 0.1154 | 0.1226 | 0.1202 | 0.1145 | 0.1237 | **0.1111** | 0.1123 |

## 4.5 RESULTS & OBSERVATIONS

**Overall Performance**. Table 3 reports the average MSE and MAE results across three regions and eight data splits. The top-performing methods include PatchTST, iTransformer, MLP, NLinear, and AutoTimes. PatchTST and iTransformer achieve the best MAE, while MLP, NLinear, and AutoTimes perform best under MSE. A small MAE but large MSE suggests high accuracy for most points with occasional large errors, whereas the opposite indicates greater stability but lower pointwise accuracy. Table 4 shows model performance on extreme cases, where FourierGNN ranks second in SEDI despite weaker MSE and MAE. Detailed split results are provided in Appendix C. Table 5 further presents the average performance of basic methods on the full dataset (with details in Appendix C.1). Among these, MLP consistently performs best, followed by GCN and TCN, consistent with the MSE and MAE trends in Table 3. This finding aligns with prior work (Kirschstein & Sun, 2024), which also shows MLP outperforming GNNs. Overall, our results show that MLP, transformer-based models (PatchTST, iTransformer), and LLM-based models (AutoTimes, GPT4TS) achieve strong performance on MAE and MSE, while the GNN-based FourierGNN excels on SEDI. Since extreme-case predictions are crucial for assessing flood risk, SEDI provides complementary insights. Notably, MAE/MSE and SEDI results are not strongly correlated, highlighting the need for multiple evaluation metrics. We also find no clear link between performance and model size. For example, NLinear $(9K)$ performs comparably to AutoTimes $(4M)$ despite its far smaller parameter count.

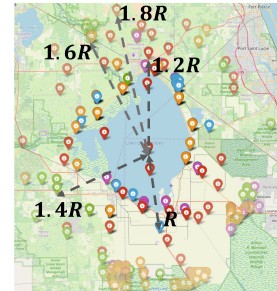

Figure 4: The different interest areas with a radius for ablation study on spatial information.

**Case Study**. To evaluate the performance of different models in real flood scenarios, we conduct a case study using Hurricane Ian, which struck Florida on September 28, 2022, causing widespread flooding in multiple locations, including Orlando (Hur, 2023). We select the middle stage of the Kissimmee River as our study area because the water monitoring stations are located near the observed flood zones, as shown in Figure 3a. This location provides an ideal setting for model evaluation, with available data for three key factors: water level, groundwater level, and rainfall. In the experiments, all models were trained to forecast water levels for the next 24 hours. Figure 3b presents the forecasting results from three representative water monitoring stations, revealing clear performance differences among the six methods. The results show that ModernTCN and DilatedRNN achieve lower accuracy compared to other approaches. While most methods successfully captured the overall flooding trend, iTransformer showed notably larger prediction errors in certain areas. These visualization results are consistent with the quantitative findings presented in Table 3.

**Ablation of Factors**. To verify and demonstrate the influence of different input factors, we conduct an ablation study using five methods. These methods contains two settings: channel-independent (including NLinear and PatchTST) and channel-dependent (including iTransformer, TimesNet, and TSMixer). For the channel-independent methods, we used all available factors as input. In contrast, for the channel-dependent methods, we primarily used the water stage data as the supervised target, while other factors were considered as potential additional inputs. As shown in Table 6, NLinear and

TSMixer achieve comparable results when the input is limited to only the water stage data, suggesting that the inclusion of other factors does not significantly improve their performance. For iTransformer, PatchTST, and TimesNet, we observe performance improvements when additional information is provided, highlighting the potential benefit of incorporating multi-faceted data for this forecasting task. Moreover, for iTransformer and PatchTST, we find excluding groundwater information (denoted as "w/o G" and "w/o GR") results in larger MSE and MAE errors compared to settings where other factors are excluded (e.g., "w/o R" for without rainfall, "w/o C" for without control data, "w/o RC", and "w/o WC"). Moreover, for iTransformer and TimesNet, providing only groundwater information as the additional input leads to the best performance among the ablation settings, suggesting that the groundwater is a particularly informative factor for these models. We also observe similar results on the SEDI(10%) metric, provided in Appendix C.2.

**Impact of Temporal and Spatial Information**. To provide insight into the impact of temporal input length and spatial information, we conduct ablation studies. We first select an interest area as the anchor area, where the water stages are regarded as the forecasting target. As shown in Figure 4, $R$ is the radius of the anchor area. Then, we incorporate information from surrounding stations by incrementally increasing the radius of the interest area. In these experiments, we consider radius scale factors of 1,1.2,1.4,1.6, and 1.8. The MAE, MSE, and SEDI results, presented in Figures 5a, show that iTransformer, PatchTST, and TSMixer experience a performance improvement as the input area expands. This indicates the effectiveness of incorporating additional spatial information for the forecasting task. We provide the detailed results in Appendix C.2.

Furthermore, maintaining the anchor area as the forecasting target, we evaluate the impact of temporal input length by considering a range of durations: 6 hours, 12 hours, and 1 to 6 days. As shown in Figure 5b, we observe that PatchTST, TSMixer, and iTransformer generally show improved performance with increasing input length. However, for iTransformer, performance begins to decrease beyond an input length of 1 day. A potential reason for this phenomenon is that longer input sequences require a larger amount of training data to effectively learn the underlying patterns. The detailed results are available in Appendix C.2. Comparing these two strategies, we find that both can enhance task performance. Increasing spatial input generally leads to a relatively stable, albeit limited, improvement. In contrast, extending the temporal input length can yield more substantial gains, particularly for models like TSMixer, where a significant reduction in MSE is observed.

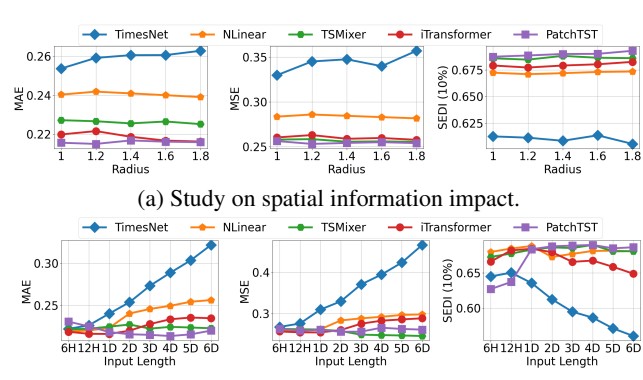

(a) Study on spatial information impact.

(b) Study on temporal information impact.

Figure 5: Combined ablation studies results.

## 5 CONCLUSION

In this paper, we introduce a new benchmark SF$^2$Bench, the dataset collected for comprehensive compound flood analysis. Our goal is to foster collaboration between the machine learning and environmental science communities by providing a resource that bridges domain expertise with advanced predictive modeling. SF$^2$Bench comprehensively covers the majority of South Florida and integrates five key factors: water level, sea level, groundwater table, rainfall, and human control activities. To assess its utility, we evaluate six types of time series forecasting approaches on this dataset and observe that different architectures exhibit distinct advantages. Furthermore, our ablation studies on input factors reveal that groundwater level is a particularly effective predictor compared to other information sources. Additionally, we conduct experiments to explore the effectiveness of increasing spatial and temporal information, and the results demonstrate that both strategies improve forecasting performance for this task.

## ETHIC STATEMENT

All authors confirm that they have read and commit to upholding the ICLR Code of Ethics. All experiments use publicly available datasets; no human subjects or sensitive data are involved.

## REPRODUCIBILITY STATEMENT

All code and datasets are included in the Supplementary Material. All experiments use publicly available datasets, and are reproducible.

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

CONTENTS

# A    DETAILED INFORMATION OF SF$^2$BENCH

We provide the detailed information about the number of monitor stations in each split in Table 7. In each split, the number of water monitor stations is more than other kinds of features, which means the water level feature is the main information and others are additional parts.

Table 7: The summary of the all splits in SF$^2$Bench

| Splits | Time Span | Interval | Water | Groundwater | Rainfall | Pump | Gate |
|---|---|---|---|---|---|---|---|
| S0 | 1985-1990 | 1 Hour | 159 | 40 | 143 | 17 | 82 |
| S1 | 1990-1995 | 1 Hour | 227 | 36 | 139 | 18 | 104 |
| S2 | 1995-2000 | 1 Hour | 332 | 44 | 170 | 26 | 94 |
| S3 | 2000-2005 | 1 Hour | 402 | 178 | 227 | 31 | 107 |
| S4 | 2005-2010 | 1 Hour | 518 | 296 | 254 | 48 | 172 |
| S5 | 2010-2015 | 1 Hour | 585 | 333 | 216 | 65 | 256 |
| S6 | 2015-2020 | 1 Hour | 670 | 317 | 186 | 85 | 300 |
| S7 | 2020-2024 | 1 Hour | 716 | 352 | 194 | 89 | 329 |

We also provide the geographical distribution information of monitor stations in different splits in Figure 6. As time goes by, the number of monitoring stations gradually increases, and in terms of spatial distribution, the locations of monitoring stations remain consistent.

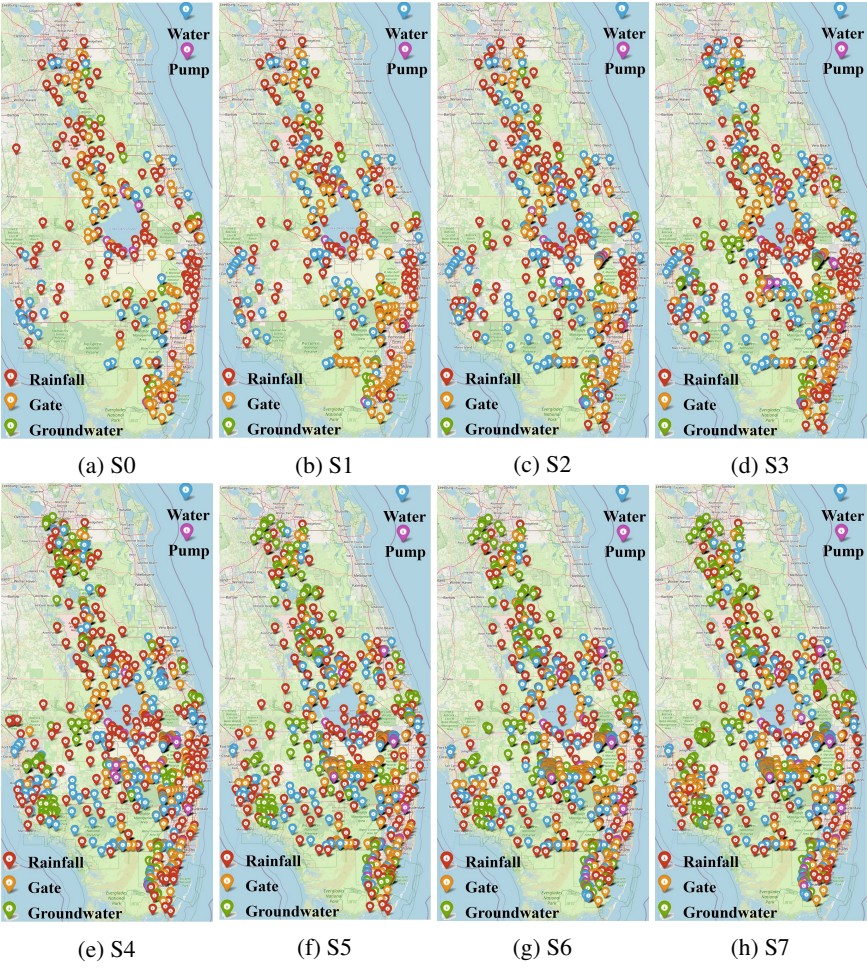

(a) S0            (b) S1            (c) S2            (d) S3

(e) S4            (f) S5            (g) S6            (h) S7

Figure 6: The location distribution in each split.

In addition, we visualize the temporal pattern of features over a year in each split. As shown in Figure 7, 8, 9, and 10, the temporal patterns in 8 splits are similar and consistent. From the view of the climate, the dry and rainy seasons are highly consistentm but the intensity of rainfall in different splits is different. These patterns that are highly aligned with the actual situation demonstrate the quality of the dataset.

## B    EXPERIMENTAL DETAILS

### B.1    DATA PREPROCESSING

**Normalization**. To make the data have a zero mean and unit variance, we follow (Franceschi et al., 2019; Zhou et al., 2021b; Yue et al., 2022) using z-score to normalize the time series data. For the time series, whose variance is less than 1E-4, we regard its variance as one to avoid the variable overflow(inf or NaN). For forecasting tasks, all the report metrics are based on the normalized data.

**Timestamp Features**. Because of some methods, such as NLinear, we do not consider extracting the timestamp feature as part of the input. For those methods that are capable of handling the timestamp feature, we ignore this part. In our code, we also provide timestamp features extracted by following (Zhou et al., 2021b; Yue et al., 2022) for further works.

### B.2    METHODS

**MLP**. We implement a classical three-layer MLP with ReLU as the activation function. The input layer dimension is determined by the input sequence length, and the output layer dimension corresponds to the forecast horizon. The hidden layer dimension is 64. For training, the learning rate is $1 \times 10^{-4}$, weight decay is $1 \times 10^{-6}$, and the batch size is 64. The model is trained for 15 epochs.

**NLinear** (Zeng et al., 2023). This is a simple linear model that treats each time series independently, modeling future values using a linear transformation of the most recent input values. Implementation is based on the NeuralForecast library[6]. The learning rate is $1 \times 10^{-4}$, weight decay is $1 \times 10^{-6}$, batch size is 64, and training is performed for 50 epochs.

**TSMixer** (Chen et al., 2023). Inspired by MLP-Mixer models from vision tasks, TSMixer is a neural network architecture for time series forecasting. It alternately applies MLPs along the time and feature axes, learning dependencies across both dimensions without requiring attention mechanisms or complex sequence modeling. Implementation follows the NeuralForecast default settings. The architecture includes 2 mixing layers, and the second feed-forward layer has 64 units. The learning rate is $1 \times 10^{-4}$, batch size is 32, and the model is trained for 10 epochs.

**TCN** (Bai et al., 2018). It incorporates causal convolutions, ensuring predictions depend only on current and past inputs, thus preserving temporal order. They also utilize dilated convolutions to efficiently capture long-range dependencies by expanding the receptive field without significantly increasing layers. Our implementation follows the official code[7] using the popular channel-wise setting. It employs a three-layer backbone with a kernel size of 3 and a fixed dilation of 1. The learning rate is $1 \times 10^{-3}$, weight decay is $1 \times 10^{-7}$, batch size is 256, and training is performed for 50 epochs.

**ModernTCN** (Luo & Wang, 2024). ModernTCN introduces a streamlined, fully convolutional architecture that aims to simplify design while enhancing performance. It incorporates components like depth-wise separable convolutions and Gated Linear Units (GLUs) to efficiently capture local and long-range temporal dependencies. Implementation follows the long-term forecasting settings from the source code[8], using the Weather dataset hyperparameters as defaults. The learning rate is $1 \times 10^{-4}$, batch size is 256, and the model is trained for 100 epochs.

**TimesNet** (Wu et al., 2023). TimesNet models temporal variations in a two-dimensional space by reshaping time series data into a pseudo-image format and applying 2D convolutional techniques. This enables it to capture both short-term dynamics and long-term dependencies. Implementation uses the Time-Series-Library[9], with default hyperparameters from the long-term forecasting setting for the Weather dataset. The learning rate is $1 \times 10^{-4}$, batch size is 32, and training is performed for

---

[6] https://github.com/Nixtla/neuralforecast
[7] https://github.com/locuslab/TCN/tree/master
[8] https://github.com/luodhhh/ModernTCN
[9] https://github.com/thuml/Time-Series-Library

10 epochs.

**LSTM** (Hochreiter & Schmidhuber, 1997). Our Long Short-Term Memory (LSTM) implementation is a two-layer model with a hidden dimension of 32. The learning rate is $1 \times 10^{-3}$, weight decay is $1 \times 10^{-6}$, batch size is 64, and the model is trained for 50 epochs.

**DeepAR** (Salinas et al., 2020). DeepAR is a global model trained on multiple related time series, which aids generalization, especially for series with limited history. It employs an RNN architecture to predict future values by modeling the conditional distribution of the next value given past observations. Implementation is based on the NeuralForecast library. The learning rate is $1 \times 10^{-3}$, batch size is 64, and training is performed for 20 epochs.

**DilatedRNN** (Chang et al., 2017). Dilated RNNs exponentially expand their receptive field by stacking layers with different dilation factors, allowing efficient capture of short- and long-range patterns without a drastic increase in parameters. This makes them suitable for forecasting tasks with wide-ranging temporal dependencies. Implementation is based on the NeuralForecast library. The learning rate is $1 \times 10^{-3}$, batch size is 64, and training is performed for 40 epochs.

**GCN** (Kipf & Welling, 2017). The GCN architecture consists of two layers with a hidden dimension of 32. The graph topology is derived from location information using Delaunay triangulation[10]. The learning rate is $1 \times 10^{-4}$, weight decay is $1 \times 10^{-5}$, batch size is 32, and the model is trained for 50 epochs.

**FourierGNN** (Yi et al., 2023). FourierGNN leverages graph neural networks and Fourier transforms to capture temporal and inter-variable dependencies. Time series variables are treated as graph nodes, with edges representing their relationships. Fourier transforms project data into the frequency domain to model periodic and long-range dependencies. Implementation follows the source code[11]. The learning rate is $1 \times 10^{-5}$, batch size is 32, and training is performed for 100 epochs.

**StemGNN** (Cao et al., 2020). StemGNN is designed to capture both temporal (via temporal convolutions) and spatial (via spectral graph convolutions) dependencies in time-series data, learning smooth representations over the graph structure and dynamic patterns. Implementation follows the source code[12]. The learning rate is $1 \times 10^{-4}$, batch size is 32, and the model is trained for 50 epochs.

**iTransformer** (Liu et al., 2024a). The iTransformer uses an encoder-decoder structure where the encoder processes the sequence in reverse order. This allows the decoder to predict future values based on this processed representation, enhancing focus on relevant temporal sequences while mitigating the computational cost of traditional transformers. Implementation is based on the NeuralForecast library. The learning rate is $1 \times 10^{-4}$, batch size is 32, and training is performed for 10 epochs.

**PatchTST** (Nie et al., 2023). PatchTST is a Transformer-based architecture employing patching and channel independence. Time series are divided into patches, which are transformed into tokens and processed by a transformer model to capture local and global dependencies via self-attention. This is particularly useful for long-term forecasting. Implementation is based on the NeuralForecast library. The learning rate is $1 \times 10^{-4}$, batch size is 128, and training is performed for 100 epochs.

**GPT4TS** (Zhou et al., 2023). GPT4TS treats time series as a language, leveraging pretrained language models (LLMs) to learn temporal patterns. Time series data is tokenized for LLM processing, enabling zero-shot or few-shot generalization. Implementation follows the source code[13], using long-term forecasting settings for the Weather dataset as default hyperparameters. The default language model is GPT-2 (Radford et al., 2019). The learning rate is $1 \times 10^{-4}$, batch size is 64, and training is performed for 10 epochs.

**AutoTimes** (Liu et al., 2024b). AutoTimes projects time series segments into the embedding space of language tokens, leveraging the autoregressive capabilities of LLMs for forecasting. By training the model to predict subsequent time series segments given preceding ones, AutoTimes generates multi-step forecasts. Implementation follows the source code[14], with GPT-2 (Radford et al., 2019) as the language model. The learning rate is $5 \times 10^{-4}$, batch size is 64, and training is performed for 10 epochs.

For all benchmark experiments, AdamW (Loshchilov & Hutter, 2019) is used as the optimizer, and the loss function is Mean Squared Error (MSE).

---

[10] https://docs.scipy.org/doc/scipy/reference/generated/scipy.spatial.Delaunay.html

[11] https://github.com/aikunyi/FourierGNN

[12] https://github.com/microsoft/StemGNN

[13] https://github.com/DAMO-DI-ML/NeurIPS2023-One-Fits-All

[14] https://github.com/thuml/AutoTimes/tree/main

### B.3 PLATFORM

All experiments are conducted on two Linux machines, one with 8 NVIDIA A100 GPUs, each with 40GB of memory, and another with 4 RTX 4090 GPUs. We used Python 3.12.9 and Pytorch 2.6.0 to construct our project.

## C DETAILED RESULTS

### C.1 DETAILED BENCHMARK RESULTS

In Section 4.5, we report the average of our benchmark results. In this section, we report the detailed results, including MSE, MAE, and three SEDI values on different prediction lengths, on three interest areas(Orlando, Miami, and Fort Myers) from Table 8 to 37. The best average results are presented in bold font, while the second-best are underlined. Furthermore, from Table 38 to 42, we report the detailed results of basic methods on the whole dataset. To provide qualitative analysis, from Figure 11 to 16, we demonstrate the case visualization of each method on the S7 split in the Orlando area.

### C.2 DETAILED OTHER RESULTS

To provide more information about the ablation study, we report the detailed results of the factor ablation study in Table 43 to 45. We report the average results on three interest areas on the S6 split. For the spatial and temporal information ablation study, the results are available from Table 46 to 49.

## D LIMITATIONS

While SF$^2$Bench provides five key factors, it currently lacks explicit topological linkage information between monitoring stations due to the intricate nature of South Florida's water system. Although we provide some flood observation data, the locations of these observations may not directly correspond to our monitoring stations. Therefore, this information is provided in a separate file rather than being integrated into the time series data.

## E LLM USAGE

In this paper, we leverage LLMs, including ChatGPT and Gemini 2.5 Pro, to refine sentence-level writing.

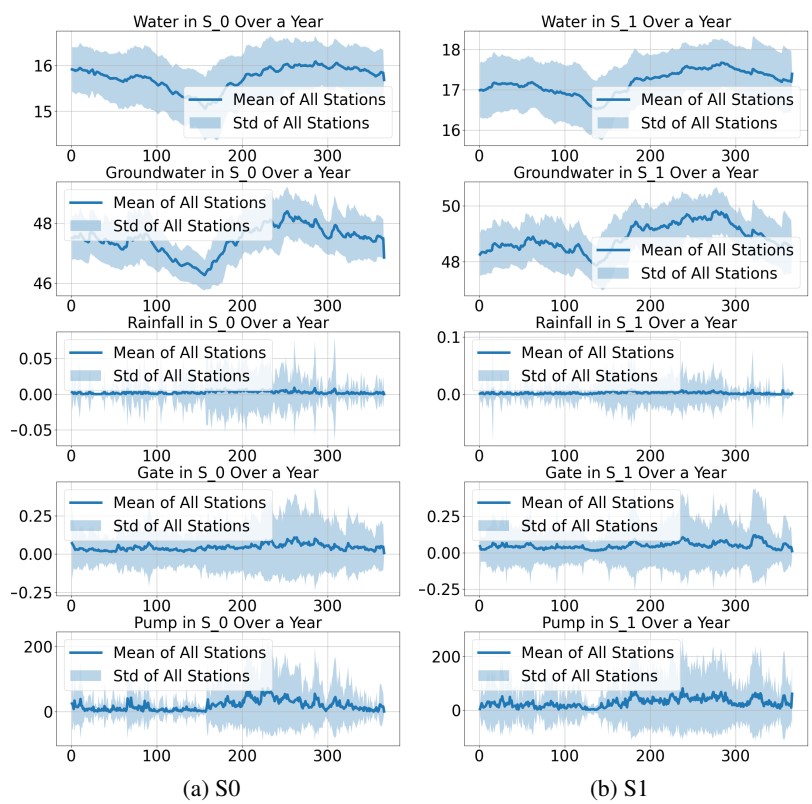

Figure 7: The temporal pattern of features of S0 and S1.

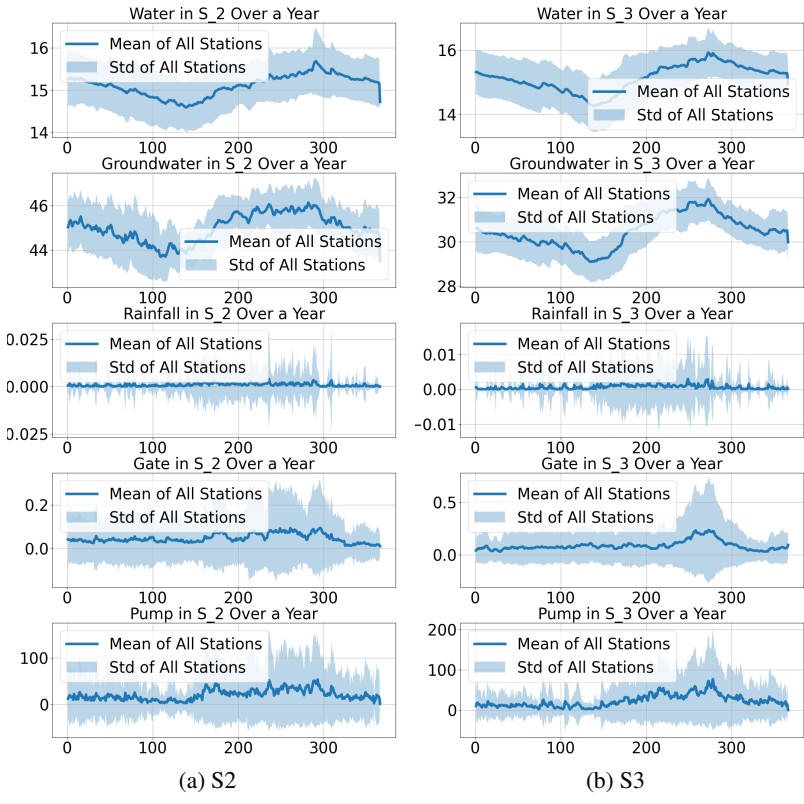

Figure 8: The temporal pattern of features of S2 and S3.

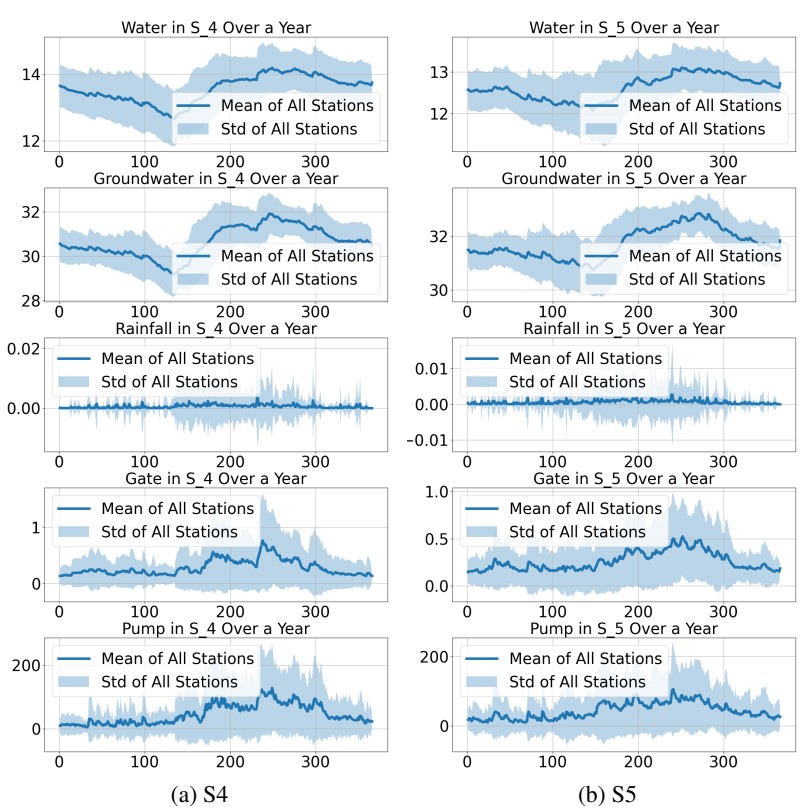

Figure 9: The temporal pattern of features of S4 and S5.

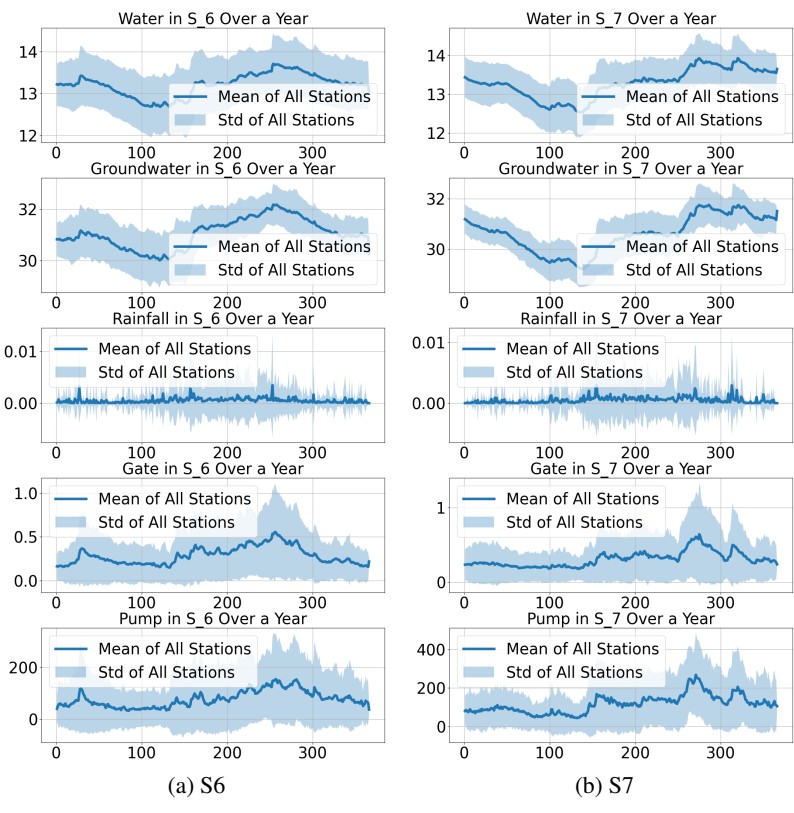

Figure 10: The temporal pattern of features of S6 and S7.

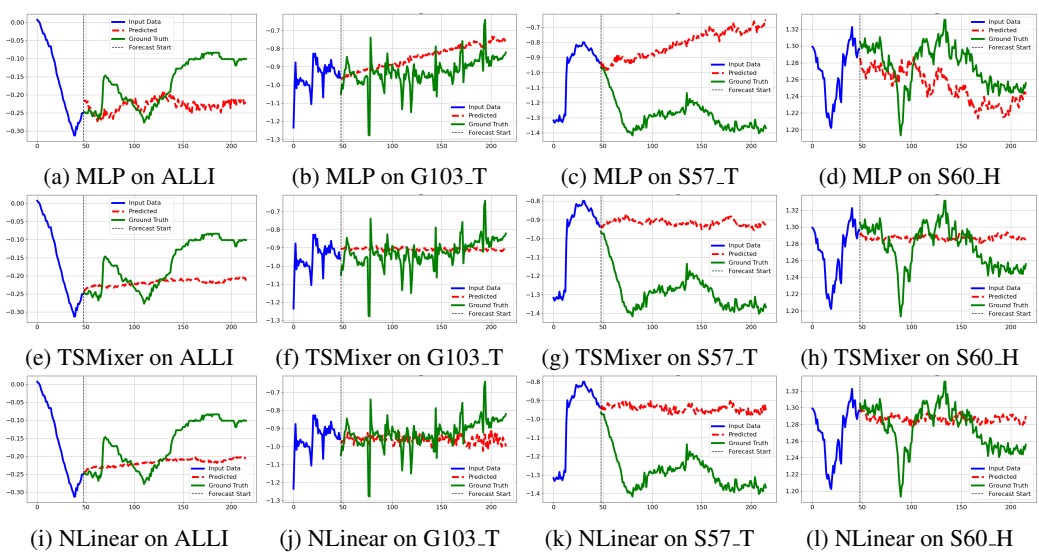

Figure 11: Cases visualization of MLP-based architecture methods on S7(Orlando area).

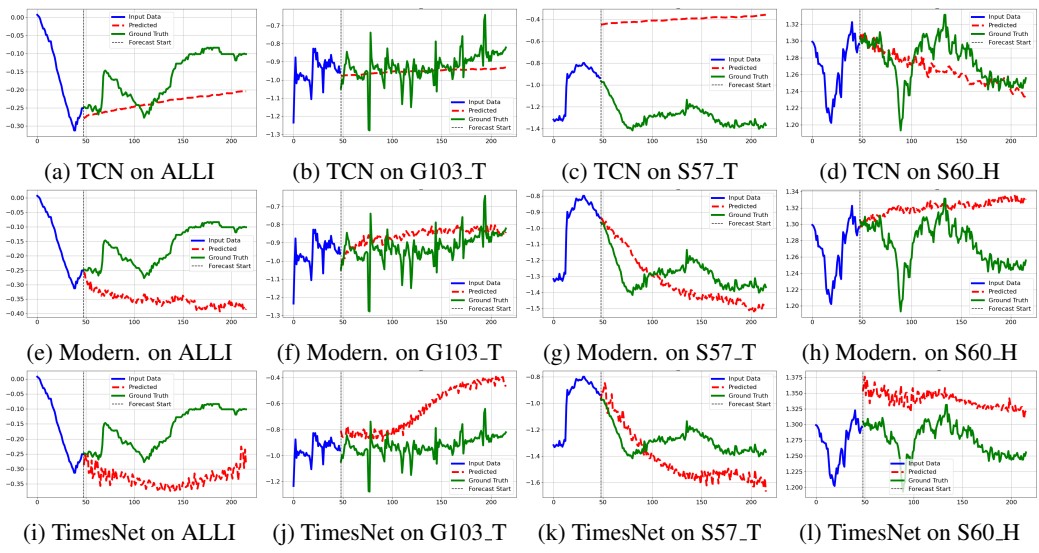

Figure 12: Cases visualization of CNN-based architecture methods on S7(Orlando area). Modern. means ModernTCN.

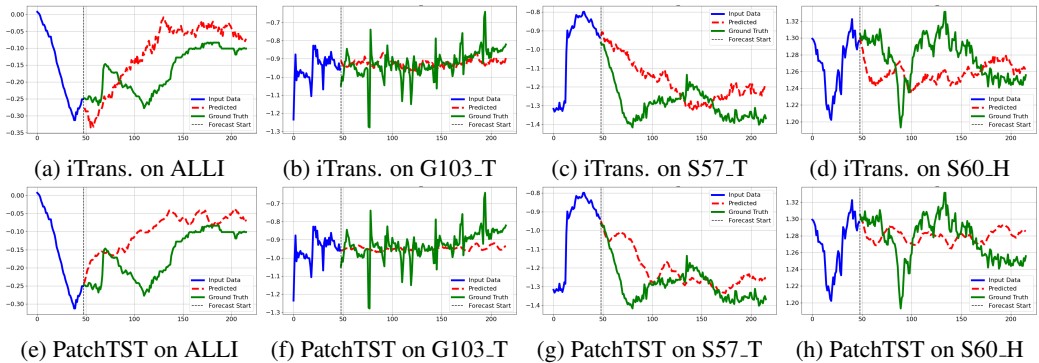

Figure 13: Cases visualization of Transformer-based architecture methods on S7(Orlando area). iTrans. means iTransformer.

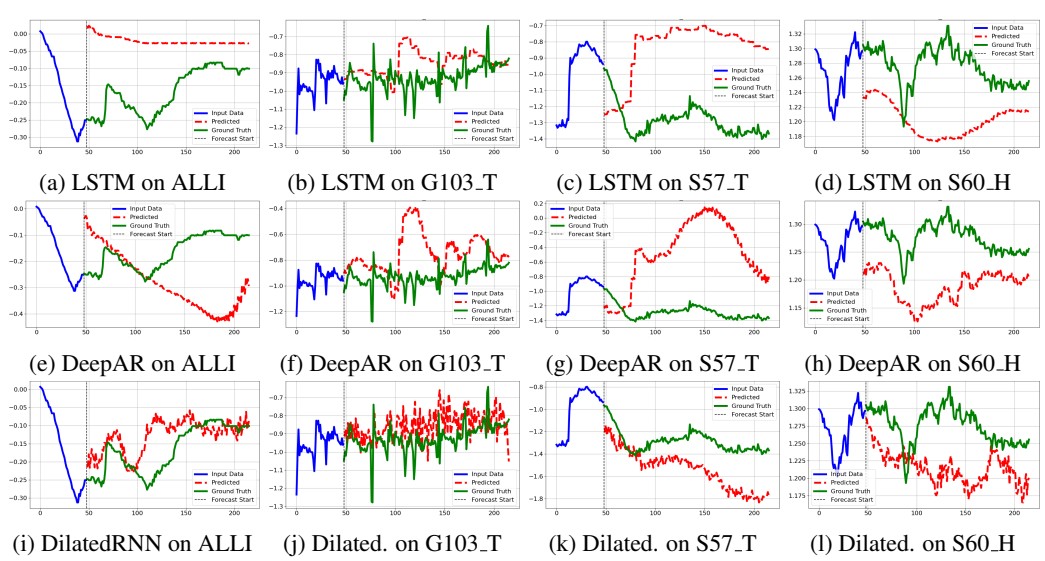

Figure 14: Cases visualization of RNN-based architecture methods on S7(Orlando area). Dilated. means DilatedRNN.

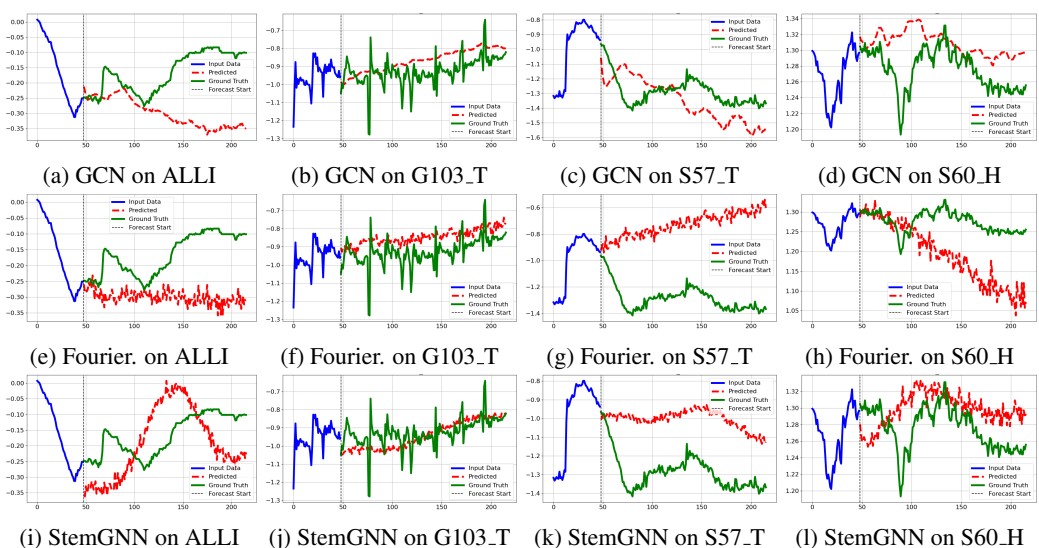

Figure 15: Cases visualization of GNN-based architecture methods on S7(Orlando area). Fourier. means FourierGNN.

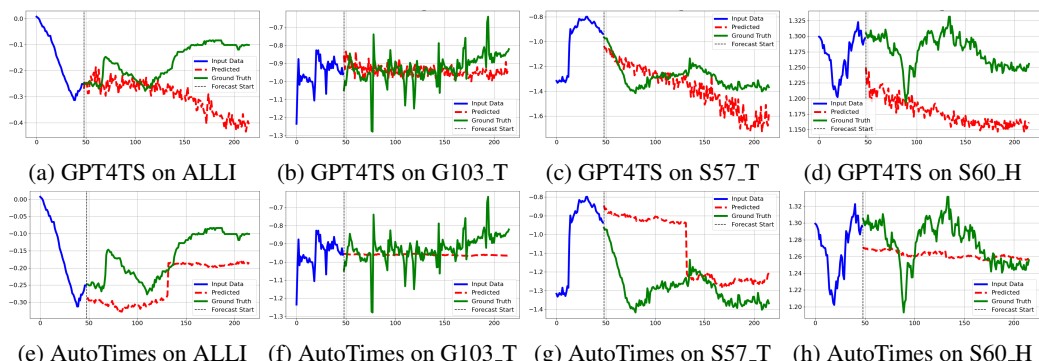

Figure 16: Cases visualization of LLM-based architecture methods on S7(Orlando area). Dilated. means DilatedRNN.

Table 8: Benchmark(MAE) on part 1(Orlando area) stations(MLP,CNN,Transformer)

| | Method | T | S0 | S1 | S2 | S3 | S4 | S5 | S6 | S7 | Avg. |
|---|---|---|---|---|---|---|---|---|---|---|---|
| **MLP** | MLP | 1D | 0.0915 | 0.1229 | 0.0679 | 0.0810 | 0.0716 | 0.0691 | 0.0598 | 0.0846 | 0.0810 |
| | | 3D | 0.1424 | 0.1928 | 0.1167 | 0.1437 | 0.1174 | 0.1154 | 0.1086 | 0.1314 | 0.1335 |
| | | 5D | 0.1746 | 0.2472 | 0.1607 | 0.1844 | 0.1503 | 0.1529 | 0.1418 | 0.1718 | 0.1730 |
| | | 7D | 0.1919 | 0.2899 | 0.1848 | 0.2244 | 0.1750 | 0.1793 | 0.1753 | 0.1970 | 0.2022 |
| | | Avg. | 0.1501 | 0.2132 | 0.1325 | 0.1584 | 0.1286 | **0.1292** | 0.1214 | 0.1462 | 0.1474 |
| | TSMixer | 1D | 0.1046 | 0.1355 | 0.0812 | 0.0846 | 0.0767 | 0.0778 | 0.1184 | 0.1015 | 0.0975 |
| | | 3D | 0.1494 | 0.2140 | 0.1252 | 0.1378 | 0.1185 | 0.1219 | 0.1500 | 0.1458 | 0.1453 |
| | | 5D | 0.1769 | 0.2673 | 0.1595 | 0.1783 | 0.1496 | 0.1547 | 0.1751 | 0.1798 | 0.1802 |
| | | 7D | 0.1938 | 0.3093 | 0.1892 | 0.2120 | 0.1765 | 0.1819 | 0.2098 | 0.2096 | 0.2103 |
| | | Avg. | 0.1562 | 0.2315 | 0.1388 | 0.1532 | 0.1303 | 0.1341 | 0.1633 | 0.1592 | 0.1583 |
| | NLinear | 1D | 0.0980 | 0.1279 | 0.0777 | 0.0820 | 0.0771 | 0.0741 | 0.0640 | 0.0929 | 0.0867 |
| | | 3D | 0.1475 | 0.2109 | 0.1234 | 0.1390 | 0.1207 | 0.1211 | 0.1079 | 0.1415 | 0.1390 |
| | | 5D | 0.1835 | 0.2663 | 0.1584 | 0.1814 | 0.1527 | 0.1541 | 0.1422 | 0.1763 | 0.1769 |
| | | 7D | 0.1964 | 0.3085 | 0.1891 | 0.2161 | 0.1803 | 0.1814 | 0.1711 | 0.2065 | 0.2062 |
| | | Avg. | 0.1563 | 0.2284 | 0.1371 | 0.1546 | 0.1327 | 0.1327 | 0.1213 | 0.1543 | 0.1522 |
| **CNN** | TCN | 1D | 0.1887 | 0.1417 | 0.2145 | 0.3830 | 0.1322 | 0.0889 | 0.1164 | 0.1913 | 0.1821 |
| | | 3D | 0.1588 | 0.2302 | 0.1857 | 0.4058 | 0.1968 | 0.1359 | 0.1471 | 0.2127 | 0.2091 |
| | | 5D | 0.2110 | 0.3028 | 0.2139 | 0.4483 | 0.2128 | 0.1928 | 0.2024 | 0.2579 | 0.2552 |
| | | 7D | 0.2135 | 0.3379 | 0.2605 | 0.5715 | 0.2686 | 0.2052 | 0.2102 | 0.2695 | 0.2921 |
| | | Avg. | 0.1930 | 0.2532 | 0.2186 | 0.4522 | 0.2026 | 0.1557 | 0.1690 | 0.2329 | 0.2346 |
| | ModernTCN | 1D | 0.0893 | 0.1238 | 0.0721 | 0.0760 | 0.0704 | 0.0661 | 0.0588 | 0.0802 | 0.0796 |
| | | 3D | 0.1486 | 0.2113 | 0.1366 | 0.1450 | 0.1212 | 0.1162 | 0.1073 | 0.1324 | 0.1398 |
| | | 5D | 0.1886 | 0.2685 | 0.1798 | 0.1887 | 0.1632 | 0.1537 | 0.1426 | 0.1697 | 0.1818 |
| | | 7D | 0.2140 | 0.3228 | 0.2240 | 0.2310 | 0.1936 | 0.1833 | 0.1728 | 0.2002 | 0.2177 |
| | | Avg. | 0.1601 | 0.2316 | 0.1531 | 0.1602 | 0.1371 | 0.1298 | 0.1204 | 0.1456 | 0.1547 |
| | TimesNet | 1D | 0.1068 | 0.1436 | 0.0805 | 0.0909 | 0.0919 | 0.0925 | 0.0787 | 0.1051 | 0.0987 |
| | | 3D | 0.1558 | 0.2190 | 0.1252 | 0.1515 | 0.1382 | 0.1402 | 0.1254 | 0.1582 | 0.1517 |
| | | 5D | 0.1813 | 0.2745 | 0.1637 | 0.1933 | 0.1735 | 0.1766 | 0.1578 | 0.1958 | 0.1896 |
| | | 7D | 0.2034 | 0.3222 | 0.2025 | 0.2309 | 0.2102 | 0.2146 | 0.1886 | 0.2263 | 0.2248 |
| | | Avg. | 0.1618 | 0.2398 | 0.1430 | 0.1666 | 0.1534 | 0.1560 | 0.1376 | 0.1714 | 0.1662 |
| **Transformer** | iTransformer | 1D | 0.0865 | 0.1227 | 0.0645 | 0.0730 | 0.0710 | 0.0700 | 0.0629 | 0.0889 | 0.0799 |
| | | 3D | 0.1395 | 0.1963 | 0.1103 | 0.1313 | 0.1193 | 0.1195 | 0.1058 | 0.1376 | 0.1325 |
| | | 5D | 0.1717 | 0.2546 | 0.1471 | 0.1723 | 0.1534 | 0.1528 | 0.1401 | 0.1775 | 0.1712 |
| | | 7D | 0.1889 | 0.2962 | 0.1801 | 0.2068 | 0.1833 | 0.1830 | 0.1708 | 0.2048 | 0.2017 |
| | | Avg. | 0.1466 | 0.2175 | 0.1255 | 0.1459 | 0.1317 | 0.1313 | 0.1199 | 0.1522 | 0.1463 |
| | PatchTST | 1D | 0.0843 | 0.1148 | 0.0638 | 0.0678 | 0.0711 | 0.0693 | 0.0617 | 0.0858 | 0.0773 |
| | | 3D | 0.1365 | 0.1922 | 0.1113 | 0.1269 | 0.1149 | 0.1199 | 0.1046 | 0.1362 | 0.1303 |
| | | 5D | 0.1677 | 0.2451 | 0.1471 | 0.1728 | 0.1489 | 0.1569 | 0.1409 | 0.1718 | 0.1689 |
| | | 7D | 0.1866 | 0.2927 | 0.1787 | 0.2054 | 0.1786 | 0.1859 | 0.1679 | 0.2019 | 0.1997 |
| | | Avg. | **0.1438** | **0.2112** | **0.1252** | **0.1432** | **0.1284** | 0.1330 | **0.1188** | 0.1489 | **0.1441** |

Table 9: Benchmark(MAE) on part 1(Orlando area) stations(RNN,GNN,LLM)

| | Method | T | S0 | S1 | S2 | S3 | S4 | S5 | S6 | S7 | Avg. |
|---|---|---|---|---|---|---|---|---|---|---|---|
| **RNN** | LSTM | 1D | 0.1275 | 0.1790 | 0.0935 | 0.1714 | 0.1044 | 0.1032 | 0.0865 | 0.1211 | 0.1233 |
| | | 3D | 0.1652 | 0.2751 | 0.1431 | 0.2458 | 0.1518 | 0.1528 | 0.1481 | 0.1647 | 0.1808 |
| | | 5D | 0.1972 | 0.3297 | 0.1801 | 0.2977 | 0.1871 | 0.1891 | 0.1784 | 0.1985 | 0.2197 |
| | | 7D | 0.2139 | 0.3517 | 0.2277 | 0.3351 | 0.2198 | 0.2143 | 0.1978 | 0.2266 | 0.2484 |
| | | Avg. | 0.1759 | 0.2839 | 0.1611 | 0.2625 | 0.1658 | 0.1648 | 0.1527 | 0.1777 | 0.1931 |
| | DeepAR | 1D | 0.1209 | 0.1855 | 0.0946 | 0.1726 | 0.1024 | 0.0971 | 0.0881 | 0.1171 | 0.1223 |
| | | 3D | 0.1667 | 0.2677 | 0.1488 | 0.2465 | 0.1534 | 0.1620 | 0.1421 | 0.1736 | 0.1826 |
| | | 5D | 0.1942 | 0.3110 | 0.1907 | 0.3237 | 0.1889 | 0.1828 | 0.1770 | 0.2045 | 0.2216 |
| | | 7D | 0.2146 | 0.3522 | 0.2210 | 0.3619 | 0.2122 | 0.2197 | 0.2123 | 0.2317 | 0.2532 |
| | | Avg. | 0.1741 | 0.2791 | 0.1638 | 0.2762 | 0.1642 | 0.1654 | 0.1548 | 0.1817 | 0.1949 |
| | DilatedRNN | 1D | 0.0939 | 0.1317 | 0.0700 | 0.1552 | 0.0836 | 0.0748 | 0.0672 | 0.1006 | 0.0971 |
| | | 3D | 0.1499 | 0.2318 | 0.1249 | 0.2031 | 0.1339 | 0.1318 | 0.1224 | 0.1457 | 0.1554 |
| | | 5D | 0.1794 | 0.2793 | 0.1627 | 0.2897 | 0.1678 | 0.1635 | 0.1520 | 0.1809 | 0.1969 |
| | | 7D | 0.2110 | 0.3329 | 0.1959 | 0.3061 | 0.1990 | 0.1945 | 0.1931 | 0.1478 | 0.2225 |
| | | Avg. | 0.1586 | 0.2439 | 0.1384 | 0.2385 | 0.1461 | 0.1411 | 0.1337 | **0.1437** | 0.1680 |
| **GNN** | GCN | 1D | 0.1003 | 0.2139 | 0.1034 | 0.3710 | 0.1030 | 0.1005 | 0.0775 | 0.0962 | 0.1457 |
| | | 3D | 0.1543 | 0.3137 | 0.1411 | 0.3384 | 0.1524 | 0.1517 | 0.1208 | 0.1392 | 0.1889 |
| | | 5D | 0.1862 | 0.3741 | 0.1665 | 0.4935 | 0.1893 | 0.1818 | 0.1610 | 0.1732 | 0.2407 |
| | | 7D | 0.2034 | 0.4156 | 0.1912 | 0.5967 | 0.2153 | 0.2073 | 0.1927 | 0.2024 | 0.2781 |
| | | Avg. | 0.1610 | 0.3293 | 0.1505 | 0.4499 | 0.1650 | 0.1603 | 0.1380 | 0.1528 | 0.2134 |
| | FourierGNN | 1D | 0.0985 | 0.1324 | 0.0796 | 0.1337 | 0.0863 | 0.0929 | 0.0672 | 0.0927 | 0.0979 |
| | | 3D | 0.1445 | 0.2034 | 0.1178 | 0.2733 | 0.1295 | 0.1246 | 0.1125 | 0.1392 | 0.1556 |
| | | 5D | 0.1746 | 0.2644 | 0.1524 | 0.3672 | 0.1712 | 0.1640 | 0.1513 | 0.1719 | 0.2021 |
| | | 7D | 0.1918 | 0.3028 | 0.1936 | 0.4107 | 0.1992 | 0.2019 | 0.1871 | 0.2068 | 0.2367 |
| | | Avg. | 0.1523 | 0.2257 | 0.1358 | 0.2962 | 0.1466 | 0.1459 | 0.1295 | 0.1527 | 0.1731 |
| | StemGNN | 1D | 0.1270 | 0.1782 | 0.0786 | 0.6896 | 0.1005 | 0.0833 | 0.0741 | 0.0969 | 0.1785 |
| | | 3D | 0.1717 | 0.3602 | 0.1536 | 0.6742 | 0.1778 | 0.1720 | 0.1498 | 0.1715 | 0.2539 |
| | | 5D | 0.2081 | 0.4895 | 0.2141 | 1.4591 | 0.2144 | 0.2729 | 0.2258 | 0.2412 | 0.4156 |
| | | 7D | 0.2477 | 0.5122 | 0.2577 | 1.4988 | 0.2552 | 0.3307 | 0.2486 | 0.2419 | 0.4491 |
| | | Avg. | 0.1886 | 0.3850 | 0.1760 | 1.0804 | 0.1870 | 0.2147 | 0.1746 | 0.1879 | 0.3243 |
| **LLM** | GPT4TS | 1D | 0.1050 | 0.1372 | 0.0833 | 0.0934 | 0.0905 | 0.0898 | 0.0778 | 0.1106 | 0.0984 |
| | | 3D | 0.1530 | 0.2169 | 0.1221 | 0.1502 | 0.1335 | 0.1380 | 0.1222 | 0.1547 | 0.1488 |
| | | 5D | 0.1817 | 0.2650 | 0.1565 | 0.1928 | 0.1666 | 0.1723 | 0.1559 | 0.1948 | 0.1857 |
| | | 7D | 0.1967 | 0.3105 | 0.1930 | 0.2294 | 0.1969 | 0.2054 | 0.1962 | 0.2224 | 0.2188 |
| | | Avg. | 0.1591 | 0.2324 | 0.1387 | 0.1664 | 0.1469 | 0.1514 | 0.1380 | 0.1706 | 0.1629 |
| | AutoTimes | 1D | 0.0967 | 0.1369 | 0.0747 | 0.0770 | 0.0768 | 0.0784 | 0.0660 | 0.0963 | 0.0878 |
| | | 3D | 0.1453 | 0.2057 | 0.1196 | 0.1296 | 0.1164 | 0.1203 | 0.1099 | 0.1409 | 0.1360 |
| | | 5D | 0.1733 | 0.2664 | 0.1499 | 0.1751 | 0.1498 | 0.1534 | 0.1411 | 0.1739 | 0.1729 |
| | | 7D | 0.1902 | 0.3029 | 0.1814 | 0.2113 | 0.1780 | 0.1823 | 0.1691 | 0.2032 | 0.2023 |
| | | Avg. | 0.1514 | 0.2280 | 0.1314 | 0.1482 | 0.1302 | 0.1336 | 0.1215 | 0.1536 | 0.1497 |

Table 10: Benchmark(MSE) on part 1(Orlando area) stations(MLP,CNN,Transformer)

| | Method | T | S0 | S1 | S2 | S3 | S4 | S5 | S6 | S7 | Avg. |
|---|---|---|---|---|---|---|---|---|---|---|---|
| **MLP** | MLP | 1D | 0.1472 | 0.1254 | 0.0376 | 0.0610 | 0.0379 | 0.0275 | 0.0248 | 0.0523 | 0.0642 |
| | | 3D | 0.2661 | 0.2014 | 0.0621 | 0.1232 | 0.0681 | 0.0536 | 0.0465 | 0.0913 | 0.1140 |
| | | 5D | 0.3569 | 0.2609 | 0.0875 | 0.1714 | 0.0999 | 0.0772 | 0.0670 | 0.1226 | 0.1554 |
| | | 7D | 0.2963 | 0.3124 | 0.1097 | 0.2215 | 0.1199 | 0.0975 | 0.0871 | 0.1498 | 0.1743 |
| | | Avg. | 0.2666 | **0.2250** | **0.0742** | 0.1443 | **0.0814** | **0.0640** | **0.0564** | 0.1040 | **0.1270** |
| | TSMixer | 1D | 0.1649 | 0.1283 | 0.0423 | 0.0696 | 0.0380 | 0.0312 | 0.0661 | 0.0633 | 0.0755 |
| | | 3D | 0.2915 | 0.2214 | 0.0679 | 0.1255 | 0.0694 | 0.0575 | 0.0862 | 0.1000 | 0.1274 |
| | | 5D | 0.1769 | 0.2903 | 0.0936 | 0.1673 | 0.0981 | 0.0812 | 0.1010 | 0.1325 | 0.1626 |
| | | 7D | 0.2817 | 0.3469 | 0.1190 | 0.2048 | 0.1248 | 0.1031 | 0.1384 | 0.1644 | 0.1854 |
| | | Avg. | 0.2688 | 0.2467 | 0.0807 | **0.1418** | 0.0826 | 0.0682 | 0.0979 | 0.1150 | 0.1377 |
| | NLinear | 1D | 0.1468 | 0.1294 | 0.0449 | 0.0678 | 0.0405 | 0.0308 | 0.0291 | 0.0580 | 0.0684 |
| | | 3D | 0.2830 | 0.2223 | 0.0699 | 0.1257 | 0.0720 | 0.0575 | 0.0510 | 0.0974 | 0.1223 |
| | | 5D | 0.4103 | 0.2915 | 0.0950 | 0.1689 | 0.1013 | 0.0812 | 0.0722 | 0.1300 | 0.1688 |
| | | 7D | 0.3001 | 0.3481 | 0.1204 | 0.2080 | 0.1291 | 0.1029 | 0.0924 | 0.1617 | 0.1828 |
| | | Avg. | 0.2851 | 0.2478 | 0.0825 | 0.1426 | 0.0857 | 0.0681 | 0.0612 | 0.1118 | 0.1356 |
| **CNN** | TCN | 1D | 0.3257 | 0.1416 | 0.1462 | 0.8044 | 0.1152 | 0.0344 | 0.0567 | 0.1260 | 0.2188 |
| | | 3D | 0.2475 | 0.2295 | 0.1342 | 0.9118 | 0.1694 | 0.0597 | 0.0731 | 0.1423 | 0.2459 |
| | | 5D | 0.2906 | 0.3102 | 0.1367 | 0.6579 | 0.1763 | 0.1011 | 0.1119 | 0.1884 | 0.2466 |
| | | 7D | 0.3160 | 0.3587 | 0.1805 | 1.3613 | 0.2556 | 0.1105 | 0.1087 | 0.2119 | 0.3629 |
| | | Avg. | 0.2950 | 0.2600 | 0.1494 | 0.9338 | 0.1791 | 0.0764 | 0.0876 | 0.1671 | 0.2686 |
| | ModernTCN | 1D | 0.1391 | 0.1432 | 0.0407 | 0.0778 | 0.0392 | 0.0279 | 0.0255 | 0.0520 | 0.0682 |
| | | 3D | 0.2746 | 0.2966 | 0.1185 | 0.1814 | 0.0799 | 0.0575 | 0.0505 | 0.0921 | 0.1439 |
| | | 5D | 0.3575 | 0.3305 | 0.1278 | 0.2059 | 0.1303 | 0.0850 | 0.0723 | 0.1257 | 0.1794 |
| | | 7D | 0.3067 | 0.4557 | 0.1821 | 0.2844 | 0.1657 | 0.1096 | 0.0950 | 0.1579 | 0.2196 |
| | | Avg. | 0.2695 | 0.3065 | 0.1173 | 0.1874 | 0.1038 | 0.0700 | 0.0608 | 0.1069 | 0.1528 |
| | TimesNet | 1D | 0.1770 | 0.1473 | 0.0429 | 0.0742 | 0.0546 | 0.0449 | 0.0394 | 0.0761 | 0.0821 |
| | | 3D | 0.3474 | 0.2545 | 0.0758 | 0.1626 | 0.1049 | 0.0829 | 0.0737 | 0.1365 | 0.1548 |
| | | 5D | 0.3436 | 0.3207 | 0.1069 | 0.2093 | 0.1475 | 0.1193 | 0.0939 | 0.1907 | 0.1915 |
| | | 7D | 0.3053 | 0.4134 | 0.1426 | 0.2522 | 0.2020 | 0.1648 | 0.1283 | 0.2267 | 0.2294 |
| | | Avg. | 0.2933 | 0.2840 | 0.0921 | 0.1746 | 0.1273 | 0.1030 | 0.0838 | 0.1575 | 0.1644 |
| **Transformer** | iTransformer | 1D | 0.1277 | 0.1337 | 0.0357 | 0.0674 | 0.0382 | 0.0314 | 0.0295 | 0.0619 | 0.0657 |
| | | 3D | 0.2760 | 0.2269 | 0.0688 | 0.1343 | 0.0755 | 0.0627 | 0.0529 | 0.1057 | 0.1254 |
| | | 5D | 0.3581 | 0.3195 | 0.1004 | 0.1846 | 0.1081 | 0.0855 | 0.0769 | 0.1483 | 0.1727 |
| | | 7D | 0.2999 | 0.3668 | 0.1288 | 0.2222 | 0.1384 | 0.1129 | 0.1003 | 0.1810 | 0.1938 |
| | | Avg. | 0.2654 | 0.2617 | 0.0834 | 0.1521 | 0.0901 | 0.0731 | 0.0649 | 0.1242 | 0.1394 |
| | PatchTST | 1D | 0.1315 | 0.1368 | 0.0370 | 0.0608 | 0.0385 | 0.0325 | 0.0321 | 0.0593 | 0.0661 |
| | | 3D | 0.3008 | 0.2298 | 0.0845 | 0.1276 | 0.0713 | 0.0645 | 0.0535 | 0.1041 | 0.1295 |
| | | 5D | 0.3504 | 0.2951 | 0.1222 | 0.1736 | 0.1042 | 0.0946 | 0.0777 | 0.1368 | 0.1693 |
| | | 7D | 0.3145 | 0.3605 | 0.1365 | 0.2170 | 0.1412 | 0.1198 | 0.0968 | 0.1708 | 0.1946 |
| | | Avg. | 0.2743 | 0.2555 | 0.0951 | 0.1448 | 0.0888 | 0.0778 | 0.0650 | 0.1177 | 0.1399 |

Table 11: Benchmark(MSE) on part 1(Orlando area) stations(RNN,GNN,LLM)

| | Method | T | S0 | S1 | S2 | S3 | S4 | S5 | S6 | S7 | Avg. |
|---|---|---|---|---|---|---|---|---|---|---|---|
| **RNN** | LSTM | 1D | 0.2408 | 0.2451 | 0.0505 | 0.2788 | 0.0655 | 0.0496 | 0.0435 | 0.0846 | 0.1323 |
| | | 3D | 0.2765 | 0.3219 | 0.0784 | 0.3653 | 0.1097 | 0.0768 | 0.0837 | 0.1162 | 0.1786 |
| | | 5D | 0.3053 | 0.3853 | 0.1085 | 0.4750 | 0.1569 | 0.1011 | 0.0953 | 0.1525 | 0.2225 |
| | | 7D | 0.3058 | 0.4057 | 0.1427 | 0.5133 | 0.1855 | 0.1226 | 0.1103 | 0.1818 | 0.2459 |
| | | Avg. | 0.2821 | 0.3395 | 0.0950 | 0.4081 | 0.1294 | 0.0875 | 0.0832 | 0.1338 | 0.1948 |
| | DeepAR | 1D | 0.2406 | 0.2367 | 0.0480 | 0.2190 | 0.0615 | 0.0435 | 0.0386 | 0.0782 | 0.1208 |
| | | 3D | 0.2720 | 0.3088 | 0.0793 | 0.4084 | 0.1117 | 0.0783 | 0.0652 | 0.1212 | 0.1806 |
| | | 5D | 0.3033 | 0.3441 | 0.1072 | 0.5899 | 0.1537 | 0.1003 | 0.0883 | 0.1564 | 0.2304 |
| | | 7D | 0.3046 | 0.4056 | 0.1356 | 0.5250 | 0.1639 | 0.1221 | 0.1070 | 0.1800 | 0.2430 |
| | | Avg. | 0.2801 | 0.3238 | 0.0925 | 0.4356 | 0.1227 | 0.0860 | 0.0748 | 0.1340 | 0.1937 |
| | DilatedRNN | 1D | 0.1818 | 0.1370 | 0.0372 | 0.3297 | 0.0496 | 0.0356 | 0.0304 | 0.0860 | 0.1109 |
| | | 3D | 0.2517 | 0.2674 | 0.0689 | 0.3124 | 0.0887 | 0.0645 | 0.0722 | 0.1150 | 0.1551 |
| | | 5D | 0.2822 | 0.3308 | 0.1004 | 0.6438 | 0.1338 | 0.0915 | 0.0916 | 0.1423 | 0.2270 |
| | | 7D | 0.3125 | 0.3989 | 0.1242 | 0.5068 | 0.1702 | 0.1159 | 0.1257 | 0.0714 | 0.2282 |
| | | Avg. | 0.2571 | 0.2835 | 0.0827 | 0.4482 | 0.1106 | 0.0769 | 0.0800 | 0.1037 | 0.1803 |
| **GNN** | GCN | 1D | 0.1465 | 0.5157 | 0.0443 | 1.1053 | 0.0519 | 0.0353 | 0.0275 | 0.0518 | 0.2473 |
| | | 3D | 0.2526 | 1.6031 | 0.0656 | 0.7137 | 0.0944 | 0.0654 | 0.0494 | 0.0891 | 0.3667 |
| | | 5D | 0.3055 | 1.9044 | 0.0867 | 1.1121 | 0.1266 | 0.0895 | 0.0730 | 0.1184 | 0.4770 |
| | | 7D | 0.2687 | 2.2856 | 0.1071 | 1.4113 | 0.1651 | 0.1087 | 0.0928 | 0.1471 | 0.5733 |
| | | Avg. | **0.2433** | 1.5772 | 0.0759 | 1.0856 | 0.1095 | 0.0747 | 0.0607 | **0.1016** | 0.4161 |
| | FourierGNN | 1D | 0.1619 | 0.1324 | 0.0412 | 0.0901 | 0.0442 | 0.0326 | 0.0275 | 0.0575 | 0.0734 |
| | | 3D | 0.2714 | 0.2058 | 0.0649 | 0.2944 | 0.0746 | 0.0560 | 0.0489 | 0.0941 | 0.1388 |
| | | 5D | 0.3066 | 0.2721 | 0.0891 | 0.4409 | 0.1106 | 0.0806 | 0.0705 | 0.1238 | 0.1868 |
| | | 7D | 0.2766 | 0.3139 | 0.1154 | 0.4880 | 0.1381 | 0.1065 | 0.0919 | 0.1528 | 0.2104 |
| | | Avg. | 0.2541 | 0.2310 | 0.0777 | 0.3283 | 0.0919 | 0.0689 | 0.0597 | 0.1071 | 0.1523 |
| | StemGNN | 1D | 0.2290 | 0.1703 | 0.0403 | 1.1688 | 0.0652 | 0.0410 | 0.0325 | 0.0723 | 0.2274 |
| | | 3D | 0.2703 | 0.4534 | 0.0903 | 1.2569 | 0.1323 | 0.0896 | 0.0743 | 0.1263 | 0.3117 |
| | | 5D | 0.3017 | 0.7711 | 0.1305 | 3.3143 | 0.1663 | 0.1663 | 0.1417 | 0.2080 | 0.6500 |
| | | 7D | 0.3225 | 0.7901 | 0.1768 | 3.4285 | 0.1979 | 0.2323 | 0.1489 | 0.1864 | 0.6854 |
| | | Avg. | 0.2809 | 0.5462 | 0.1095 | 2.2921 | 0.1404 | 0.1323 | 0.0994 | 0.1483 | 0.4686 |
| **LLM** | GPT4TS | 1D | 0.1984 | 0.1399 | 0.0456 | 0.0882 | 0.0542 | 0.0431 | 0.0387 | 0.0935 | 0.0877 |
| | | 3D | 0.3883 | 0.2385 | 0.0811 | 0.1648 | 0.0986 | 0.0788 | 0.0690 | 0.1343 | 0.1567 |
| | | 5D | 0.3668 | 0.3102 | 0.0997 | 0.2037 | 0.1304 | 0.1091 | 0.0870 | 0.1876 | 0.1868 |
| | | 7D | 0.2928 | 0.3700 | 0.1378 | 0.2503 | 0.1622 | 0.1494 | 0.1319 | 0.2233 | 0.2147 |
| | | Avg. | 0.3116 | 0.2646 | 0.0911 | 0.1768 | 0.1114 | 0.0951 | 0.0816 | 0.1597 | 0.1615 |
| | AutoTimes | 1D | 0.1531 | 0.1507 | 0.0401 | 0.0671 | 0.0390 | 0.0310 | 0.0287 | 0.0623 | 0.0715 |
| | | 3D | 0.3072 | 0.2333 | 0.0683 | 0.1278 | 0.0686 | 0.0572 | 0.0508 | 0.0978 | 0.1264 |
| | | 5D | 0.3527 | 0.2975 | 0.0916 | 0.1700 | 0.0992 | 0.0811 | 0.0715 | 0.1319 | 0.1619 |
| | | 7D | 0.2913 | 0.3428 | 0.1181 | 0.2061 | 0.1265 | 0.1031 | 0.0916 | 0.1612 | 0.1801 |
| | | Avg. | 0.2761 | 0.2561 | 0.0795 | 0.1427 | 0.0833 | 0.0681 | 0.0606 | 0.1133 | 0.1350 |

Table 12: Benchmark(SEDI(10%)) on part 1(Orlando area) stations(MLP,CNN,Transformer)

| | Method | T | S0 | S1 | S2 | S3 | S4 | S5 | S6 | S7 | Avg. |
|---|---|---|---|---|---|---|---|---|---|---|---|
| **MLP** | MLP | 1D | 0.8143 | 0.9334 | 0.9011 | 0.8780 | 0.6133 | 0.7659 | 0.9351 | 0.7772 | 0.8273 |
| | | 3D | 0.7599 | 0.8673 | 0.8592 | 0.9286 | 0.5201 | 0.7173 | 0.9040 | 0.7097 | 0.7833 |
| | | 5D | 0.6435 | 0.8095 | 0.8142 | 0.8702 | 0.4517 | 0.6956 | 0.8571 | 0.6573 | 0.7249 |
| | | 7D | 0.6022 | 0.8056 | 0.7759 | 0.8158 | 0.4198 | 0.6622 | 0.8516 | 0.5969 | 0.6912 |
| | | Avg. | **0.7050** | **0.8540** | **0.8376** | **0.8731** | **0.5012** | **0.7103** | **0.8870** | **0.6853** | **0.7567** |
| | TSMixer | 1D | 0.7483 | 0.8690 | 0.8917 | 0.9141 | 0.5971 | 0.7394 | 0.7901 | 0.7191 | 0.7836 |
| | | 3D | 0.5937 | 0.7493 | 0.8103 | 0.8420 | 0.4798 | 0.6766 | 0.7626 | 0.6334 | 0.6935 |
| | | 5D | 0.5167 | 0.6755 | 0.7538 | 0.7783 | 0.4139 | 0.6214 | 0.7289 | 0.5685 | 0.6321 |
| | | 7D | 0.4706 | 0.6241 | 0.7042 | 0.7236 | 0.3734 | 0.5854 | 0.6798 | 0.5107 | 0.5840 |
| | | Avg. | 0.5823 | 0.7295 | 0.7900 | 0.8145 | 0.4660 | 0.6557 | 0.7403 | 0.6079 | 0.6733 |
| | NLinear | 1D | 0.7729 | 0.8803 | 0.8951 | 0.9171 | 0.5984 | 0.7844 | 0.9259 | 0.7302 | 0.8130 |
| | | 3D | 0.6007 | 0.7581 | 0.8166 | 0.8390 | 0.4784 | 0.6961 | 0.8550 | 0.6410 | 0.7106 |
| | | 5D | 0.5198 | 0.6821 | 0.7576 | 0.7721 | 0.4129 | 0.6535 | 0.7956 | 0.5750 | 0.6461 |
| | | 7D | 0.4731 | 0.6290 | 0.7063 | 0.7159 | 0.3701 | 0.6155 | 0.7512 | 0.5173 | 0.5973 |
| | | Avg. | 0.5916 | 0.7374 | 0.7939 | 0.8110 | 0.4649 | 0.6873 | 0.8319 | 0.6159 | 0.6918 |
| **CNN** | TCN | 1D | 0.5634 | 0.8939 | 0.7243 | 0.7307 | 0.4343 | 0.7156 | 0.8132 | 0.5039 | 0.6724 |
| | | 3D | 0.5462 | 0.8010 | 0.6694 | 0.7131 | 0.3092 | 0.7072 | 0.7778 | 0.3687 | 0.6116 |
| | | 5D | 0.4606 | 0.7638 | 0.6291 | 0.6572 | 0.2804 | 0.6123 | 0.7136 | 0.3351 | 0.5565 |
| | | 7D | 0.4598 | 0.7014 | 0.6092 | 0.5674 | 0.2201 | 0.5762 | 0.5692 | 0.3620 | 0.5082 |
| | | Avg. | 0.5075 | 0.7900 | 0.6580 | 0.6671 | 0.3110 | 0.6528 | 0.7184 | 0.3924 | 0.5872 |
| | ModernTCN | 1D | 0.7912 | 0.8737 | 0.8857 | 0.8915 | 0.6023 | 0.7692 | 0.9240 | 0.7395 | 0.8096 |
| | | 3D | 0.6031 | 0.7637 | 0.7893 | 0.8041 | 0.4970 | 0.6746 | 0.8469 | 0.6370 | 0.7020 |
| | | 5D | 0.4960 | 0.6839 | 0.7201 | 0.7403 | 0.4277 | 0.6142 | 0.7898 | 0.5668 | 0.6298 |
| | | 7D | 0.4357 | 0.6233 | 0.6667 | 0.6818 | 0.3869 | 0.5726 | 0.7499 | 0.5076 | 0.5781 |
| | | Avg. | 0.5815 | 0.7362 | 0.7654 | 0.7794 | 0.4785 | 0.6576 | 0.8277 | 0.6127 | 0.6799 |
| | TimesNet | 1D | 0.7224 | 0.8584 | 0.8701 | 0.8789 | 0.5727 | 0.7000 | 0.9025 | 0.7044 | 0.7762 |
| | | 3D | 0.5820 | 0.7493 | 0.7919 | 0.7959 | 0.4368 | 0.6345 | 0.7953 | 0.5956 | 0.6727 |
| | | 5D | 0.4866 | 0.6771 | 0.7336 | 0.7370 | 0.3854 | 0.5657 | 0.7453 | 0.5191 | 0.6062 |
| | | 7D | 0.4083 | 0.6298 | 0.6863 | 0.6808 | 0.3325 | 0.5181 | 0.7137 | 0.4662 | 0.5545 |
| | | Avg. | 0.5498 | 0.7286 | 0.7705 | 0.7732 | 0.4318 | 0.6045 | 0.7892 | 0.5713 | 0.6524 |
| **Transformer** | iTransformer | 1D | 0.8049 | 0.8881 | 0.9039 | 0.9057 | 0.5997 | 0.7663 | 0.9234 | 0.7186 | 0.8138 |
| | | 3D | 0.6433 | 0.7820 | 0.8294 | 0.8350 | 0.4807 | 0.6836 | 0.8500 | 0.6270 | 0.7164 |
| | | 5D | 0.5540 | 0.7202 | 0.7733 | 0.7744 | 0.4121 | 0.6302 | 0.7956 | 0.5481 | 0.6510 |
| | | 7D | 0.5043 | 0.6631 | 0.7177 | 0.7288 | 0.3558 | 0.5843 | 0.7486 | 0.4997 | 0.6003 |
| | | Avg. | 0.6266 | 0.7634 | 0.8061 | 0.8110 | 0.4621 | 0.6661 | 0.8294 | 0.5983 | 0.6954 |
| | PatchTST | 1D | 0.8082 | 0.8872 | 0.8947 | 0.9082 | 0.5937 | 0.7683 | 0.9141 | 0.7252 | 0.8125 |
| | | 3D | 0.6607 | 0.7851 | 0.8203 | 0.8352 | 0.4994 | 0.6789 | 0.8537 | 0.6293 | 0.7203 |
| | | 5D | 0.5672 | 0.7188 | 0.7626 | 0.7659 | 0.4175 | 0.6272 | 0.7914 | 0.5625 | 0.6516 |
| | | 7D | 0.5081 | 0.6574 | 0.7167 | 0.7186 | 0.3708 | 0.5887 | 0.7506 | 0.5049 | 0.6020 |
| | | Avg. | 0.6361 | 0.7621 | 0.7986 | 0.8070 | 0.4704 | 0.6658 | 0.8274 | 0.6055 | 0.6966 |

Table 13: Benchmark(SEDI(10%)) on part 1(Orlando area) stations(RNN, GNN,LLM)

| | Method | T | S0 | S1 | S2 | S3 | S4 | S5 | S6 | S7 | Avg. |
|---|---|---|---|---|---|---|---|---|---|---|---|
| **RNN** | LSTM | 1D | 0.7709 | 0.8270 | 0.8825 | 0.8781 | 0.5134 | 0.7182 | 0.9039 | 0.7302 | 0.7780 |
| | | 3D | 0.6027 | 0.8070 | 0.8162 | 0.8833 | 0.4448 | 0.5963 | 0.8482 | 0.6473 | 0.7057 |
| | | 5D | 0.5247 | 0.7396 | 0.7646 | 0.7848 | 0.3667 | 0.6529 | 0.7852 | 0.5842 | 0.6503 |
| | | 7D | 0.4409 | 0.6678 | 0.7459 | 0.7427 | 0.3836 | 0.6063 | 0.7591 | 0.4874 | 0.6042 |
| | | Avg. | 0.5848 | 0.7604 | 0.8023 | 0.8222 | 0.4271 | 0.6434 | 0.8241 | 0.6123 | 0.6846 |
| | DeepAR | 1D | 0.7411 | 0.8286 | 0.9006 | 0.9469 | 0.5732 | 0.7033 | 0.9113 | 0.7681 | 0.7966 |
| | | 3D | 0.6168 | 0.7816 | 0.7859 | 0.7710 | 0.4175 | 0.6046 | 0.8704 | 0.6535 | 0.6877 |
| | | 5D | 0.5086 | 0.7014 | 0.7872 | 0.8217 | 0.3716 | 0.6180 | 0.7106 | 0.6096 | 0.6411 |
| | | 7D | 0.4686 | 0.7151 | 0.7627 | 0.8162 | 0.3346 | 0.6131 | 0.6993 | 0.5065 | 0.6145 |
| | | Avg. | 0.5838 | 0.7567 | 0.8091 | 0.8390 | 0.4242 | 0.6348 | 0.7979 | 0.6344 | 0.6850 |
| | DilatedRNN | 1D | 0.8033 | 0.9139 | 0.8972 | 0.9067 | 0.6086 | 0.7706 | 0.9528 | 0.7437 | 0.8246 |
| | | 3D | 0.6053 | 0.8119 | 0.7907 | 0.8376 | 0.4752 | 0.5884 | 0.8004 | 0.6515 | 0.6951 |
| | | 5D | 0.5254 | 0.7325 | 0.7443 | 0.8017 | 0.4259 | 0.6454 | 0.8208 | 0.6270 | 0.6654 |
| | | 7D | 0.4664 | 0.7255 | 0.7337 | 0.7261 | 0.4031 | 0.6275 | 0.7647 | 0.5970 | 0.6305 |
| | | Avg. | 0.6001 | 0.7960 | 0.7915 | 0.8180 | 0.4782 | 0.6580 | 0.8347 | 0.6548 | 0.7039 |
| **GNN** | GCN | 1D | 0.8018 | 0.7777 | 0.8496 | 0.7072 | 0.5680 | 0.7274 | 0.9103 | 0.7172 | 0.7574 |
| | | 3D | 0.6240 | 0.7415 | 0.8169 | 0.6969 | 0.5115 | 0.6977 | 0.8412 | 0.6788 | 0.7011 |
| | | 5D | 0.5324 | 0.7147 | 0.7394 | 0.6223 | 0.4217 | 0.6488 | 0.8378 | 0.6075 | 0.6406 |
| | | 7D | 0.4949 | 0.6795 | 0.6887 | 0.5567 | 0.3824 | 0.6114 | 0.7941 | 0.5809 | 0.5986 |
| | | Avg. | 0.6133 | 0.7283 | 0.7737 | 0.6458 | 0.4709 | 0.6713 | 0.8458 | 0.6461 | 0.6744 |
| | FourierGNN | 1D | 0.8505 | 0.8952 | 0.9035 | 0.8991 | 0.5899 | 0.7981 | 0.9211 | 0.7834 | 0.8301 |
| | | 3D | 0.6929 | 0.8059 | 0.8177 | 0.7987 | 0.5029 | 0.6747 | 0.8521 | 0.6853 | 0.7288 |
| | | 5D | 0.6012 | 0.7985 | 0.7753 | 0.7296 | 0.4659 | 0.6222 | 0.8349 | 0.6308 | 0.6823 |
| | | 7D | 0.5372 | 0.7731 | 0.7619 | 0.6927 | 0.4257 | 0.6830 | 0.8133 | 0.5499 | 0.6546 |
| | | Avg. | 0.6705 | 0.8182 | 0.8146 | 0.7800 | 0.4961 | 0.6945 | 0.8554 | 0.6624 | 0.7240 |
| | StemGNN | 1D | 0.6812 | 0.8950 | 0.8559 | 0.4137 | 0.5486 | 0.7748 | 0.9207 | 0.7578 | 0.7310 |
| | | 3D | 0.5934 | 0.7234 | 0.8229 | 0.3851 | 0.4289 | 0.6430 | 0.8470 | 0.6376 | 0.6352 |
| | | 5D | 0.5151 | 0.5492 | 0.6765 | 0.0829 | 0.3765 | 0.4472 | 0.7608 | 0.5663 | 0.4968 |
| | | 7D | 0.4576 | 0.5168 | 0.6141 | 0.0549 | 0.3011 | 0.4146 | 0.8487 | 0.5028 | 0.4638 |
| | | Avg. | 0.5618 | 0.6711 | 0.7423 | 0.2341 | 0.4138 | 0.5699 | 0.8443 | 0.6161 | 0.5817 |
| **LLM** | GPT4TS | 1D | 0.7588 | 0.8697 | 0.8790 | 0.8738 | 0.5594 | 0.7173 | 0.9066 | 0.6863 | 0.7814 |
| | | 3D | 0.5816 | 0.7515 | 0.8012 | 0.7983 | 0.4484 | 0.6322 | 0.8277 | 0.5967 | 0.6797 |
| | | 5D | 0.4917 | 0.6840 | 0.7480 | 0.7384 | 0.3970 | 0.5830 | 0.7591 | 0.5241 | 0.6157 |
| | | 7D | 0.4388 | 0.6309 | 0.6941 | 0.6878 | 0.3582 | 0.5430 | 0.7099 | 0.4672 | 0.5662 |
| | | Avg. | 0.5677 | 0.7340 | 0.7805 | 0.7746 | 0.4408 | 0.6189 | 0.8008 | 0.5686 | 0.6607 |
| | AutoTimes | 1D | 0.7957 | 0.8638 | 0.8896 | 0.9167 | 0.5985 | 0.7343 | 0.9218 | 0.7225 | 0.8054 |
| | | 3D | 0.6299 | 0.7622 | 0.8190 | 0.8520 | 0.4892 | 0.6712 | 0.8516 | 0.6389 | 0.7142 |
| | | 5D | 0.5392 | 0.6780 | 0.7622 | 0.7817 | 0.4135 | 0.6243 | 0.7982 | 0.5804 | 0.6472 |
| | | 7D | 0.4895 | 0.6363 | 0.7204 | 0.7309 | 0.3710 | 0.5879 | 0.7553 | 0.5180 | 0.6012 |
| | | Avg. | 0.6136 | 0.7351 | 0.7978 | 0.8203 | 0.4680 | 0.6544 | 0.8317 | 0.6149 | 0.6920 |

Table 14: Benchmark(SEDI(5%)) on part 1(Orlando area) stations(MLP,CNN,Transformer)

| | Method | T | S0 | S1 | S2 | S3 | S4 | S5 | S6 | S7 | Avg. |
|---|---|---|---|---|---|---|---|---|---|---|---|
| **MLP** | MLP | 1D | 0.5128 | 0.9250 | 0.7735 | 0.8437 | 0.4047 | 0.7105 | 0.8731 | 0.6329 | 0.7095 |
| | | 3D | 0.5032 | 0.8575 | 0.7227 | 0.9190 | 0.3281 | 0.6590 | 0.8035 | 0.5460 | 0.6674 |
| | | 5D | 0.4580 | 0.7990 | 0.6804 | 0.8465 | 0.2779 | 0.6314 | 0.7573 | 0.5308 | 0.6227 |
| | | 7D | 0.4174 | 0.8105 | 0.6355 | 0.7808 | 0.2684 | 0.5818 | 0.7227 | 0.4304 | 0.5809 |
| | | Avg. | **0.4729** | **0.8480** | **0.7030** | **0.8475** | 0.3198 | 0.6457 | **0.7892** | 0.5350 | **0.6451** |
| | TSMixer | 1D | 0.5204 | 0.8556 | 0.7619 | 0.8916 | 0.3911 | 0.7094 | 0.7193 | 0.6167 | 0.6833 |
| | | 3D | 0.3386 | 0.7182 | 0.6688 | 0.7902 | 0.3201 | 0.6698 | 0.6612 | 0.4533 | 0.5775 |
| | | 5D | 0.2734 | 0.6303 | 0.6004 | 0.7015 | 0.2664 | 0.5025 | 0.6127 | 0.3537 | 0.4926 |
| | | 7D | 0.2257 | 0.5757 | 0.5444 | 0.6277 | 0.2313 | 0.4536 | 0.5548 | 0.2847 | 0.4372 |
| | | Avg. | 0.3395 | 0.6950 | 0.6439 | 0.7528 | 0.3022 | 0.5838 | 0.6370 | 0.4271 | 0.5477 |
| | NLinear | 1D | 0.5802 | 0.8689 | 0.7743 | 0.9014 | 0.4122 | 0.7091 | 0.8529 | 0.6285 | 0.7159 |
| | | 3D | 0.3752 | 0.7312 | 0.6721 | 0.7904 | 0.3451 | 0.6185 | 0.7482 | 0.4648 | 0.5932 |
| | | 5D | 0.3456 | 0.6392 | 0.6011 | 0.6967 | 0.2872 | 0.5471 | 0.6727 | 0.3603 | 0.5187 |
| | | 7D | 0.2322 | 0.5818 | 0.5408 | 0.6184 | 0.2505 | 0.4979 | 0.6121 | 0.2912 | 0.4531 |
| | | Avg. | 0.3833 | 0.7053 | 0.6471 | 0.7517 | **0.3238** | 0.5932 | 0.7215 | 0.4362 | 0.5702 |
| **CNN** | TCN | 1D | 0.3567 | 0.8742 | 0.2636 | 0.6239 | 0.2001 | 0.6360 | 0.4088 | 0.4724 | 0.4795 |
| | | 3D | 0.4252 | 0.7796 | 0.3238 | 0.6102 | 0.1573 | 0.4839 | 0.4081 | 0.3764 | 0.4456 |
| | | 5D | 0.2808 | 0.7319 | 0.2196 | 0.5901 | 0.1484 | 0.3628 | 0.4501 | 0.3326 | 0.3895 |
| | | 7D | 0.3380 | 0.6245 | 0.2753 | 0.4466 | 0.1223 | 0.3825 | 0.2879 | 0.2875 | 0.3456 |
| | | Avg. | 0.3502 | 0.7526 | 0.2706 | 0.5677 | 0.1570 | 0.4663 | 0.3887 | 0.3672 | 0.4150 |
| | ModernTCN | 1D | 0.5274 | 0.8480 | 0.7496 | 0.8700 | 0.3762 | 0.7081 | 0.8590 | 0.6339 | 0.6965 |
| | | 3D | 0.3525 | 0.7260 | 0.6128 | 0.7416 | 0.3183 | 0.5902 | 0.7544 | 0.4769 | 0.5716 |
| | | 5D | 0.2805 | 0.6333 | 0.5298 | 0.6535 | 0.2708 | 0.5110 | 0.6824 | 0.3742 | 0.4919 |
| | | 7D | 0.2334 | 0.5539 | 0.4603 | 0.5819 | 0.2352 | 0.4597 | 0.6208 | 0.3082 | 0.4317 |
| | | Avg. | 0.3484 | 0.6903 | 0.5881 | 0.7118 | 0.3001 | 0.5673 | 0.7292 | 0.4483 | 0.5479 |
| | TimesNet | 1D | 0.4528 | 0.8341 | 0.7355 | 0.8507 | 0.3561 | 0.5952 | 0.8295 | 0.5725 | 0.6533 |
| | | 3D | 0.3185 | 0.7133 | 0.6279 | 0.7271 | 0.2782 | 0.5261 | 0.7186 | 0.4477 | 0.5447 |
| | | 5D | 0.2639 | 0.6133 | 0.5616 | 0.6561 | 0.2305 | 0.4305 | 0.6461 | 0.3400 | 0.4678 |
| | | 7D | 0.2184 | 0.5629 | 0.4993 | 0.5816 | 0.1851 | 0.3748 | 0.5961 | 0.2652 | 0.4104 |
| | | Avg. | 0.3134 | 0.6809 | 0.6061 | 0.7039 | 0.2625 | 0.4816 | 0.6976 | 0.4064 | 0.5190 |
| **Transformer** | iTransformer | 1D | 0.5441 | 0.8814 | 0.7818 | 0.8940 | 0.3864 | 0.7065 | 0.8510 | 0.5962 | 0.7052 |
| | | 3D | 0.3974 | 0.7637 | 0.6710 | 0.7924 | 0.3104 | 0.5943 | 0.7502 | 0.4656 | 0.5931 |
| | | 5D | 0.3080 | 0.6876 | 0.6000 | 0.7083 | 0.2453 | 0.5257 | 0.6786 | 0.3554 | 0.5136 |
| | | 7D | 0.2514 | 0.6234 | 0.5337 | 0.6420 | 0.2047 | 0.4701 | 0.6208 | 0.2896 | 0.4545 |
| | | Avg. | 0.3752 | 0.7390 | 0.6466 | 0.7592 | 0.2867 | 0.5742 | 0.7252 | 0.4267 | 0.5666 |
| | PatchTST | 1D | 0.5491 | 0.8715 | 0.7655 | 0.8953 | 0.3878 | 0.7004 | 0.8402 | 0.6004 | 0.7013 |
| | | 3D | 0.3800 | 0.7584 | 0.6663 | 0.7948 | 0.3210 | 0.5907 | 0.7519 | 0.4659 | 0.5911 |
| | | 5D | 0.3052 | 0.6788 | 0.5937 | 0.6959 | 0.2631 | 0.5219 | 0.6719 | 0.3734 | 0.5130 |
| | | 7D | 0.2502 | 0.6018 | 0.5339 | 0.6317 | 0.2196 | 0.4734 | 0.6156 | 0.3054 | 0.4539 |
| | | Avg. | 0.3711 | 0.7276 | 0.6399 | 0.7544 | 0.2979 | 0.5716 | 0.7199 | 0.4363 | 0.5648 |

Table 15: Benchmark(SEDI(5%)) on part 1(Orlando area) stations(RNN,GNN,LLM)

| | Method | T | S0 | S1 | S2 | S3 | S4 | S5 | S6 | S7 | Avg. |
|---|---|---|---|---|---|---|---|---|---|---|---|
| **RNN** | LSTM | 1D | 0.4017 | 0.8107 | 0.7419 | 0.8659 | 0.3198 | 0.6701 | 0.8120 | 0.5639 | 0.6483 |
| | | 3D | 0.3042 | 0.7513 | 0.6543 | 0.8644 | 0.2781 | 0.5208 | 0.6626 | 0.3957 | 0.5539 |
| | | 5D | 0.2563 | 0.7351 | 0.5817 | 0.7538 | 0.2252 | 0.4925 | 0.6287 | 0.3346 | 0.5010 |
| | | 7D | 0.2557 | 0.6068 | 0.5389 | 0.7051 | 0.2231 | 0.4831 | 0.5848 | 0.2670 | 0.4581 |
| | | Avg. | 0.3045 | 0.7260 | 0.6292 | 0.7973 | 0.2615 | 0.5416 | 0.6720 | 0.3903 | 0.5403 |
| | DeepAR | 1D | 0.5253 | 0.8201 | 0.7607 | 0.9462 | 0.3488 | 0.6711 | 0.8081 | 0.5904 | 0.6838 |
| | | 3D | 0.3304 | 0.7414 | 0.6407 | 0.7018 | 0.2825 | 0.5496 | 0.6188 | 0.4553 | 0.5401 |
| | | 5D | 0.2763 | 0.6725 | 0.5825 | 0.8152 | 0.2105 | 0.5123 | 0.5172 | 0.3650 | 0.4939 |
| | | 7D | 0.2556 | 0.6896 | 0.5741 | 0.8037 | 0.2025 | 0.4243 | 0.4710 | 0.3234 | 0.4680 |
| | | Avg. | 0.3469 | 0.7309 | 0.6395 | 0.8167 | 0.2611 | 0.5393 | 0.6038 | 0.4335 | 0.5465 |
| | DilatedRNN | 1D | 0.5340 | 0.8917 | 0.7632 | 0.8782 | 0.3685 | 0.7112 | 0.8792 | 0.6419 | 0.7085 |
| | | 3D | 0.3663 | 0.8079 | 0.6105 | 0.7865 | 0.2975 | 0.5017 | 0.6871 | 0.4803 | 0.5672 |
| | | 5D | 0.3059 | 0.6907 | 0.5751 | 0.7343 | 0.2560 | 0.5750 | 0.6966 | 0.4549 | 0.5361 |
| | | 7D | 0.3052 | 0.6703 | 0.5421 | 0.6376 | 0.2480 | 0.5455 | 0.6476 | 0.4271 | 0.5029 |
| | | Avg. | 0.3779 | 0.7652 | 0.6227 | 0.7591 | 0.2925 | 0.5833 | 0.7276 | 0.5010 | 0.5787 |
| **GNN** | GCN | 1D | 0.5137 | 0.7241 | 0.7046 | 0.6410 | 0.3511 | 0.7351 | 0.8413 | 0.5751 | 0.6357 |
| | | 3D | 0.3818 | 0.6910 | 0.6424 | 0.6378 | 0.3101 | 0.7135 | 0.7400 | 0.4640 | 0.5726 |
| | | 5D | 0.3292 | 0.6557 | 0.5711 | 0.5477 | 0.2499 | 0.6674 | 0.7444 | 0.4156 | 0.5226 |
| | | 7D | 0.3021 | 0.6169 | 0.5142 | 0.4740 | 0.2346 | 0.5327 | 0.6955 | 0.4517 | 0.4777 |
| | | Avg. | 0.3817 | 0.6719 | 0.6081 | 0.5751 | 0.2864 | **0.6622** | 0.7553 | 0.4766 | 0.5522 |
| | FourierGNN | 1D | 0.5676 | 0.8926 | 0.7800 | 0.8434 | 0.3851 | 0.7068 | 0.8439 | 0.6727 | 0.7115 |
| | | 3D | 0.4629 | 0.7868 | 0.6564 | 0.7028 | 0.3275 | 0.5714 | 0.7651 | 0.5707 | 0.6054 |
| | | 5D | 0.3950 | 0.7889 | 0.6028 | 0.6333 | 0.2748 | 0.5038 | 0.7352 | 0.5055 | 0.5549 |
| | | 7D | 0.3873 | 0.7530 | 0.6115 | 0.5815 | 0.2572 | 0.5880 | 0.7235 | 0.4659 | 0.5460 |
| | | Avg. | 0.4532 | 0.8053 | 0.6626 | 0.6903 | 0.3112 | 0.5925 | 0.7669 | **0.5537** | 0.6045 |
| | StemGNN | 1D | 0.3886 | 0.8590 | 0.7176 | 0.2562 | 0.3499 | 0.7175 | 0.8579 | 0.6320 | 0.5973 |
| | | 3D | 0.3247 | 0.6710 | 0.6642 | 0.1607 | 0.2578 | 0.5487 | 0.7490 | 0.5132 | 0.4862 |
| | | 5D | 0.2763 | 0.4268 | 0.4857 | 0.0234 | 0.2251 | 0.3075 | 0.6028 | 0.3641 | 0.3390 |
| | | 7D | 0.2189 | 0.3938 | 0.3786 | 0.0456 | 0.1851 | 0.2623 | 0.7196 | 0.3385 | 0.3178 |
| | | Avg. | 0.3021 | 0.5877 | 0.5615 | 0.1215 | 0.2545 | 0.4590 | 0.7324 | 0.4619 | 0.4351 |
| **LLM** | GPT4TS | 1D | 0.4904 | 0.8517 | 0.7383 | 0.8528 | 0.3567 | 0.6286 | 0.8333 | 0.5737 | 0.6657 |
| | | 3D | 0.3418 | 0.7131 | 0.6424 | 0.7412 | 0.2897 | 0.5298 | 0.7352 | 0.4407 | 0.5542 |
| | | 5D | 0.2811 | 0.6474 | 0.5588 | 0.6647 | 0.2451 | 0.4542 | 0.6619 | 0.3371 | 0.4813 |
| | | 7D | 0.2508 | 0.5786 | 0.4827 | 0.6002 | 0.2173 | 0.4100 | 0.5996 | 0.2547 | 0.4243 |
| | | Avg. | 0.3410 | 0.6977 | 0.6055 | 0.7147 | 0.2772 | 0.5056 | 0.7075 | 0.4016 | 0.5314 |
| | AutoTimes | 1D | 0.5235 | 0.8448 | 0.7499 | 0.9034 | 0.3786 | 0.6736 | 0.8310 | 0.5961 | 0.6876 |
| | | 3D | 0.3733 | 0.7387 | 0.6568 | 0.8177 | 0.3245 | 0.5828 | 0.7471 | 0.4660 | 0.5884 |
| | | 5D | 0.3023 | 0.6336 | 0.5923 | 0.7174 | 0.2638 | 0.5145 | 0.6738 | 0.3625 | 0.5075 |
| | | 7D | 0.2343 | 0.5856 | 0.5382 | 0.6401 | 0.2301 | 0.4620 | 0.6092 | 0.3000 | 0.4499 |
| | | Avg. | 0.3584 | 0.7007 | 0.6343 | 0.7696 | 0.2993 | 0.5582 | 0.7153 | 0.4311 | 0.5584 |

Table 16: Benchmark(SEDI(1%)) on part 1(Orlando area) stations(MLP,CNN,Transformer)

| | Method | T | S0 | S1 | S2 | S3 | S4 | S5 | S6 | S7 | Avg. |
|---|---|---|---|---|---|---|---|---|---|---|---|
| **MLP** | MLP | 1D | 0.2398 | 0.8330 | 0.3903 | 0.7439 | 0.2199 | 0.2365 | 0.4548 | 0.1809 | 0.4124 |
| | | 3D | 0.1994 | 0.7910 | 0.2967 | 0.7730 | 0.2051 | 0.1198 | 0.3723 | 0.1528 | 0.3638 |
| | | 5D | 0.1508 | 0.7647 | 0.2900 | 0.7580 | 0.1927 | 0.1346 | 0.3785 | 0.1380 | 0.3509 |
| | | 7D | 0.1877 | 0.7879 | 0.2579 | 0.6627 | 0.1935 | 0.1097 | 0.3831 | 0.1107 | 0.3367 |
| | | Avg. | **0.1944** | **0.7942** | **0.3087** | **0.7344** | **0.2028** | 0.1502 | **0.3972** | 0.1456 | **0.3660** |
| | TSMixer | 1D | 0.2154 | 0.7569 | 0.3670 | 0.7701 | 0.2092 | 0.1228 | 0.2195 | 0.1553 | 0.3520 |
| | | 3D | 0.1462 | 0.5898 | 0.2661 | 0.5455 | 0.1401 | 0.0936 | 0.1884 | 0.0939 | 0.2580 |
| | | 5D | 0.1104 | 0.4818 | 0.1939 | 0.4161 | 0.1069 | 0.0697 | 0.1647 | 0.0636 | 0.2009 |
| | | 7D | 0.1078 | 0.3947 | 0.1464 | 0.3318 | 0.0871 | 0.0548 | 0.1533 | 0.0456 | 0.1652 |
| | | Avg. | 0.1450 | 0.5558 | 0.2433 | 0.5159 | 0.1358 | 0.0852 | 0.1815 | 0.0896 | 0.2440 |
| | NLinear | 1D | 0.2205 | 0.7772 | 0.3813 | 0.7807 | 0.2302 | 0.2270 | 0.3975 | 0.1560 | 0.3963 |
| | | 3D | 0.1508 | 0.6056 | 0.2785 | 0.5478 | 0.1460 | 0.1937 | 0.3198 | 0.0888 | 0.2914 |
| | | 5D | 0.1156 | 0.4949 | 0.2062 | 0.4135 | 0.1131 | 0.1686 | 0.2773 | 0.0607 | 0.2313 |
| | | 7D | 0.1104 | 0.4048 | 0.1576 | 0.3231 | 0.0919 | 0.1536 | 0.2289 | 0.0453 | 0.1895 |
| | | Avg. | 0.1493 | 0.5706 | 0.2559 | 0.5163 | 0.1453 | **0.1857** | 0.3059 | 0.0877 | 0.2771 |
| **CNN** | TCN | 1D | 0.1520 | 0.7471 | 0.1639 | 0.2160 | 0.0801 | 0.0385 | 0.0000 | 0.0404 | 0.1797 |
| | | 3D | 0.1481 | 0.6252 | 0.1179 | 0.3218 | 0.0634 | 0.0303 | 0.0232 | 0.0397 | 0.1712 |
| | | 5D | 0.1935 | 0.5904 | 0.0941 | 0.2732 | 0.0516 | 0.0161 | 0.0496 | 0.0063 | 0.1593 |
| | | 7D | 0.1863 | 0.3808 | 0.1512 | 0.1816 | 0.0468 | 0.0121 | 0.0149 | 0.0185 | 0.1240 |
| | | Avg. | 0.1700 | 0.5859 | 0.1318 | 0.2481 | 0.0605 | 0.0243 | 0.0219 | 0.0262 | 0.1586 |
| | ModernTCN | 1D | 0.1937 | 0.7645 | 0.3466 | 0.7558 | 0.2153 | 0.2077 | 0.3743 | 0.1827 | 0.3801 |
| | | 3D | 0.1202 | 0.6078 | 0.2371 | 0.4858 | 0.1331 | 0.1410 | 0.2984 | 0.1105 | 0.2667 |
| | | 5D | 0.0829 | 0.5308 | 0.1742 | 0.3875 | 0.1018 | 0.1315 | 0.2742 | 0.0778 | 0.2201 |
| | | 7D | 0.0804 | 0.4283 | 0.1325 | 0.3070 | 0.0795 | 0.0623 | 0.2074 | 0.0606 | 0.1698 |
| | | Avg. | 0.1193 | 0.5829 | 0.2226 | 0.4840 | 0.1324 | 0.1356 | 0.2886 | 0.1079 | 0.2592 |
| | TimesNet | 1D | 0.1451 | 0.7329 | 0.3521 | 0.6753 | 0.1455 | 0.1175 | 0.3185 | 0.1265 | 0.3267 |
| | | 3D | 0.0951 | 0.5803 | 0.2451 | 0.4443 | 0.1134 | 0.0911 | 0.2359 | 0.0766 | 0.2352 |
| | | 5D | 0.0651 | 0.4805 | 0.1894 | 0.3933 | 0.0900 | 0.0482 | 0.2077 | 0.0577 | 0.1915 |
| | | 7D | 0.0372 | 0.3977 | 0.1499 | 0.2984 | 0.0631 | 0.0399 | 0.1878 | 0.0400 | 0.1518 |
| | | Avg. | 0.0856 | 0.5479 | 0.2341 | 0.4528 | 0.1030 | 0.0742 | 0.2375 | 0.0752 | 0.2263 |
| **Transformer** | iTransformer | 1D | 0.2066 | 0.7969 | 0.3789 | 0.7930 | 0.2251 | 0.1355 | 0.3441 | 0.1508 | 0.3789 |
| | | 3D | 0.1275 | 0.6825 | 0.2843 | 0.5750 | 0.1318 | 0.0913 | 0.2617 | 0.0926 | 0.2808 |
| | | 5D | 0.0961 | 0.5919 | 0.2149 | 0.4326 | 0.0930 | 0.0727 | 0.2305 | 0.0576 | 0.2237 |
| | | 7D | 0.0714 | 0.4919 | 0.1664 | 0.3504 | 0.0721 | 0.0542 | 0.1959 | 0.0448 | 0.1809 |
| | | Avg. | 0.1254 | 0.6408 | 0.2611 | 0.5378 | 0.1305 | 0.0884 | 0.2581 | 0.0864 | 0.2661 |
| | PatchTST | 1D | 0.2149 | 0.7937 | 0.3601 | 0.8041 | 0.2110 | 0.1291 | 0.3226 | 0.1593 | 0.3744 |
| | | 3D | 0.1327 | 0.6565 | 0.2770 | 0.5851 | 0.1429 | 0.0968 | 0.2652 | 0.0913 | 0.2809 |
| | | 5D | 0.0915 | 0.5578 | 0.2143 | 0.4411 | 0.1047 | 0.0734 | 0.2243 | 0.0673 | 0.2218 |
| | | 7D | 0.0757 | 0.4552 | 0.1661 | 0.3639 | 0.0799 | 0.0570 | 0.1975 | 0.0518 | 0.1809 |
| | | Avg. | 0.1287 | 0.6158 | 0.2544 | 0.5485 | 0.1346 | 0.0891 | 0.2524 | 0.0924 | 0.2645 |

Table 17: Benchmark(SEDI(1%)) on part 1(Orlando area) stations(RNN,GNN,LLM)

|  | Method | T | S0 | S1 | S2 | S3 | S4 | S5 | S6 | S7 | Avg. |
|---|---|---|---|---|---|---|---|---|---|---|---|
| **RNN** | LSTM | 1D | 0.1231 | 0.5677 | 0.3286 | 0.5730 | 0.1160 | 0.1235 | 0.2804 | 0.1423 | 0.2818 |
|  |  | 3D | 0.0722 | 0.3377 | 0.2315 | 0.4083 | 0.1017 | 0.0801 | 0.1284 | 0.0580 | 0.1772 |
|  |  | 5D | 0.0462 | 0.4222 | 0.1899 | 0.1858 | 0.0782 | 0.0618 | 0.2246 | 0.0612 | 0.1587 |
|  |  | 7D | 0.0583 | 0.2870 | 0.0658 | 0.3472 | 0.0747 | 0.0526 | 0.1112 | 0.0411 | 0.1297 |
|  |  | Avg. | 0.0749 | 0.4036 | 0.2040 | 0.3786 | 0.0927 | 0.0795 | 0.1862 | 0.0756 | 0.1869 |
|  | DeepAR | 1D | 0.1138 | 0.6157 | 0.3092 | 0.4319 | 0.1276 | 0.1392 | 0.2927 | 0.1144 | 0.2681 |
|  |  | 3D | 0.0954 | 0.3621 | 0.1573 | 0.2978 | 0.0974 | 0.1047 | 0.1110 | 0.0291 | 0.1568 |
|  |  | 5D | 0.0588 | 0.3228 | 0.1519 | 0.2054 | 0.0821 | 0.0691 | 0.0985 | 0.0716 | 0.1325 |
|  |  | 7D | 0.1016 | 0.2944 | 0.0939 | 0.0666 | 0.0793 | 0.0690 | 0.1249 | 0.0787 | 0.1135 |
|  |  | Avg. | 0.0924 | 0.3987 | 0.1781 | 0.2504 | 0.0966 | 0.0955 | 0.1568 | 0.0734 | 0.1677 |
|  | DilatedRNN | 1D | 0.2111 | 0.7473 | 0.3720 | 0.7648 | 0.1895 | 0.1384 | 0.3324 | 0.2115 | 0.3709 |
|  |  | 3D | 0.1417 | 0.6452 | 0.2337 | 0.5917 | 0.1539 | 0.0764 | 0.1583 | 0.1111 | 0.2640 |
|  |  | 5D | 0.1170 | 0.5410 | 0.1941 | 0.4338 | 0.1278 | 0.0712 | 0.2353 | 0.1309 | 0.2314 |
|  |  | 7D | 0.0923 | 0.4478 | 0.1716 | 0.3167 | 0.1039 | 0.0766 | 0.1639 | 0.0805 | 0.1817 |
|  |  | Avg. | 0.1405 | 0.5953 | 0.2428 | 0.5267 | 0.1438 | 0.0906 | 0.2225 | 0.1335 | 0.2620 |
| **GNN** | GCN | 1D | 0.2025 | 0.5785 | 0.3234 | 0.4610 | 0.2208 | 0.1729 | 0.3612 | 0.1966 | 0.3146 |
|  |  | 3D | 0.1393 | 0.5720 | 0.2439 | 0.4175 | 0.0475 | 0.1439 | 0.3082 | 0.1379 | 0.2513 |
|  |  | 5D | 0.1185 | 0.5251 | 0.1685 | 0.3352 | 0.0296 | 0.1351 | 0.3633 | 0.1096 | 0.2231 |
|  |  | 7D | 0.1255 | 0.4695 | 0.1648 | 0.2308 | 0.0269 | 0.1249 | 0.3558 | 0.1684 | 0.2083 |
|  |  | Avg. | 0.1465 | 0.5363 | 0.2252 | 0.3611 | 0.0812 | 0.1442 | 0.3471 | **0.1531** | 0.2493 |
|  | FourierGNN | 1D | 0.2074 | 0.7982 | 0.3768 | 0.7408 | 0.2155 | 0.1250 | 0.3068 | 0.1698 | 0.3675 |
|  |  | 3D | 0.1567 | 0.7499 | 0.2746 | 0.6374 | 0.1849 | 0.1035 | 0.2813 | 0.1395 | 0.3160 |
|  |  | 5D | 0.1326 | 0.7111 | 0.2177 | 0.5194 | 0.1881 | 0.1060 | 0.2724 | 0.1089 | 0.2820 |
|  |  | 7D | 0.1830 | 0.6629 | 0.2348 | 0.4327 | 0.1766 | 0.1203 | 0.3300 | 0.1289 | 0.2836 |
|  |  | Avg. | 0.1699 | 0.7305 | 0.2760 | 0.5826 | 0.1913 | 0.1137 | 0.2976 | 0.1368 | 0.3123 |
|  | StemGNN | 1D | 0.1717 | 0.4968 | 0.3191 | 0.0206 | 0.1620 | 0.1239 | 0.2858 | 0.2005 | 0.2225 |
|  |  | 3D | 0.1410 | 0.3176 | 0.2135 | 0.0176 | 0.1152 | 0.0530 | 0.1569 | 0.1344 | 0.1437 |
|  |  | 5D | 0.1395 | 0.0319 | 0.1679 | 0.0013 | 0.0691 | 0.0051 | 0.1753 | 0.0661 | 0.0820 |
|  |  | 7D | 0.1056 | 0.0337 | 0.1386 | 0.0003 | 0.0784 | 0.0036 | 0.2534 | 0.0487 | 0.0828 |
|  |  | Avg. | 0.1395 | 0.2200 | 0.2098 | 0.0100 | 0.1062 | 0.0464 | 0.2179 | 0.1124 | 0.1328 |
| **LLM** | GPT4TS | 1D | 0.1732 | 0.7775 | 0.3626 | 0.7129 | 0.1662 | 0.1221 | 0.3125 | 0.1272 | 0.3443 |
|  |  | 3D | 0.1047 | 0.6083 | 0.2679 | 0.4717 | 0.1214 | 0.0922 | 0.2501 | 0.0771 | 0.2492 |
|  |  | 5D | 0.0698 | 0.5106 | 0.1923 | 0.3910 | 0.0886 | 0.0498 | 0.2192 | 0.0558 | 0.1971 |
|  |  | 7D | 0.0579 | 0.4151 | 0.1366 | 0.3045 | 0.0733 | 0.0356 | 0.2041 | 0.0334 | 0.1576 |
|  |  | Avg. | 0.1014 | 0.5779 | 0.2399 | 0.4700 | 0.1124 | 0.0749 | 0.2465 | 0.0734 | 0.2370 |
|  | AutoTimes | 1D | 0.1985 | 0.7427 | 0.3622 | 0.8053 | 0.2213 | 0.1268 | 0.3335 | 0.1362 | 0.3658 |
|  |  | 3D | 0.1179 | 0.6341 | 0.2763 | 0.5863 | 0.1497 | 0.0970 | 0.2677 | 0.0801 | 0.2761 |
|  |  | 5D | 0.0788 | 0.5089 | 0.2057 | 0.4334 | 0.1039 | 0.0723 | 0.2216 | 0.0601 | 0.2106 |
|  |  | 7D | 0.0714 | 0.4392 | 0.1579 | 0.3310 | 0.0873 | 0.0559 | 0.1895 | 0.0444 | 0.1721 |
|  |  | Avg. | 0.1167 | 0.5812 | 0.2505 | 0.5390 | 0.1406 | 0.0880 | 0.2531 | 0.0802 | 0.2562 |

Table 18: Benchmark(MAE) on part 2(Miami area) stations(MLP,CNN,Transformer)

| | Method | T | S0 | S1 | S2 | S3 | S4 | S5 | S6 | S7 | Avg. |
|---|---|---|---|---|---|---|---|---|---|---|---|
| **MLP** | MLP | 1D | 0.0739 | 0.0969 | 0.1538 | 0.1234 | 0.1172 | 0.1069 | 0.1168 | 0.1548 | 0.1180 |
| | | 3D | 0.1588 | 0.1621 | 0.2502 | 0.2129 | 0.1952 | 0.1638 | 0.1855 | 0.2459 | 0.1968 |
| | | 5D | 0.2147 | 0.2079 | 0.3080 | 0.2637 | 0.2568 | 0.1984 | 0.2268 | 0.3070 | 0.2479 |
| | | 7D | 0.2650 | 0.2436 | 0.3517 | 0.3046 | 0.2989 | 0.2236 | 0.2595 | 0.3454 | 0.2865 |
| | | Avg. | 0.1781 | 0.1776 | 0.2659 | 0.2262 | 0.2170 | 0.1732 | 0.1971 | 0.2633 | 0.2123 |
| | TSMixer | 1D | 0.0932 | 0.1147 | 0.1777 | 0.1381 | 0.1322 | 0.1152 | 0.1279 | 0.2124 | 0.1389 |
| | | 3D | 0.1637 | 0.1770 | 0.2667 | 0.2123 | 0.2056 | 0.1680 | 0.1926 | 0.2856 | 0.2089 |
| | | 5D | 0.2179 | 0.2209 | 0.3239 | 0.2646 | 0.2615 | 0.2014 | 0.2349 | 0.3376 | 0.2578 |
| | | 7D | 0.2621 | 0.2558 | 0.3663 | 0.3055 | 0.3081 | 0.2258 | 0.2663 | 0.3779 | 0.2960 |
| | | Avg. | 0.1842 | 0.1921 | 0.2836 | 0.2301 | 0.2269 | 0.1776 | 0.2054 | 0.3034 | 0.2254 |
| | NLinear | 1D | 0.0857 | 0.1055 | 0.1672 | 0.1274 | 0.1207 | 0.1085 | 0.1191 | 0.1484 | 0.1228 |
| | | 3D | 0.1603 | 0.1709 | 0.2606 | 0.2059 | 0.1988 | 0.1642 | 0.1882 | 0.2414 | 0.1988 |
| | | 5D | 0.2153 | 0.2167 | 0.3190 | 0.2608 | 0.2560 | 0.1984 | 0.2311 | 0.3010 | 0.2498 |
| | | 7D | 0.2603 | 0.2526 | 0.3618 | 0.3022 | 0.3031 | 0.2237 | 0.2635 | 0.3457 | 0.2891 |
| | | Avg. | 0.1804 | 0.1864 | 0.2772 | 0.2241 | 0.2196 | 0.1737 | 0.2005 | 0.2591 | 0.2151 |
| **CNN** | TCN | 1D | 0.1747 | 0.1498 | 0.1937 | 0.1790 | 0.1882 | 0.1957 | 0.1439 | 0.7995 | 0.2531 |
| | | 3D | 0.2303 | 0.1938 | 0.2971 | 0.2766 | 0.2811 | 0.2347 | 0.2049 | 0.9442 | 0.3328 |
| | | 5D | 0.3770 | 0.2350 | 0.3711 | 0.3214 | 0.3427 | 0.2733 | 0.2633 | 0.8945 | 0.3848 |
| | | 7D | 0.3407 | 0.2960 | 0.3925 | 0.3659 | 0.3887 | 0.3089 | 0.2950 | 1.0090 | 0.4246 |
| | | Avg. | 0.2807 | 0.2186 | 0.3136 | 0.2857 | 0.3002 | 0.2531 | 0.2268 | 0.9118 | 0.3488 |
| | ModernTCN | 1D | 0.0777 | 0.1049 | 0.1764 | 0.1385 | 0.1134 | 0.1034 | 0.1154 | 0.1565 | 0.1233 |
| | | 3D | 0.1611 | 0.1813 | 0.2938 | 0.2362 | 0.1982 | 0.1669 | 0.1906 | 0.2810 | 0.2136 |
| | | 5D | 0.2289 | 0.2449 | 0.3621 | 0.3098 | 0.2576 | 0.2038 | 0.2351 | 0.3383 | 0.2726 |
| | | 7D | 0.2809 | 0.2903 | 0.4201 | 0.3565 | 0.3054 | 0.2314 | 0.2702 | 0.3758 | 0.3163 |
| | | Avg. | 0.1871 | 0.2054 | 0.3131 | 0.2603 | 0.2186 | 0.1764 | 0.2028 | 0.2879 | 0.2315 |
| | TimesNet | 1D | 0.0962 | 0.1269 | 0.1984 | 0.1537 | 0.1505 | 0.1309 | 0.1520 | 0.1927 | 0.1502 |
| | | 3D | 0.1684 | 0.1943 | 0.2871 | 0.2323 | 0.2249 | 0.1854 | 0.2122 | 0.2767 | 0.2227 |
| | | 5D | 0.2212 | 0.2409 | 0.3516 | 0.2849 | 0.2861 | 0.2200 | 0.2587 | 0.3347 | 0.2748 |
| | | 7D | 0.2722 | 0.2761 | 0.3937 | 0.3329 | 0.3326 | 0.2409 | 0.2915 | 0.3775 | 0.3147 |
| | | Avg. | 0.1895 | 0.2095 | 0.3077 | 0.2509 | 0.2485 | 0.1943 | 0.2286 | 0.2954 | 0.2406 |
| **Transformer** | iTransformer | 1D | 0.0731 | 0.0925 | 0.1477 | 0.1178 | 0.1129 | 0.1024 | 0.1104 | 0.1409 | 0.1122 |
| | | 3D | 0.1446 | 0.1603 | 0.2490 | 0.1975 | 0.1901 | 0.1593 | 0.1807 | 0.2334 | 0.1894 |
| | | 5D | 0.2057 | 0.2052 | 0.3095 | 0.2518 | 0.2498 | 0.1938 | 0.2248 | 0.3007 | 0.2427 |
| | | 7D | 0.2513 | 0.2460 | 0.3529 | 0.2933 | 0.2968 | 0.2193 | 0.2555 | 0.3480 | 0.2829 |
| | | Avg. | 0.1687 | **0.1760** | 0.2648 | **0.2151** | **0.2124** | 0.1687 | **0.1928** | 0.2558 | 0.2068 |
| | PatchTST | 1D | 0.0691 | 0.0926 | 0.1472 | 0.1158 | 0.1114 | 0.1012 | 0.1104 | 0.1394 | 0.1109 |
| | | 3D | 0.1419 | 0.1610 | 0.2449 | 0.1975 | 0.1927 | 0.1590 | 0.1828 | 0.2376 | 0.1897 |
| | | 5D | 0.2002 | 0.2094 | 0.3042 | 0.2524 | 0.2504 | 0.1943 | 0.2251 | 0.2999 | 0.2420 |
| | | 7D | 0.2463 | 0.2461 | 0.3499 | 0.2947 | 0.3005 | 0.2200 | 0.2574 | 0.3444 | 0.2824 |
| | | Avg. | **0.1644** | 0.1773 | **0.2615** | **0.2151** | 0.2138 | **0.1686** | 0.1939 | **0.2553** | **0.2062** |

Table 19: Benchmark(MAE) on part 2(Miami area) stations(RNN,GNN,LLM)

| | Method | T | S0 | S1 | S2 | S3 | S4 | S5 | S6 | S7 | Avg. |
|---|---|---|---|---|---|---|---|---|---|---|---|
| **RNN** | LSTM | 1D | 0.1150 | 0.1373 | 0.2165 | 0.1678 | 0.1691 | 0.1378 | 0.1531 | 0.2537 | 0.1688 |
| | | 3D | 0.2287 | 0.2081 | 0.3074 | 0.2574 | 0.2650 | 0.2002 | 0.2272 | 0.3706 | 0.2581 |
| | | 5D | 0.2646 | 0.2556 | 0.3832 | 0.3241 | 0.3190 | 0.2378 | 0.2694 | 0.4339 | 0.3109 |
| | | 7D | 0.3326 | 0.2989 | 0.4425 | 0.3728 | 0.3855 | 0.2579 | 0.2909 | 0.4743 | 0.3569 |
| | | Avg. | 0.2352 | 0.2250 | 0.3374 | 0.2805 | 0.2847 | 0.2084 | 0.2352 | 0.3831 | 0.2737 |
| | DeepAR | 1D | 0.1196 | 0.1356 | 0.2067 | 0.1678 | 0.1659 | 0.1410 | 0.1546 | 0.2557 | 0.1684 |
| | | 3D | 0.2086 | 0.2169 | 0.3113 | 0.2615 | 0.2693 | 0.2047 | 0.2301 | 0.3624 | 0.2581 |
| | | 5D | 0.2839 | 0.2586 | 0.3710 | 0.3123 | 0.3166 | 0.2299 | 0.2790 | 0.4224 | 0.3092 |
| | | 7D | 0.3404 | 0.2992 | 0.4478 | 0.3695 | 0.3657 | 0.2598 | 0.3007 | 0.4670 | 0.3562 |
| | | Avg. | 0.2381 | 0.2276 | 0.3342 | 0.2778 | 0.2794 | 0.2088 | 0.2411 | 0.3769 | 0.2730 |
| | DilatedRNN | 1D | 0.0781 | 0.1134 | 0.1706 | 0.1359 | 0.1309 | 0.1155 | 0.1217 | 0.2111 | 0.1347 |
| | | 3D | 0.1724 | 0.1856 | 0.2843 | 0.2268 | 0.2393 | 0.1769 | 0.2149 | 0.3164 | 0.2271 |
| | | 5D | 0.2463 | 0.2351 | 0.3483 | 0.2967 | 0.3012 | 0.2133 | 0.2548 | 0.3866 | 0.2853 |
| | | 7D | 0.3312 | 0.2732 | 0.4052 | 0.3472 | 0.3469 | 0.2460 | 0.2917 | 0.4338 | 0.3344 |
| | | Avg. | 0.2070 | 0.2018 | 0.3021 | 0.2516 | 0.2546 | 0.1879 | 0.2208 | 0.3370 | 0.2454 |
| **GNN** | GCN | 1D | 0.0910 | 0.1254 | 0.1880 | 0.1761 | 0.1626 | 0.1374 | 0.1528 | 0.9648 | 0.2498 |
| | | 3D | 0.1718 | 0.1904 | 0.2843 | 0.2485 | 0.2410 | 0.1928 | 0.2173 | 0.5350 | 0.2601 |
| | | 5D | 0.2460 | 0.2340 | 0.3390 | 0.3061 | 0.3085 | 0.2319 | 0.2571 | 0.7279 | 0.3313 |
| | | 7D | 0.2998 | 0.2700 | 0.3911 | 0.3452 | 0.3508 | 0.2609 | 0.2914 | 0.6410 | 0.3563 |
| | | Avg. | 0.2022 | 0.2050 | 0.3006 | 0.2690 | 0.2657 | 0.2057 | 0.2297 | 0.7172 | 0.2994 |
| | FourierGNN | 1D | 0.0825 | 0.1072 | 0.1604 | 0.1365 | 0.1226 | 0.1165 | 0.1234 | 0.2444 | 0.1367 |
| | | 3D | 0.1593 | 0.1722 | 0.2527 | 0.2059 | 0.2024 | 0.1734 | 0.1925 | 0.3399 | 0.2123 |
| | | 5D | 0.2109 | 0.2189 | 0.3114 | 0.2658 | 0.2568 | 0.2123 | 0.2320 | 0.4031 | 0.2639 |
| | | 7D | 0.2896 | 0.2506 | 0.3590 | 0.3121 | 0.3067 | 0.2357 | 0.2624 | 0.4222 | 0.3048 |
| | | Avg. | 0.1856 | 0.1872 | 0.2709 | 0.2301 | 0.2221 | 0.1845 | 0.2026 | 0.3524 | 0.2294 |
| | StemGNN | 1D | 0.0859 | 0.1404 | 0.1762 | 0.1382 | 0.1444 | 0.1204 | 0.1252 | 0.3656 | 0.1620 |
| | | 3D | 0.2047 | 0.2609 | 0.2958 | 0.2616 | 0.2763 | 0.2432 | 0.2227 | 0.4821 | 0.2809 |
| | | 5D | 0.2874 | 0.3173 | 0.3743 | 0.3535 | 0.4289 | 0.2839 | 0.2765 | 0.5247 | 0.3558 |
| | | 7D | 0.3603 | 0.3402 | 0.4218 | 0.4172 | 0.4253 | 0.3211 | 0.3283 | 0.6507 | 0.4081 |
| | | Avg. | 0.2346 | 0.2647 | 0.3170 | 0.2926 | 0.3187 | 0.2422 | 0.2382 | 0.5058 | 0.3017 |
| **LLM** | GPT4TS | 1D | 0.0931 | 0.1264 | 0.1860 | 0.1431 | 0.1392 | 0.1304 | 0.1510 | 0.1974 | 0.1458 |
| | | 3D | 0.1639 | 0.1959 | 0.2833 | 0.2283 | 0.2281 | 0.1895 | 0.2159 | 0.2917 | 0.2246 |
| | | 5D | 0.2256 | 0.2429 | 0.3469 | 0.2834 | 0.2897 | 0.2195 | 0.2535 | 0.3389 | 0.2751 |
| | | 7D | 0.2728 | 0.2842 | 0.3941 | 0.3275 | 0.3342 | 0.2466 | 0.2878 | 0.3960 | 0.3179 |
| | | Avg. | 0.1888 | 0.2124 | 0.3026 | 0.2456 | 0.2478 | 0.1965 | 0.2271 | 0.3060 | 0.2408 |
| | AutoTimes | 1D | 0.0863 | 0.1029 | 0.1665 | 0.1318 | 0.1270 | 0.1140 | 0.1312 | 0.1587 | 0.1273 |
| | | 3D | 0.1530 | 0.1663 | 0.2545 | 0.2069 | 0.2008 | 0.1668 | 0.1926 | 0.2450 | 0.1982 |
| | | 5D | 0.2065 | 0.2109 | 0.3079 | 0.2619 | 0.2556 | 0.2017 | 0.2342 | 0.3065 | 0.2481 |
| | | 7D | 0.2552 | 0.2469 | 0.3515 | 0.3009 | 0.3029 | 0.2267 | 0.2654 | 0.3482 | 0.2872 |
| | | Avg. | 0.1752 | 0.1817 | 0.2701 | 0.2254 | 0.2216 | 0.1773 | 0.2058 | 0.2646 | 0.2152 |

Table 20: Benchmark(MSE) on part 2(Miami area) stations(MLP,CNN,Transformer)

| | Method | T | S0 | S1 | S2 | S3 | S4 | S5 | S6 | S7 | Avg. |
|---|---|---|---|---|---|---|---|---|---|---|---|
| **MLP** | MLP | 1D | 0.0315 | 0.0629 | 0.1405 | 0.0774 | 0.0602 | 0.0525 | 0.0787 | 0.1295 | 0.0792 |
| | | 3D | 0.0808 | 0.1311 | 0.2777 | 0.1506 | 0.1289 | 0.0935 | 0.1471 | 0.2762 | 0.1607 |
| | | 5D | 0.1300 | 0.1819 | 0.3613 | 0.2040 | 0.1912 | 0.1197 | 0.1905 | 0.3950 | 0.2217 |
| | | 7D | 0.1783 | 0.2232 | 0.4256 | 0.2480 | 0.2456 | 0.1399 | 0.2230 | 0.4617 | 0.2682 |
| | | Avg. | **0.1052** | **0.1498** | **0.3012** | **0.1700** | **0.1565** | **0.1014** | **0.1599** | **0.3156** | **0.1824** |
| | TSMixer | 1D | 0.0397 | 0.0747 | 0.1695 | 0.0877 | 0.0697 | 0.0578 | 0.0868 | 0.2450 | 0.1039 |
| | | 3D | 0.0932 | 0.1486 | 0.3241 | 0.1611 | 0.1419 | 0.0985 | 0.1605 | 0.3833 | 0.1889 |
| | | 5D | 0.1460 | 0.2061 | 0.4276 | 0.2192 | 0.2111 | 0.1263 | 0.2114 | 0.4777 | 0.2532 |
| | | 7D | 0.1968 | 0.2537 | 0.5054 | 0.2679 | 0.2786 | 0.1475 | 0.2509 | 0.5544 | 0.3069 |
| | | Avg. | 0.1189 | 0.1708 | 0.3566 | 0.1840 | 0.1753 | 0.1075 | 0.1774 | 0.4151 | 0.2132 |
| | NLinear | 1D | 0.0374 | 0.0687 | 0.1605 | 0.0830 | 0.0639 | 0.0554 | 0.0816 | 0.1269 | 0.0847 |
| | | 3D | 0.0924 | 0.1442 | 0.3186 | 0.1581 | 0.1376 | 0.0974 | 0.1568 | 0.2825 | 0.1734 |
| | | 5D | 0.1450 | 0.2026 | 0.4224 | 0.2174 | 0.2065 | 0.1253 | 0.2083 | 0.3881 | 0.2395 |
| | | 7D | 0.1962 | 0.2511 | 0.5021 | 0.2666 | 0.2743 | 0.1471 | 0.2484 | 0.4696 | 0.2944 |
| | | Avg. | 0.1177 | 0.1666 | 0.3509 | 0.1813 | 0.1706 | 0.1063 | 0.1738 | 0.3168 | 0.1980 |
| **CNN** | TCN | 1D | 0.0976 | 0.0963 | 0.1756 | 0.1090 | 0.1377 | 0.1694 | 0.0984 | 5.8002 | 0.8355 |
| | | 3D | 0.1318 | 0.1480 | 0.3271 | 0.2042 | 0.2212 | 0.2113 | 0.1527 | 7.0577 | 1.0567 |
| | | 5D | 0.3490 | 0.2009 | 0.4521 | 0.2532 | 0.3033 | 0.2986 | 0.2079 | 5.1319 | 0.8996 |
| | | 7D | 0.2479 | 0.2662 | 0.4783 | 0.2929 | 0.3587 | 0.2911 | 0.2391 | 6.7494 | 1.1155 |
| | | Avg. | 0.2066 | 0.1779 | 0.3583 | 0.2148 | 0.2552 | 0.2426 | 0.1745 | 6.1848 | 0.9768 |
| | ModernTCN | 1D | 0.0395 | 0.0838 | 0.2180 | 0.1137 | 0.0630 | 0.0539 | 0.0818 | 0.3165 | 0.1213 |
| | | 3D | 0.1168 | 0.1894 | 0.4307 | 0.2323 | 0.1464 | 0.1036 | 0.1678 | 2.9814 | 0.5460 |
| | | 5D | 0.1869 | 0.3223 | 0.5611 | 0.3355 | 0.2198 | 0.1352 | 0.2221 | 2.7558 | 0.5923 |
| | | 7D | 0.2499 | 0.4475 | 0.7525 | 0.3950 | 0.2877 | 0.1613 | 0.2656 | 1.4768 | 0.5045 |
| | | Avg. | 0.1483 | 0.2608 | 0.4906 | 0.2691 | 0.1792 | 0.1135 | 0.1843 | 1.8826 | 0.4411 |
| | TimesNet | 1D | 0.0453 | 0.0899 | 0.1964 | 0.1033 | 0.0884 | 0.0700 | 0.1090 | 0.1729 | 0.1094 |
| | | 3D | 0.1100 | 0.1839 | 0.3598 | 0.1995 | 0.1694 | 0.1154 | 0.1857 | 0.3311 | 0.2069 |
| | | 5D | 0.1666 | 0.2634 | 0.4952 | 0.2586 | 0.2486 | 0.1484 | 0.2477 | 0.4500 | 0.2848 |
| | | 7D | 0.2283 | 0.3207 | 0.5949 | 0.3303 | 0.3296 | 0.1674 | 0.2892 | 0.5382 | 0.3498 |
| | | Avg. | 0.1376 | 0.2145 | 0.4116 | 0.2230 | 0.2090 | 0.1253 | 0.2079 | 0.3731 | 0.2377 |
| **Transformer** | iTransformer | 1D | 0.0337 | 0.0645 | 0.1413 | 0.0798 | 0.0614 | 0.0536 | 0.0766 | 0.1210 | 0.0790 |
| | | 3D | 0.0880 | 0.1424 | 0.3086 | 0.1614 | 0.1361 | 0.0964 | 0.1512 | 0.2755 | 0.1700 |
| | | 5D | 0.1486 | 0.2002 | 0.4253 | 0.2206 | 0.2052 | 0.1266 | 0.2082 | 0.3973 | 0.2415 |
| | | 7D | 0.2032 | 0.2658 | 0.4903 | 0.2697 | 0.2727 | 0.1490 | 0.2408 | 0.5022 | 0.2992 |
| | | Avg. | 0.1184 | 0.1682 | 0.3414 | 0.1828 | 0.1688 | 0.1064 | 0.1692 | 0.3240 | 0.1974 |
| | PatchTST | 1D | 0.0326 | 0.0667 | 0.1392 | 0.0789 | 0.0615 | 0.0525 | 0.0781 | 0.1213 | 0.0789 |
| | | 3D | 0.0872 | 0.1459 | 0.2950 | 0.1614 | 0.1389 | 0.0963 | 0.1586 | 0.2825 | 0.1707 |
| | | 5D | 0.1402 | 0.2096 | 0.3952 | 0.2208 | 0.2068 | 0.1249 | 0.2112 | 0.3960 | 0.2381 |
| | | 7D | 0.1899 | 0.2624 | 0.4758 | 0.2686 | 0.2763 | 0.1473 | 0.2494 | 0.4766 | 0.2933 |
| | | Avg. | 0.1125 | 0.1711 | 0.3263 | 0.1824 | 0.1709 | 0.1053 | 0.1743 | 0.3191 | 0.1952 |

Table 21: Benchmark(MSE) on part 2(Miami area) stations(RNN,GNN,LLM)

| | Method | Metric | S0 | S1 | S2 | S3 | S4 | S5 | S6 | S7 | Avg. |
|---|---|---|---|---|---|---|---|---|---|---|---|
| **RNN** | LSTM | 1D | 0.0575 | 0.1123 | 0.2405 | 0.1158 | 0.1092 | 0.0775 | 0.1224 | 1.1111 | 0.2433 |
| | | 3D | 0.1531 | 0.1869 | 0.3854 | 0.2070 | 0.2091 | 0.1206 | 0.1944 | 1.4368 | 0.3617 |
| | | 5D | 0.1983 | 0.2390 | 0.4905 | 0.2732 | 0.2894 | 0.1479 | 0.2368 | 1.3811 | 0.4070 |
| | | 7D | 0.2734 | 0.2901 | 0.5826 | 0.3426 | 0.3981 | 0.1677 | 0.2615 | 1.8380 | 0.5192 |
| | | Avg. | 0.1706 | 0.2071 | 0.4248 | 0.2346 | 0.2515 | 0.1284 | 0.2038 | 1.4417 | 0.3828 |
| | DeepAR | 1D | 0.0611 | 0.1108 | 0.2390 | 0.1248 | 0.1152 | 0.0790 | 0.1230 | 0.9006 | 0.2192 |
| | | 3D | 0.1276 | 0.1894 | 0.3864 | 0.2041 | 0.2173 | 0.1226 | 0.1953 | 1.3494 | 0.3490 |
| | | 5D | 0.1973 | 0.2385 | 0.4877 | 0.2623 | 0.2918 | 0.1463 | 0.2344 | 1.4248 | 0.4104 |
| | | 7D | 0.2820 | 0.3010 | 0.5995 | 0.3402 | 0.3564 | 0.1667 | 0.2657 | 1.4515 | 0.4704 |
| | | Avg. | 0.1670 | 0.2099 | 0.4282 | 0.2328 | 0.2452 | 0.1286 | 0.2046 | 1.2816 | 0.3622 |
| | DilatedRNN | 1D | 0.0366 | 0.0808 | 0.1810 | 0.0931 | 0.0823 | 0.0635 | 0.0944 | 0.8270 | 0.1823 |
| | | 3D | 0.1137 | 0.1710 | 0.3481 | 0.1894 | 0.2205 | 0.1151 | 0.1746 | 1.0864 | 0.3023 |
| | | 5D | 0.1970 | 0.2275 | 0.4660 | 0.2738 | 0.3055 | 0.1469 | 0.2277 | 1.3925 | 0.4046 |
| | | 7D | 0.2899 | 0.2742 | 0.5486 | 0.3313 | 0.3679 | 0.1781 | 0.2746 | 1.3743 | 0.4549 |
| | | Avg. | 0.1593 | 0.1884 | 0.3859 | 0.2219 | 0.2441 | 0.1259 | 0.1928 | 1.1701 | 0.3360 |
| **GNN** | GCN | 1D | 0.0333 | 0.0964 | 0.1750 | 0.1817 | 0.1209 | 0.0909 | 0.0948 | 170.0996 | 21.3616 |
| | | 3D | 0.0829 | 0.1800 | 0.3166 | 0.2245 | 0.1894 | 0.1255 | 0.1616 | 32.4988 | 4.2224 |
| | | 5D | 0.1407 | 0.2346 | 0.4015 | 0.2996 | 0.2709 | 0.1500 | 0.2053 | 44.7123 | 5.8019 |
| | | 7D | 0.2021 | 0.2599 | 0.4775 | 0.3266 | 0.3373 | 0.1828 | 0.2379 | 17.1457 | 2.3962 |
| | | Avg. | 0.1147 | 0.1927 | 0.3426 | 0.2581 | 0.2296 | 0.1373 | 0.1749 | 66.1141 | 8.4455 |
| | FourierGNN | 1D | 0.0339 | 0.0703 | 0.1442 | 0.0825 | 0.0631 | 0.0563 | 0.0809 | 0.6083 | 0.1425 |
| | | 3D | 0.0851 | 0.1359 | 0.2816 | 0.1522 | 0.1320 | 0.0988 | 0.1493 | 0.9670 | 0.2502 |
| | | 5D | 0.1331 | 0.1885 | 0.3656 | 0.2117 | 0.1916 | 0.1290 | 0.1920 | 1.6289 | 0.3800 |
| | | 7D | 0.2053 | 0.2240 | 0.4330 | 0.2525 | 0.2511 | 0.1483 | 0.2250 | 1.0897 | 0.3536 |
| | | Avg. | 0.1144 | 0.1547 | 0.3061 | 0.1747 | 0.1594 | 0.1081 | 0.1618 | 1.0735 | 0.2816 |
| | StemGNN | 1D | 0.0391 | 0.1055 | 0.1724 | 0.0912 | 0.0873 | 0.0665 | 0.0898 | 1.2556 | 0.2384 |
| | | 3D | 0.1283 | 0.2207 | 0.3498 | 0.2164 | 0.2680 | 0.1644 | 0.1796 | 1.5368 | 0.3830 |
| | | 5D | 0.2199 | 0.2945 | 0.4663 | 0.3239 | 0.5381 | 0.2094 | 0.2414 | 1.6949 | 0.4985 |
| | | 7D | 0.3149 | 0.3582 | 0.5362 | 0.3986 | 0.4688 | 0.2426 | 0.3031 | 2.0180 | 0.5801 |
| | | Avg. | 0.1755 | 0.2447 | 0.3812 | 0.2575 | 0.3406 | 0.1707 | 0.2035 | 1.6263 | 0.4250 |
| **LLM** | GPT4TS | 1D | 0.0473 | 0.0949 | 0.1828 | 0.0977 | 0.0819 | 0.0735 | 0.1124 | 0.1921 | 0.1103 |
| | | 3D | 0.1163 | 0.1995 | 0.3605 | 0.2009 | 0.1762 | 0.1251 | 0.1953 | 0.3751 | 0.2186 |
| | | 5D | 0.1908 | 0.2827 | 0.4853 | 0.2619 | 0.2654 | 0.1534 | 0.2487 | 0.4691 | 0.2947 |
| | | 7D | 0.2394 | 0.3739 | 0.5955 | 0.3207 | 0.3350 | 0.1826 | 0.2946 | 0.5944 | 0.3670 |
| | | Avg. | 0.1484 | 0.2377 | 0.4060 | 0.2203 | 0.2146 | 0.1337 | 0.2127 | 0.4077 | 0.2477 |
| | AutoTimes | 1D | 0.0375 | 0.0718 | 0.1582 | 0.0879 | 0.0687 | 0.0581 | 0.0900 | 0.1373 | 0.0887 |
| | | 3D | 0.0889 | 0.1438 | 0.3015 | 0.1598 | 0.1410 | 0.0984 | 0.1615 | 0.2986 | 0.1742 |
| | | 5D | 0.1406 | 0.2011 | 0.3952 | 0.2175 | 0.2057 | 0.1265 | 0.2115 | 0.4012 | 0.2374 |
| | | 7D | 0.1944 | 0.2496 | 0.4659 | 0.2653 | 0.2754 | 0.1485 | 0.2510 | 0.4759 | 0.2908 |
| | | Avg. | 0.1154 | 0.1666 | 0.3302 | 0.1826 | 0.1727 | 0.1079 | 0.1785 | 0.3283 | 0.1978 |

Table 22: Benchmark(SEDI10) on part 2(Miami area) stations(MLP,CNN,Transformer)

| | Method | T | S0 | S1 | S2 | S3 | S4 | S5 | S6 | S7 | Avg. |
|---|---|---|---|---|---|---|---|---|---|---|---|
| **MLP** | MLP | 1D | 0.8741 | 0.8301 | 0.8503 | 0.8179 | 0.8890 | 0.5135 | 0.4864 | 0.7414 | 0.7503 |
| | | 3D | 0.8258 | 0.7763 | 0.8076 | 0.7682 | 0.8627 | 0.4220 | 0.4150 | 0.6270 | 0.6881 |
| | | 5D | 0.7627 | 0.7409 | 0.7604 | 0.7023 | 0.8228 | 0.3802 | 0.3853 | 0.6126 | 0.6459 |
| | | 7D | 0.7108 | 0.7069 | 0.7572 | 0.6765 | 0.8028 | 0.3550 | 0.3867 | 0.5522 | 0.6185 |
| | | Avg. | **0.7933** | **0.7636** | **0.7939** | **0.7412** | **0.8443** | **0.4177** | **0.4183** | **0.6333** | **0.6757** |
| | TSMixer | 1D | 0.8506 | 0.7672 | 0.8000 | 0.7572 | 0.8709 | 0.4446 | 0.4225 | 0.5891 | 0.6878 |
| | | 3D | 0.7622 | 0.6598 | 0.7016 | 0.6497 | 0.7979 | 0.3370 | 0.3233 | 0.4826 | 0.5893 |
| | | 5D | 0.6958 | 0.5955 | 0.6414 | 0.5820 | 0.7421 | 0.2694 | 0.2707 | 0.4073 | 0.5255 |
| | | 7D | 0.6433 | 0.5499 | 0.5962 | 0.5360 | 0.6957 | 0.2335 | 0.2379 | 0.3550 | 0.4810 |
| | | Avg. | 0.7380 | 0.6431 | 0.6848 | 0.6312 | 0.7767 | 0.3211 | 0.3136 | 0.4585 | 0.5709 |
| | NLinear | 1D | 0.8486 | 0.7786 | 0.8039 | 0.7660 | 0.8697 | 0.4931 | 0.4452 | 0.7005 | 0.7132 |
| | | 3D | 0.7606 | 0.6705 | 0.7059 | 0.6597 | 0.7969 | 0.3753 | 0.3424 | 0.5532 | 0.6081 |
| | | 5D | 0.6961 | 0.6019 | 0.6452 | 0.5928 | 0.7418 | 0.3265 | 0.2868 | 0.4605 | 0.5439 |
| | | 7D | 0.6445 | 0.5544 | 0.6000 | 0.5465 | 0.6962 | 0.2868 | 0.2522 | 0.3980 | 0.4973 |
| | | Avg. | 0.7374 | 0.6514 | 0.6887 | 0.6412 | 0.7761 | 0.3704 | 0.3316 | 0.5280 | 0.5906 |
| **CNN** | TCN | 1D | 0.7593 | 0.7900 | 0.8204 | 0.6993 | 0.7827 | 0.2702 | 0.4436 | 0.2651 | 0.6038 |
| | | 3D | 0.7316 | 0.6984 | 0.7447 | 0.5739 | 0.7257 | 0.1869 | 0.3902 | 0.1932 | 0.5306 |
| | | 5D | 0.5618 | 0.6040 | 0.6806 | 0.5003 | 0.6732 | 0.1566 | 0.3651 | 0.1776 | 0.4649 |
| | | 7D | 0.6365 | 0.5666 | 0.6778 | 0.4988 | 0.5719 | 0.1185 | 0.2944 | 0.1443 | 0.4386 |
| | | Avg. | 0.6723 | 0.6647 | 0.7309 | 0.5681 | 0.6884 | 0.1831 | 0.3733 | 0.1951 | 0.5095 |
| | ModernTCN | 1D | 0.8450 | 0.7649 | 0.7755 | 0.7226 | 0.8710 | 0.4506 | 0.4383 | 0.6989 | 0.6959 |
| | | 3D | 0.7464 | 0.6562 | 0.6524 | 0.5939 | 0.7859 | 0.3206 | 0.3270 | 0.5369 | 0.5774 |
| | | 5D | 0.6737 | 0.5673 | 0.5828 | 0.5158 | 0.7293 | 0.2661 | 0.2691 | 0.4358 | 0.5050 |
| | | 7D | 0.6180 | 0.5240 | 0.5309 | 0.4738 | 0.6825 | 0.2327 | 0.2278 | 0.3797 | 0.4587 |
| | | Avg. | 0.7208 | 0.6281 | 0.6354 | 0.5766 | 0.7672 | 0.3175 | 0.3156 | 0.5128 | 0.5592 |
| | TimesNet | 1D | 0.8231 | 0.7314 | 0.7478 | 0.7043 | 0.8409 | 0.3910 | 0.3584 | 0.6258 | 0.6528 |
| | | 3D | 0.7335 | 0.6302 | 0.6612 | 0.6004 | 0.7609 | 0.2651 | 0.2711 | 0.4914 | 0.5517 |
| | | 5D | 0.6745 | 0.5599 | 0.5876 | 0.5382 | 0.7072 | 0.2241 | 0.2204 | 0.4053 | 0.4896 |
| | | 7D | 0.6158 | 0.5245 | 0.5430 | 0.4907 | 0.6482 | 0.1888 | 0.1921 | 0.3477 | 0.4438 |
| | | Avg. | 0.7117 | 0.6115 | 0.6349 | 0.5834 | 0.7393 | 0.2672 | 0.2605 | 0.4675 | 0.5345 |
| **Transformer** | iTransformer | 1D | 0.8637 | 0.8038 | 0.8207 | 0.7756 | 0.8762 | 0.4669 | 0.4545 | 0.7247 | 0.7233 |
| | | 3D | 0.7686 | 0.7074 | 0.7133 | 0.6568 | 0.8054 | 0.3441 | 0.3459 | 0.5810 | 0.6153 |
| | | 5D | 0.6971 | 0.6270 | 0.6584 | 0.5889 | 0.7435 | 0.2774 | 0.2809 | 0.4665 | 0.5424 |
| | | 7D | 0.6451 | 0.5751 | 0.6083 | 0.5437 | 0.6964 | 0.2396 | 0.2535 | 0.3979 | 0.4949 |
| | | Avg. | 0.7436 | 0.6783 | 0.7002 | 0.6412 | 0.7803 | 0.3320 | 0.3337 | 0.5425 | 0.5940 |
| | PatchTST | 1D | 0.8595 | 0.7994 | 0.8269 | 0.7725 | 0.8746 | 0.4772 | 0.4750 | 0.7220 | 0.7259 |
| | | 3D | 0.7678 | 0.6914 | 0.7225 | 0.6534 | 0.7956 | 0.3561 | 0.3503 | 0.5633 | 0.6125 |
| | | 5D | 0.7022 | 0.6194 | 0.6605 | 0.5880 | 0.7365 | 0.3018 | 0.2951 | 0.4659 | 0.5462 |
| | | 7D | 0.6513 | 0.5667 | 0.6103 | 0.5407 | 0.6906 | 0.2540 | 0.2564 | 0.4046 | 0.4968 |
| | | Avg. | 0.7452 | 0.6692 | 0.7051 | 0.6386 | 0.7743 | 0.3473 | 0.3442 | 0.5389 | 0.5954 |

Table 23: Benchmark(SEDI10) on part 2(Miami area) stations(RNN,GNN,LLM)

| | Method | T | S0 | S1 | S2 | S3 | S4 | S5 | S6 | S7 | Avg. |
|---|---|---|---|---|---|---|---|---|---|---|---|
| **RNN** | LSTM | 1D | 0.8360 | 0.7782 | 0.8055 | 0.7590 | 0.8905 | 0.4015 | 0.3896 | 0.6608 | 0.6901 |
| | | 3D | 0.7519 | 0.6710 | 0.6999 | 0.6089 | 0.8233 | 0.2542 | 0.2837 | 0.5149 | 0.5760 |
| | | 5D | 0.6692 | 0.6430 | 0.6611 | 0.5801 | 0.7650 | 0.2298 | 0.2401 | 0.3879 | 0.5220 |
| | | 7D | 0.6402 | 0.5991 | 0.6269 | 0.4682 | 0.7060 | 0.1930 | 0.2364 | 0.3669 | 0.4796 |
| | | Avg. | 0.7243 | 0.6728 | 0.6984 | 0.6041 | 0.7962 | 0.2696 | 0.2874 | 0.4826 | 0.5669 |
| | DeepAR | 1D | 0.8152 | 0.7886 | 0.7825 | 0.7384 | 0.8582 | 0.3910 | 0.4268 | 0.6741 | 0.6843 |
| | | 3D | 0.7450 | 0.6511 | 0.7208 | 0.6443 | 0.8068 | 0.2397 | 0.2744 | 0.5008 | 0.5729 |
| | | 5D | 0.6943 | 0.6334 | 0.6518 | 0.5570 | 0.7574 | 0.2019 | 0.2405 | 0.4519 | 0.5235 |
| | | 7D | 0.6155 | 0.5923 | 0.6195 | 0.5072 | 0.7319 | 0.2241 | 0.2245 | 0.3572 | 0.4840 |
| | | Avg. | 0.7175 | 0.6664 | 0.6936 | 0.6117 | 0.7886 | 0.2642 | 0.2916 | 0.4960 | 0.5662 |
| | DilatedRNN | 1D | 0.8692 | 0.7880 | 0.8197 | 0.7892 | 0.8793 | 0.4684 | 0.4362 | 0.6942 | 0.7180 |
| | | 3D | 0.7796 | 0.7056 | 0.7306 | 0.6319 | 0.7575 | 0.2916 | 0.3150 | 0.5490 | 0.5951 |
| | | 5D | 0.6999 | 0.6585 | 0.6645 | 0.5788 | 0.7220 | 0.2488 | 0.2843 | 0.4543 | 0.5389 |
| | | 7D | 0.6469 | 0.5807 | 0.5970 | 0.5429 | 0.6725 | 0.2202 | 0.2308 | 0.3934 | 0.4856 |
| | | Avg. | 0.7489 | 0.6832 | 0.7029 | 0.6357 | 0.7578 | 0.3072 | 0.3166 | 0.5227 | 0.5844 |
| **GNN** | GCN | 1D | 0.8394 | 0.7752 | 0.8086 | 0.7437 | 0.8631 | 0.4585 | 0.4685 | 0.6737 | 0.7038 |
| | | 3D | 0.7913 | 0.7114 | 0.7760 | 0.6987 | 0.8134 | 0.4180 | 0.3846 | 0.5844 | 0.6472 |
| | | 5D | 0.7726 | 0.6603 | 0.7433 | 0.6416 | 0.8037 | 0.3545 | 0.3369 | 0.5404 | 0.6067 |
| | | 7D | 0.7195 | 0.6370 | 0.7368 | 0.6266 | 0.7624 | 0.2961 | 0.3282 | 0.5247 | 0.5789 |
| | | Avg. | 0.7807 | 0.6960 | 0.7662 | 0.6777 | 0.8107 | 0.3818 | 0.3795 | 0.5808 | 0.6342 |
| | FourierGNN | 1D | 0.8842 | 0.8339 | 0.8446 | 0.8034 | 0.8794 | 0.4924 | 0.4661 | 0.5660 | 0.7213 |
| | | 3D | 0.8071 | 0.7862 | 0.7878 | 0.6988 | 0.8383 | 0.3740 | 0.3840 | 0.5319 | 0.6510 |
| | | 5D | 0.7275 | 0.6700 | 0.7674 | 0.6396 | 0.8200 | 0.3434 | 0.3735 | 0.5033 | 0.6056 |
| | | 7D | 0.7208 | 0.6275 | 0.7505 | 0.6695 | 0.7939 | 0.3157 | 0.3493 | 0.4627 | 0.5863 |
| | | Avg. | 0.7849 | 0.7294 | 0.7876 | 0.7028 | 0.8329 | 0.3814 | 0.3932 | 0.5160 | 0.6410 |
| | StemGNN | 1D | 0.8646 | 0.7785 | 0.7954 | 0.7146 | 0.8769 | 0.4439 | 0.4379 | 0.4597 | 0.6714 |
| | | 3D | 0.7744 | 0.6132 | 0.7012 | 0.6006 | 0.7568 | 0.2438 | 0.2951 | 0.3788 | 0.5455 |
| | | 5D | 0.6814 | 0.6045 | 0.6612 | 0.5584 | 0.5665 | 0.2083 | 0.2529 | 0.3115 | 0.4806 |
| | | 7D | 0.6648 | 0.5171 | 0.6013 | 0.4415 | 0.5642 | 0.1738 | 0.2100 | 0.2598 | 0.4291 |
| | | Avg. | 0.7463 | 0.6283 | 0.6898 | 0.5788 | 0.6911 | 0.2674 | 0.2990 | 0.3525 | 0.5316 |
| **LLM** | GPT4TS | 1D | 0.8216 | 0.7340 | 0.7701 | 0.7331 | 0.8414 | 0.3715 | 0.3565 | 0.6124 | 0.6551 |
| | | 3D | 0.7389 | 0.6277 | 0.6603 | 0.6071 | 0.7518 | 0.2678 | 0.2655 | 0.4756 | 0.5493 |
| | | 5D | 0.6746 | 0.5605 | 0.5939 | 0.5327 | 0.6894 | 0.2279 | 0.2255 | 0.3964 | 0.4876 |
| | | 7D | 0.6139 | 0.5161 | 0.5541 | 0.4812 | 0.6419 | 0.1929 | 0.1982 | 0.3275 | 0.4407 |
| | | Avg. | 0.7123 | 0.6096 | 0.6446 | 0.5885 | 0.7311 | 0.2650 | 0.2614 | 0.4530 | 0.5332 |
| | AutoTimes | 1D | 0.8574 | 0.7975 | 0.8157 | 0.7598 | 0.8726 | 0.4113 | 0.4080 | 0.7040 | 0.7033 |
| | | 3D | 0.7652 | 0.6858 | 0.7175 | 0.6466 | 0.8032 | 0.3067 | 0.3215 | 0.5668 | 0.6017 |
| | | 5D | 0.7002 | 0.6266 | 0.6601 | 0.5837 | 0.7408 | 0.2576 | 0.2676 | 0.4668 | 0.5379 |
| | | 7D | 0.6453 | 0.5683 | 0.6149 | 0.5348 | 0.6967 | 0.2213 | 0.2319 | 0.4040 | 0.4896 |
| | | Avg. | 0.7420 | 0.6695 | 0.7021 | 0.6312 | 0.7783 | 0.2992 | 0.3073 | 0.5354 | 0.5831 |

Table 24: Benchmark(SEDI5) on part 2(Miami area) stations(MLP,CNN,Transformer)

|  | Method | T | S0 | S1 | S2 | S3 | S4 | S5 | S6 | S7 | Avg. |
|---|---|---|---|---|---|---|---|---|---|---|---|
| **MLP** | MLP | 1D | 0.8203 | 0.8018 | 0.8453 | 0.6927 | 0.8843 | 0.3386 | 0.3853 | 0.6361 | 0.6755 |
| | | 3D | 0.7427 | 0.7459 | 0.8080 | 0.6251 | 0.8496 | 0.2768 | 0.3320 | 0.5376 | 0.6147 |
| | | 5D | 0.6747 | 0.7152 | 0.7456 | 0.5607 | 0.8084 | 0.2658 | 0.3134 | 0.5075 | 0.5739 |
| | | 7D | 0.6180 | 0.6833 | 0.7520 | 0.5447 | 0.7774 | 0.2555 | 0.3284 | 0.4584 | 0.5522 |
| | | Avg. | **0.7139** | **0.7366** | **0.7878** | **0.6058** | **0.8299** | **0.2842** | **0.3398** | **0.5349** | **0.6041** |
| | TSMixer | 1D | 0.7829 | 0.7231 | 0.7853 | 0.6187 | 0.8603 | 0.2823 | 0.3224 | 0.4521 | 0.6034 |
| | | 3D | 0.6518 | 0.5942 | 0.6704 | 0.5094 | 0.7743 | 0.2085 | 0.2223 | 0.3548 | 0.4982 |
| | | 5D | 0.5559 | 0.5210 | 0.5950 | 0.4429 | 0.7069 | 0.1762 | 0.1753 | 0.2891 | 0.4328 |
| | | 7D | 0.4809 | 0.4667 | 0.5379 | 0.3967 | 0.6516 | 0.1424 | 0.1484 | 0.2437 | 0.3835 |
| | | Avg. | 0.6179 | 0.5763 | 0.6471 | 0.4919 | 0.7483 | 0.2023 | 0.2171 | 0.3349 | 0.4795 |
| | NLinear | 1D | 0.7842 | 0.7341 | 0.7857 | 0.6272 | 0.8583 | 0.3155 | 0.3398 | 0.5837 | 0.6286 |
| | | 3D | 0.6549 | 0.6065 | 0.6750 | 0.5297 | 0.7740 | 0.2379 | 0.2452 | 0.4378 | 0.5201 |
| | | 5D | 0.5606 | 0.5291 | 0.6000 | 0.4519 | 0.7081 | 0.2114 | 0.1966 | 0.3530 | 0.4513 |
| | | 7D | 0.4881 | 0.4734 | 0.5439 | 0.4077 | 0.6537 | 0.1884 | 0.1690 | 0.2964 | 0.4026 |
| | | Avg. | 0.6220 | 0.5858 | 0.6512 | 0.5041 | 0.7485 | 0.2383 | 0.2377 | 0.4177 | 0.5007 |
| **CNN** | TCN | 1D | 0.6732 | 0.7264 | 0.8085 | 0.4636 | 0.6769 | 0.1780 | 0.3211 | 0.1701 | 0.5022 |
| | | 3D | 0.6003 | 0.6072 | 0.7341 | 0.3332 | 0.6076 | 0.1251 | 0.2746 | 0.1106 | 0.4241 |
| | | 5D | 0.3759 | 0.4967 | 0.6449 | 0.3185 | 0.5094 | 0.0967 | 0.2207 | 0.1031 | 0.3457 |
| | | 7D | 0.3713 | 0.4688 | 0.6172 | 0.2808 | 0.4117 | 0.0658 | 0.1851 | 0.0710 | 0.3089 |
| | | Avg. | 0.5052 | 0.5748 | 0.7012 | 0.3490 | 0.5514 | 0.1164 | 0.2504 | 0.1137 | 0.3952 |
| | ModernTCN | 1D | 0.7753 | 0.7152 | 0.7470 | 0.5845 | 0.8521 | 0.2784 | 0.3290 | 0.5653 | 0.6058 |
| | | 3D | 0.6425 | 0.5891 | 0.6141 | 0.4597 | 0.7544 | 0.1988 | 0.2123 | 0.4098 | 0.4851 |
| | | 5D | 0.5456 | 0.4885 | 0.5393 | 0.3821 | 0.6921 | 0.1645 | 0.1691 | 0.3149 | 0.4120 |
| | | 7D | 0.4749 | 0.4345 | 0.4788 | 0.3437 | 0.6387 | 0.1410 | 0.1413 | 0.2653 | 0.3648 |
| | | Avg. | 0.6096 | 0.5568 | 0.5948 | 0.4425 | 0.7343 | 0.1957 | 0.2129 | 0.3888 | 0.4669 |
| | TimesNet | 1D | 0.7487 | 0.6768 | 0.7300 | 0.5653 | 0.8128 | 0.2235 | 0.2528 | 0.4910 | 0.5626 |
| | | 3D | 0.6236 | 0.5582 | 0.6225 | 0.4521 | 0.7272 | 0.1473 | 0.1642 | 0.3592 | 0.4568 |
| | | 5D | 0.5369 | 0.4827 | 0.5380 | 0.3935 | 0.6590 | 0.1184 | 0.1226 | 0.2851 | 0.3920 |
| | | 7D | 0.4596 | 0.4353 | 0.4911 | 0.3486 | 0.5869 | 0.1043 | 0.0991 | 0.2359 | 0.3451 |
| | | Avg. | 0.5922 | 0.5383 | 0.5954 | 0.4399 | 0.6965 | 0.1484 | 0.1597 | 0.3428 | 0.4391 |
| **Transformer** | iTransformer | 1D | 0.7985 | 0.7670 | 0.8047 | 0.6444 | 0.8568 | 0.2787 | 0.3549 | 0.6033 | 0.6385 |
| | | 3D | 0.6767 | 0.6518 | 0.6871 | 0.5192 | 0.7699 | 0.1998 | 0.2785 | 0.4604 | 0.5304 |
| | | 5D | 0.5748 | 0.5559 | 0.6193 | 0.4492 | 0.6981 | 0.1652 | 0.1791 | 0.3529 | 0.4493 |
| | | 7D | 0.5036 | 0.4938 | 0.5666 | 0.4040 | 0.6433 | 0.1402 | 0.1705 | 0.2874 | 0.4012 |
| | | Avg. | 0.6384 | 0.6171 | 0.6694 | 0.5042 | 0.7420 | 0.1960 | 0.2457 | 0.4260 | 0.5049 |
| | PatchTST | 1D | 0.7948 | 0.7663 | 0.8131 | 0.6364 | 0.8546 | 0.2859 | 0.3648 | 0.6001 | 0.6395 |
| | | 3D | 0.6706 | 0.6408 | 0.6990 | 0.5158 | 0.7641 | 0.2161 | 0.2558 | 0.4454 | 0.5260 |
| | | 5D | 0.5757 | 0.5538 | 0.6241 | 0.4486 | 0.6977 | 0.1833 | 0.2059 | 0.3516 | 0.4551 |
| | | 7D | 0.5016 | 0.4906 | 0.5665 | 0.4029 | 0.6450 | 0.1581 | 0.1751 | 0.2979 | 0.4047 |
| | | Avg. | 0.6357 | 0.6129 | 0.6757 | 0.5009 | 0.7404 | 0.2108 | 0.2504 | 0.4238 | 0.5063 |

Table 25: Benchmark(SEDI5) on part 2(Miami area) stations(RNN,GNN,LLM)

| | Method | T | S0 | S1 | S2 | S3 | S4 | S5 | S6 | S7 | Avg. |
|---|---|---|---|---|---|---|---|---|---|---|---|
| RNN | LSTM | 1D | 0.7782 | 0.7372 | 0.7975 | 0.6088 | 0.8707 | 0.2226 | 0.2661 | 0.5371 | 0.6023 |
| | | 3D | 0.6193 | 0.6102 | 0.6635 | 0.4559 | 0.7919 | 0.1437 | 0.1772 | 0.4233 | 0.4856 |
| | | 5D | 0.5323 | 0.5690 | 0.6237 | 0.3969 | 0.7194 | 0.1253 | 0.1418 | 0.3040 | 0.4266 |
| | | 7D | 0.4947 | 0.4718 | 0.5666 | 0.3301 | 0.6305 | 0.1009 | 0.1329 | 0.2593 | 0.3734 |
| | | Avg. | 0.6061 | 0.5971 | 0.6628 | 0.4479 | 0.7531 | 0.1481 | 0.1795 | 0.3809 | 0.4720 |
| | DeepAR | 1D | 0.7506 | 0.7444 | 0.7702 | 0.5797 | 0.8298 | 0.2470 | 0.2639 | 0.5466 | 0.5915 |
| | | 3D | 0.6252 | 0.6021 | 0.6902 | 0.4647 | 0.7750 | 0.1426 | 0.1678 | 0.4273 | 0.4869 |
| | | 5D | 0.5361 | 0.5584 | 0.5951 | 0.3839 | 0.7114 | 0.1120 | 0.1259 | 0.3603 | 0.4229 |
| | | 7D | 0.4436 | 0.5078 | 0.5647 | 0.3391 | 0.6956 | 0.0967 | 0.1104 | 0.2599 | 0.3772 |
| | | Avg. | 0.5889 | 0.6032 | 0.6551 | 0.4419 | 0.7530 | 0.1496 | 0.1670 | 0.3985 | 0.4696 |
| | DilatedRNN | 1D | 0.8054 | 0.7393 | 0.8027 | 0.6504 | 0.8471 | 0.3023 | 0.3125 | 0.5724 | 0.6290 |
| | | 3D | 0.6871 | 0.6505 | 0.7110 | 0.5126 | 0.7126 | 0.1749 | 0.2102 | 0.4345 | 0.5117 |
| | | 5D | 0.5834 | 0.5993 | 0.6388 | 0.4463 | 0.6757 | 0.1499 | 0.1713 | 0.3450 | 0.4512 |
| | | 7D | 0.5413 | 0.4900 | 0.5612 | 0.4197 | 0.6188 | 0.1372 | 0.1388 | 0.2847 | 0.3990 |
| | | Avg. | 0.6543 | 0.6198 | 0.6784 | 0.5072 | 0.7136 | 0.1911 | 0.2082 | 0.4092 | 0.4977 |
| GNN | GCN | 1D | 0.7502 | 0.7320 | 0.8148 | 0.6321 | 0.8439 | 0.3394 | 0.3757 | 0.5786 | 0.6333 |
| | | 3D | 0.6899 | 0.6528 | 0.7953 | 0.5605 | 0.7889 | 0.2891 | 0.3176 | 0.5118 | 0.5757 |
| | | 5D | 0.6957 | 0.5923 | 0.7658 | 0.5245 | 0.7818 | 0.2388 | 0.2643 | 0.4678 | 0.5414 |
| | | 7D | 0.6440 | 0.5577 | 0.7625 | 0.4949 | 0.7402 | 0.2244 | 0.2587 | 0.4532 | 0.5169 |
| | | Avg. | 0.6949 | 0.6337 | 0.7846 | 0.5530 | 0.7887 | 0.2729 | 0.3041 | 0.5029 | 0.5668 |
| | FourierGNN | 1D | 0.8279 | 0.7998 | 0.8378 | 0.6750 | 0.8669 | 0.3249 | 0.3500 | 0.4899 | 0.6465 |
| | | 3D | 0.7233 | 0.7394 | 0.7814 | 0.5680 | 0.8201 | 0.2450 | 0.2873 | 0.4436 | 0.5760 |
| | | 5D | 0.6198 | 0.6098 | 0.7593 | 0.5051 | 0.7966 | 0.2324 | 0.2902 | 0.4006 | 0.5267 |
| | | 7D | 0.6588 | 0.5544 | 0.7447 | 0.5131 | 0.7637 | 0.2154 | 0.2739 | 0.4022 | 0.5158 |
| | | Avg. | 0.7074 | 0.6758 | 0.7808 | 0.5653 | 0.8118 | 0.2544 | 0.3003 | 0.4340 | 0.5662 |
| | StemGNN | 1D | 0.8015 | 0.7321 | 0.7884 | 0.5807 | 0.8594 | 0.2851 | 0.3385 | 0.3684 | 0.5943 |
| | | 3D | 0.6608 | 0.5265 | 0.7032 | 0.4503 | 0.7169 | 0.1409 | 0.1868 | 0.2969 | 0.4603 |
| | | 5D | 0.5775 | 0.5009 | 0.6416 | 0.4189 | 0.3915 | 0.1244 | 0.1569 | 0.2450 | 0.3821 |
| | | 7D | 0.4858 | 0.4275 | 0.5752 | 0.2864 | 0.4690 | 0.0999 | 0.1371 | 0.1949 | 0.3345 |
| | | Avg. | 0.6314 | 0.5468 | 0.6771 | 0.4341 | 0.6092 | 0.1626 | 0.2048 | 0.2763 | 0.4428 |
| LLM | GPT4TS | 1D | 0.7424 | 0.6726 | 0.7407 | 0.5721 | 0.8175 | 0.2057 | 0.2580 | 0.4738 | 0.5604 |
| | | 3D | 0.6382 | 0.5471 | 0.6227 | 0.4534 | 0.7177 | 0.1509 | 0.1622 | 0.3482 | 0.4550 |
| | | 5D | 0.5484 | 0.4662 | 0.5447 | 0.3914 | 0.6360 | 0.1230 | 0.1265 | 0.2813 | 0.3897 |
| | | 7D | 0.4653 | 0.4102 | 0.4972 | 0.3462 | 0.5821 | 0.1040 | 0.1057 | 0.2184 | 0.3411 |
| | | Avg. | 0.5986 | 0.5240 | 0.6013 | 0.4408 | 0.6883 | 0.1459 | 0.1631 | 0.3304 | 0.4366 |
| | AutoTimes | 1D | 0.7923 | 0.7578 | 0.8113 | 0.6238 | 0.8590 | 0.2414 | 0.3024 | 0.5828 | 0.6213 |
| | | 3D | 0.6587 | 0.6310 | 0.6989 | 0.5032 | 0.7737 | 0.1776 | 0.2137 | 0.4392 | 0.5120 |
| | | 5D | 0.5670 | 0.5597 | 0.6312 | 0.4430 | 0.7028 | 0.1544 | 0.1690 | 0.3459 | 0.4466 |
| | | 7D | 0.4857 | 0.4895 | 0.5722 | 0.3936 | 0.6512 | 0.1312 | 0.1424 | 0.2904 | 0.3945 |
| | | Avg. | 0.6259 | 0.6095 | 0.6784 | 0.4909 | 0.7467 | 0.1761 | 0.2069 | 0.4146 | 0.4936 |

Table 26: Benchmark(SEDI1) on part 2(Miami area) stations(MLP,CNN,Transformer)

| | Method | T | S0 | S1 | S2 | S3 | S4 | S5 | S6 | S7 | Avg. |
|---|---|---|---|---|---|---|---|---|---|---|---|
| **MLP** | MLP | 1D | 0.6392 | 0.7486 | 0.7968 | 0.4636 | 0.7523 | 0.1510 | 0.1849 | 0.4622 | 0.5248 |
| | | 3D | 0.5460 | 0.6948 | 0.7434 | 0.4294 | 0.6635 | 0.1429 | 0.1752 | 0.3631 | 0.4698 |
| | | 5D | 0.5736 | 0.6829 | 0.6508 | 0.3551 | 0.6734 | 0.1527 | 0.1643 | 0.3667 | 0.4524 |
| | | 7D | 0.5641 | 0.6625 | 0.7139 | 0.4331 | 0.6920 | 0.1401 | 0.1617 | 0.3547 | 0.4653 |
| | | Avg. | **0.5807** | **0.6972** | **0.7263** | **0.4203** | **0.6953** | 0.1467 | **0.1715** | **0.3867** | **0.4781** |
| | TSMixer | 1D | 0.5899 | 0.6011 | 0.6559 | 0.3407 | 0.6954 | 0.1186 | 0.1495 | 0.2268 | 0.4222 |
| | | 3D | 0.4351 | 0.4299 | 0.4696 | 0.2279 | 0.5486 | 0.0778 | 0.0867 | 0.1369 | 0.3016 |
| | | 5D | 0.3255 | 0.3379 | 0.3531 | 0.1776 | 0.4431 | 0.0599 | 0.0690 | 0.1006 | 0.2333 |
| | | 7D | 0.2522 | 0.2693 | 0.2787 | 0.1494 | 0.3661 | 0.0524 | 0.0697 | 0.0791 | 0.1896 |
| | | Avg. | 0.4007 | 0.4096 | 0.4393 | 0.2239 | 0.5133 | 0.0772 | 0.0937 | 0.1359 | 0.2867 |
| | NLinear | 1D | 0.6032 | 0.6324 | 0.6721 | 0.3915 | 0.7185 | 0.1412 | 0.1508 | 0.3811 | 0.4613 |
| | | 3D | 0.4433 | 0.4556 | 0.4838 | 0.2745 | 0.5680 | 0.1052 | 0.1023 | 0.2296 | 0.3328 |
| | | 5D | 0.3408 | 0.3553 | 0.3639 | 0.2278 | 0.4626 | 0.0956 | 0.0801 | 0.1734 | 0.2624 |
| | | 7D | 0.2637 | 0.2851 | 0.2905 | 0.1992 | 0.3820 | 0.0851 | 0.0702 | 0.1435 | 0.2149 |
| | | Avg. | 0.4128 | 0.4321 | 0.4526 | 0.2732 | 0.5328 | 0.1068 | 0.1008 | 0.2319 | 0.3179 |
| **CNN** | TCN | 1D | 0.1324 | 0.4480 | 0.7226 | 0.2066 | 0.2128 | 0.0640 | 0.1211 | 0.0506 | 0.2448 |
| | | 3D | 0.0330 | 0.3732 | 0.5677 | 0.1723 | 0.1441 | 0.0387 | 0.1129 | 0.0265 | 0.1835 |
| | | 5D | 0.0258 | 0.2706 | 0.4580 | 0.1274 | 0.1506 | 0.0312 | 0.1130 | 0.0283 | 0.1506 |
| | | 7D | 0.0571 | 0.2566 | 0.4746 | 0.1193 | 0.1263 | 0.0128 | 0.0770 | 0.0104 | 0.1418 |
| | | Avg. | 0.0621 | 0.3371 | 0.5557 | 0.1564 | 0.1584 | 0.0367 | 0.1060 | 0.0289 | 0.1802 |
| | ModernTCN | 1D | 0.6055 | 0.5939 | 0.6605 | 0.3129 | 0.7173 | 0.1196 | 0.1522 | 0.3788 | 0.4426 |
| | | 3D | 0.4336 | 0.4005 | 0.4627 | 0.1837 | 0.5687 | 0.0721 | 0.0722 | 0.2224 | 0.3020 |
| | | 5D | 0.3390 | 0.2918 | 0.3464 | 0.1267 | 0.4670 | 0.0555 | 0.0590 | 0.1442 | 0.2287 |
| | | 7D | 0.2672 | 0.2162 | 0.2717 | 0.1025 | 0.3878 | 0.0519 | 0.0415 | 0.1163 | 0.1819 |
| | | Avg. | 0.4113 | 0.3756 | 0.4353 | 0.1814 | 0.5352 | 0.0748 | 0.0812 | 0.2154 | 0.2888 |
| | TimesNet | 1D | 0.5404 | 0.5278 | 0.6337 | 0.2851 | 0.6289 | 0.0906 | 0.0988 | 0.2989 | 0.3880 |
| | | 3D | 0.4211 | 0.3841 | 0.4394 | 0.1624 | 0.5016 | 0.0434 | 0.0522 | 0.1626 | 0.2708 |
| | | 5D | 0.3299 | 0.2943 | 0.3291 | 0.1157 | 0.3647 | 0.0295 | 0.0269 | 0.1104 | 0.2001 |
| | | 7D | 0.2249 | 0.2508 | 0.2616 | 0.0851 | 0.3262 | 0.0234 | 0.0230 | 0.0944 | 0.1612 |
| | | Avg. | 0.3791 | 0.3643 | 0.4160 | 0.1621 | 0.4554 | 0.0467 | 0.0502 | 0.1666 | 0.2550 |
| **Transformer** | iTransformer | 1D | 0.6163 | 0.6752 | 0.7437 | 0.3883 | 0.7228 | 0.1268 | 0.1668 | 0.4121 | 0.4815 |
| | | 3D | 0.4566 | 0.5015 | 0.5470 | 0.2322 | 0.5716 | 0.0745 | 0.0972 | 0.2676 | 0.3435 |
| | | 5D | 0.3503 | 0.3820 | 0.4225 | 0.1779 | 0.4512 | 0.0560 | 0.0565 | 0.1561 | 0.2566 |
| | | 7D | 0.2829 | 0.2964 | 0.3415 | 0.1450 | 0.3636 | 0.0446 | 0.0921 | 0.1163 | 0.2103 |
| | | Avg. | 0.4265 | 0.4638 | 0.5137 | 0.2359 | 0.5273 | 0.0755 | 0.1031 | 0.2380 | 0.3230 |
| | PatchTST | 1D | 0.6112 | 0.6690 | 0.7429 | 0.3695 | 0.7238 | 0.1261 | 0.1691 | 0.4179 | 0.4787 |
| | | 3D | 0.4618 | 0.4940 | 0.5613 | 0.2286 | 0.5741 | 0.0822 | 0.0949 | 0.2473 | 0.3430 |
| | | 5D | 0.3572 | 0.3782 | 0.4444 | 0.1734 | 0.4708 | 0.0668 | 0.0695 | 0.1708 | 0.2664 |
| | | 7D | 0.2791 | 0.3030 | 0.3527 | 0.1388 | 0.3865 | 0.0589 | 0.0600 | 0.1358 | 0.2143 |
| | | Avg. | 0.4273 | 0.4610 | 0.5253 | 0.2276 | 0.5388 | 0.0835 | 0.0984 | 0.2429 | 0.3256 |

Table 27: Benchmark(SEDI1) on part 2(Miami area) stations(RNN,GNN,LLM)

| | Method | T | S0 | S1 | S2 | S3 | S4 | S5 | S6 | S7 | Avg. |
|---|---|---|---|---|---|---|---|---|---|---|---|
| **RNN** | LSTM | 1D | 0.5173 | 0.5951 | 0.6803 | 0.2523 | 0.4421 | 0.0731 | 0.0896 | 0.3249 | 0.3718 |
| | | 3D | 0.3565 | 0.3467 | 0.4334 | 0.1171 | 0.2033 | 0.0380 | 0.0398 | 0.1530 | 0.2110 |
| | | 5D | 0.3267 | 0.2902 | 0.3336 | 0.0565 | 0.2339 | 0.0143 | 0.0343 | 0.1030 | 0.1740 |
| | | 7D | 0.2477 | 0.2608 | 0.4109 | 0.0553 | 0.0981 | 0.0078 | 0.0051 | 0.0914 | 0.1471 |
| | | Avg. | 0.3621 | 0.3732 | 0.4646 | 0.1203 | 0.2443 | 0.0333 | 0.0422 | 0.1681 | 0.2260 |
| | DeepAR | 1D | 0.4712 | 0.6332 | 0.6489 | 0.2446 | 0.4433 | 0.0756 | 0.1161 | 0.3283 | 0.3701 |
| | | 3D | 0.3647 | 0.3977 | 0.4097 | 0.0820 | 0.1804 | 0.0310 | 0.0387 | 0.1553 | 0.2074 |
| | | 5D | 0.2339 | 0.2331 | 0.2853 | 0.0355 | 0.1330 | 0.0220 | 0.0048 | 0.1158 | 0.1329 |
| | | 7D | 0.1074 | 0.2155 | 0.2837 | 0.0609 | 0.0813 | 0.0165 | 0.0063 | 0.1074 | 0.1099 |
| | | Avg. | 0.2943 | 0.3699 | 0.4069 | 0.1058 | 0.2095 | 0.0363 | 0.0415 | 0.1767 | 0.2051 |
| | DilatedRNN | 1D | 0.6372 | 0.5916 | 0.7119 | 0.4065 | 0.7022 | 0.1088 | 0.1333 | 0.3922 | 0.4605 |
| | | 3D | 0.5222 | 0.4714 | 0.6236 | 0.2306 | 0.4730 | 0.0741 | 0.0689 | 0.2576 | 0.3402 |
| | | 5D | 0.3632 | 0.4556 | 0.5025 | 0.1754 | 0.4427 | 0.0560 | 0.0513 | 0.1841 | 0.2789 |
| | | 7D | 0.3681 | 0.3293 | 0.4419 | 0.1630 | 0.4273 | 0.0491 | 0.0411 | 0.1600 | 0.2475 |
| | | Avg. | 0.4727 | 0.4620 | 0.5700 | 0.2439 | 0.5113 | 0.0720 | 0.0737 | 0.2485 | 0.3317 |
| **GNN** | GCN | 1D | 0.5814 | 0.5680 | 0.7359 | 0.3744 | 0.6712 | 0.1554 | 0.1852 | 0.4307 | 0.4628 |
| | | 3D | 0.5052 | 0.5117 | 0.7286 | 0.2892 | 0.5979 | 0.1691 | 0.1555 | 0.3875 | 0.4181 |
| | | 5D | 0.4462 | 0.4254 | 0.6934 | 0.2584 | 0.5692 | 0.1573 | 0.1310 | 0.3559 | 0.3796 |
| | | 7D | 0.4309 | 0.3783 | 0.6980 | 0.2593 | 0.5114 | 0.1639 | 0.1560 | 0.3369 | 0.3668 |
| | | Avg. | 0.4909 | 0.4709 | 0.7140 | 0.2953 | 0.5874 | **0.1614** | 0.1569 | 0.3777 | 0.4068 |
| | FourierGNN | 1D | 0.6570 | 0.6977 | 0.7677 | 0.4359 | 0.7147 | 0.1492 | 0.1707 | 0.3647 | 0.4947 |
| | | 3D | 0.5526 | 0.6366 | 0.6920 | 0.3083 | 0.6153 | 0.1285 | 0.1482 | 0.2883 | 0.4212 |
| | | 5D | 0.4619 | 0.4865 | 0.6791 | 0.2633 | 0.5934 | 0.1286 | 0.1450 | 0.2778 | 0.3794 |
| | | 7D | 0.6103 | 0.4415 | 0.7151 | 0.3320 | 0.6158 | 0.1261 | 0.1514 | 0.2791 | 0.4089 |
| | | Avg. | 0.5705 | 0.5656 | 0.7135 | 0.3349 | 0.6348 | 0.1331 | 0.1538 | 0.3025 | 0.4261 |
| | StemGNN | 1D | 0.5891 | 0.5290 | 0.7186 | 0.3042 | 0.6594 | 0.1308 | 0.1546 | 0.2231 | 0.4136 |
| | | 3D | 0.3107 | 0.2996 | 0.6110 | 0.1650 | 0.4861 | 0.0484 | 0.0712 | 0.1342 | 0.2658 |
| | | 5D | 0.3170 | 0.2849 | 0.5104 | 0.1400 | 0.1212 | 0.0405 | 0.0469 | 0.1600 | 0.2026 |
| | | 7D | 0.2300 | 0.2427 | 0.3921 | 0.0846 | 0.1728 | 0.0315 | 0.0513 | 0.0838 | 0.1611 |
| | | Avg. | 0.3617 | 0.3391 | 0.5580 | 0.1735 | 0.3599 | 0.0628 | 0.0810 | 0.1503 | 0.2608 |
| **LLM** | GPT4TS | 1D | 0.5577 | 0.5380 | 0.6412 | 0.2952 | 0.6627 | 0.0716 | 0.0904 | 0.2589 | 0.3895 |
| | | 3D | 0.4106 | 0.3726 | 0.4585 | 0.1692 | 0.4833 | 0.0402 | 0.0416 | 0.1690 | 0.2681 |
| | | 5D | 0.3439 | 0.2834 | 0.3346 | 0.1284 | 0.3308 | 0.0317 | 0.0291 | 0.1154 | 0.1997 |
| | | 7D | 0.2543 | 0.2254 | 0.2716 | 0.0894 | 0.2808 | 0.0250 | 0.0233 | 0.0790 | 0.1561 |
| | | Avg. | 0.3917 | 0.3548 | 0.4265 | 0.1706 | 0.4394 | 0.0421 | 0.0461 | 0.1556 | 0.2533 |
| | AutoTimes | 1D | 0.5904 | 0.6595 | 0.7269 | 0.3583 | 0.6964 | 0.1035 | 0.1381 | 0.3728 | 0.4557 |
| | | 3D | 0.4307 | 0.4794 | 0.5365 | 0.2296 | 0.5553 | 0.0609 | 0.0708 | 0.2425 | 0.3257 |
| | | 5D | 0.3386 | 0.3883 | 0.4334 | 0.2038 | 0.4503 | 0.0524 | 0.0534 | 0.1652 | 0.2607 |
| | | 7D | 0.2539 | 0.2968 | 0.3465 | 0.1403 | 0.3743 | 0.0457 | 0.0386 | 0.1488 | 0.2056 |
| | | Avg. | 0.4034 | 0.4560 | 0.5108 | 0.2330 | 0.5191 | 0.0656 | 0.0752 | 0.2323 | 0.3119 |

Table 28: Benchmark(MAE) on part 3(Fort Myers area) stations(MLP,CNN,Transformer)

| | Method | T | S0 | S1 | S2 | S3 | S4 | S5 | S6 | S7 | Avg. |
|---|---|---|---|---|---|---|---|---|---|---|---|
| **MLP** | MLP | 1D | 0.0251 | 0.0246 | 0.0207 | 0.0463 | 0.0302 | 0.0628 | 0.0505 | 0.0380 | 0.0373 |
| | | 3D | 0.0526 | 0.0607 | 0.0523 | 0.0777 | 0.0603 | 0.1228 | 0.0977 | 0.0757 | 0.0750 |
| | | 5D | 0.0839 | 0.0940 | 0.0814 | 0.1102 | 0.0958 | 0.1610 | 0.1305 | 0.1105 | 0.1084 |
| | | 7D | 0.1005 | 0.1202 | 0.1013 | 0.1327 | 0.1143 | 0.1891 | 0.1558 | 0.1281 | 0.1303 |
| | | Avg. | 0.0655 | **0.0749** | 0.0639 | 0.0917 | 0.0752 | 0.1339 | 0.1086 | 0.0881 | 0.0877 |
| | TSMixer | 1D | 0.0247 | 0.0327 | 0.0265 | 0.0477 | 0.0353 | 0.0684 | 0.0548 | 0.0461 | 0.0420 |
| | | 3D | 0.0503 | 0.0683 | 0.0560 | 0.0821 | 0.0665 | 0.1226 | 0.0987 | 0.0811 | 0.0782 |
| | | 5D | 0.0734 | 0.0989 | 0.0827 | 0.1093 | 0.0899 | 0.1596 | 0.1307 | 0.1108 | 0.1069 |
| | | 7D | 0.0946 | 0.1254 | 0.1081 | 0.1330 | 0.1100 | 0.1873 | 0.1561 | 0.1368 | 0.1314 |
| | | Avg. | 0.0608 | 0.0813 | 0.0683 | 0.0930 | 0.0754 | 0.1345 | 0.1101 | 0.0937 | 0.0896 |
| | NLinear | 1D | 0.0184 | 0.0243 | 0.0200 | 0.0427 | 0.0307 | 0.0615 | 0.0480 | 0.0377 | 0.0354 |
| | | 3D | 0.0463 | 0.0613 | 0.0516 | 0.0798 | 0.0636 | 0.1196 | 0.0947 | 0.0752 | 0.0740 |
| | | 5D | 0.0705 | 0.0931 | 0.0796 | 0.1089 | 0.0880 | 0.1577 | 0.1279 | 0.1055 | 0.1039 |
| | | 7D | 0.0933 | 0.1207 | 0.1053 | 0.1334 | 0.1078 | 0.1861 | 0.1545 | 0.1322 | 0.1292 |
| | | Avg. | **0.0571** | **0.0749** | 0.0641 | 0.0912 | 0.0725 | **0.1312** | **0.1063** | 0.0877 | 0.0856 |
| **CNN** | TCN | 1D | 0.0577 | 0.0505 | 0.0318 | 0.0970 | 0.1129 | 0.0920 | 0.2015 | 0.1758 | 0.1024 |
| | | 3D | 0.0968 | 0.1382 | 0.0689 | 0.1048 | 0.1373 | 0.1937 | 0.1485 | 0.2512 | 0.1424 |
| | | 5D | 0.1406 | 0.1002 | 0.0931 | 0.1558 | 0.1816 | 0.2363 | 0.1885 | 0.2557 | 0.1690 |
| | | 7D | 0.2256 | 0.1780 | 0.1241 | 0.1633 | 0.2116 | 0.2240 | 0.2846 | 0.2676 | 0.2098 |
| | | Avg. | 0.1302 | 0.1167 | 0.0795 | 0.1302 | 0.1608 | 0.1865 | 0.2058 | 0.2376 | 0.1559 |
| | ModernTCN | 1D | 0.0221 | 0.0265 | 0.0205 | 0.0443 | 0.0278 | 0.0654 | 0.0491 | 0.0357 | 0.0364 |
| | | 3D | 0.0567 | 0.0702 | 0.0510 | 0.0861 | 0.0603 | 0.1332 | 0.1010 | 0.0719 | 0.0788 |
| | | 5D | 0.0871 | 0.1048 | 0.0834 | 0.1215 | 0.0883 | 0.1774 | 0.1384 | 0.1015 | 0.1128 |
| | | 7D | 0.1202 | 0.1362 | 0.1056 | 0.1499 | 0.1094 | 0.2057 | 0.1686 | 0.1280 | 0.1405 |
| | | Avg. | 0.0715 | 0.0845 | 0.0651 | 0.1005 | 0.0714 | 0.1454 | 0.1143 | 0.0843 | 0.0921 |
| | TimesNet | 1D | 0.0199 | 0.0288 | 0.0242 | 0.0513 | 0.0445 | 0.0809 | 0.0651 | 0.0528 | 0.0459 |
| | | 3D | 0.0467 | 0.0651 | 0.0558 | 0.0875 | 0.0724 | 0.1426 | 0.1164 | 0.0866 | 0.0841 |
| | | 5D | 0.0703 | 0.0970 | 0.0838 | 0.1148 | 0.0988 | 0.1852 | 0.1484 | 0.1127 | 0.1139 |
| | | 7D | 0.0923 | 0.1284 | 0.1121 | 0.1438 | 0.1158 | 0.2145 | 0.1759 | 0.1417 | 0.1406 |
| | | Avg. | 0.0573 | 0.0798 | 0.0690 | 0.0993 | 0.0829 | 0.1558 | 0.1264 | 0.0984 | 0.0961 |
| **Transformer** | iTransformer | 1D | 0.0190 | 0.0253 | 0.0192 | 0.0383 | 0.0278 | 0.0641 | 0.0482 | 0.0345 | 0.0346 |
| | | 3D | 0.0469 | 0.0617 | 0.0487 | 0.0726 | 0.0576 | 0.1244 | 0.0959 | 0.0706 | 0.0723 |
| | | 5D | 0.0709 | 0.0948 | 0.0766 | 0.1023 | 0.0834 | 0.1633 | 0.1331 | 0.0977 | 0.1028 |
| | | 7D | 0.0914 | 0.1226 | 0.1041 | 0.1275 | 0.1048 | 0.1889 | 0.1596 | 0.1214 | 0.1275 |
| | | Avg. | **0.0571** | 0.0761 | 0.0621 | **0.0852** | **0.0684** | 0.1352 | 0.1092 | **0.0811** | **0.0843** |
| | PatchTST | 1D | 0.0183 | 0.0241 | 0.0177 | 0.0381 | 0.0288 | 0.0637 | 0.0481 | 0.0349 | 0.0342 |
| | | 3D | 0.0464 | 0.0654 | 0.0545 | 0.0765 | 0.0588 | 0.1242 | 0.1007 | 0.0714 | 0.0747 |
| | | 5D | 0.0762 | 0.0951 | 0.0761 | 0.1057 | 0.0835 | 0.1651 | 0.1354 | 0.1014 | 0.1048 |
| | | 7D | 0.0984 | 0.1266 | 0.1032 | 0.1319 | 0.1053 | 0.1955 | 0.1608 | 0.1257 | 0.1309 |
| | | Avg. | 0.0598 | 0.0778 | 0.0629 | 0.0881 | 0.0691 | 0.1371 | 0.1112 | 0.0833 | 0.0862 |

Table 29: Benchmark(MAE) on part 3(Fort Myers area) stations(RNN,GNN,LLM)

| | Method | T | S0 | S1 | S2 | S3 | S4 | S5 | S6 | S7 | Avg. |
|---|---|---|---|---|---|---|---|---|---|---|---|
| **RNN** | LSTM | 1D | 0.0486 | 0.0447 | 0.0393 | 0.0640 | 0.0465 | 0.1120 | 0.0839 | 0.0606 | 0.0625 |
| | | 3D | 0.0895 | 0.0878 | 0.0788 | 0.1048 | 0.0828 | 0.1775 | 0.1299 | 0.1091 | 0.1075 |
| | | 5D | 0.1246 | 0.1247 | 0.1167 | 0.1344 | 0.1172 | 0.2057 | 0.1590 | 0.1281 | 0.1388 |
| | | 7D | 0.1621 | 0.1584 | 0.1496 | 0.1725 | 0.1389 | 0.2203 | 0.1823 | 0.1567 | 0.1676 |
| | | Avg. | 0.1062 | 0.1039 | 0.0961 | 0.1189 | 0.0964 | 0.1789 | 0.1388 | 0.1136 | 0.1191 |
| | DeepAR | 1D | 0.0569 | 0.0461 | 0.0381 | 0.0632 | 0.0486 | 0.1023 | 0.0763 | 0.0694 | 0.0626 |
| | | 3D | 0.0859 | 0.0950 | 0.0851 | 0.1180 | 0.0949 | 0.1640 | 0.1403 | 0.0993 | 0.1103 |
| | | 5D | 0.1086 | 0.1219 | 0.1137 | 0.1424 | 0.1388 | 0.2005 | 0.1742 | 0.1466 | 0.1433 |
| | | 7D | 0.1496 | 0.1542 | 0.1549 | 0.1745 | 0.1467 | 0.2247 | 0.2012 | 0.1485 | 0.1693 |
| | | Avg. | 0.1003 | 0.1043 | 0.0980 | 0.1245 | 0.1073 | 0.1729 | 0.1480 | 0.1159 | 0.1214 |
| | DilatedRNN | 1D | 0.0394 | 0.0268 | 0.0277 | 0.0441 | 0.0472 | 0.0691 | 0.0596 | 0.0372 | 0.0439 |
| | | 3D | 0.0726 | 0.0714 | 0.0679 | 0.0978 | 0.0648 | 0.1390 | 0.1186 | 0.0837 | 0.0895 |
| | | 5D | 0.1100 | 0.1028 | 0.0887 | 0.1257 | 0.0966 | 0.2000 | 0.1578 | 0.1128 | 0.1243 |
| | | 7D | 0.1312 | 0.1500 | 0.1175 | 0.1499 | 0.1234 | 0.2363 | 0.1870 | 0.1478 | 0.1554 |
| | | Avg. | 0.0883 | 0.0877 | 0.0754 | 0.1043 | 0.0830 | 0.1611 | 0.1308 | 0.0954 | 0.1033 |
| **GNN** | GCN | 1D | 0.0543 | 0.0596 | 0.0264 | 0.3354 | 0.1422 | 0.0789 | 0.0946 | 0.1151 | 0.1133 |
| | | 3D | 0.0858 | 0.0942 | 0.0598 | 0.3643 | 0.1756 | 0.1443 | 0.1374 | 0.1593 | 0.1526 |
| | | 5D | 0.1006 | 0.1192 | 0.0851 | 0.3859 | 0.2002 | 0.1767 | 0.1713 | 0.1952 | 0.1793 |
| | | 7D | 0.1196 | 0.1408 | 0.1141 | 0.4079 | 0.2197 | 0.2033 | 0.2068 | 0.2315 | 0.2055 |
| | | Avg. | 0.0901 | 0.1034 | 0.0713 | 0.3734 | 0.1845 | 0.1508 | 0.1525 | 0.1753 | 0.1627 |
| | FourierGNN | 1D | 10.0224 | 0.0295 | 0.0262 | 0.0462 | 0.0370 | 0.0732 | 0.0543 | 0.0436 | 0.0415 |
| | | 3D | 0.0548 | 0.0673 | 0.0538 | 0.0829 | 0.0784 | 0.1336 | 0.1087 | 0.0841 | 0.0830 |
| | | 5D | 0.0830 | 0.0978 | 0.0814 | 0.1153 | 0.0967 | 0.1668 | 0.1511 | 0.1109 | 0.1129 |
| | | 7D | 0.1064 | 0.1236 | 0.1114 | 0.1452 | 0.1335 | 0.2015 | 0.1823 | 0.1354 | 0.1424 |
| | | Avg. | 0.0666 | 0.0796 | 0.0682 | 0.0974 | 0.0864 | 0.1438 | 0.1241 | 0.0935 | 0.0949 |
| | StemGNN | 1D | 0.0636 | 0.0445 | 0.0335 | 0.0635 | 0.0586 | 0.1001 | 0.0599 | 0.0489 | 0.0591 |
| | | 3D | 0.1114 | 0.1001 | 0.0788 | 0.1273 | 0.1028 | 0.1524 | 0.1421 | 0.1408 | 0.1195 |
| | | 5D | 0.1690 | 0.1688 | 0.1022 | 0.1698 | 0.1352 | 0.1980 | 0.2366 | 0.2161 | 0.1745 |
| | | 7D | 0.2094 | 0.2090 | 0.1719 | 0.2016 | 0.1625 | 0.2325 | 0.2514 | 0.2730 | 0.2139 |
| | | Avg. | 0.1384 | 0.1306 | 0.0966 | 0.1406 | 0.1148 | 0.1707 | 0.1725 | 0.1697 | 0.1417 |
| **LLM** | GPT4TS | 1D | 0.0209 | 0.0287 | 0.0261 | 0.0482 | 0.0407 | 0.0740 | 0.0596 | 0.7614 | 0.1324 |
| | | 3D | 0.0467 | 0.0638 | 0.0553 | 0.0845 | 0.0730 | 0.1431 | 0.1105 | 0.0869 | 0.0830 |
| | | 5D | 0.0702 | 0.0946 | 0.0832 | 0.1141 | 0.0980 | 0.1823 | 0.1455 | 0.1120 | 0.1125 |
| | | 7D | 0.0909 | 0.1231 | 0.1105 | 0.1392 | 0.1160 | 0.2087 | 0.1715 | 0.1397 | 0.1375 |
| | | Avg. | 0.0572 | 0.0776 | 0.0688 | 0.0965 | 0.0819 | 0.1520 | 0.1218 | 0.2750 | 0.1163 |
| | AutoTimes | 1D | 0.0228 | 0.0282 | 0.0199 | 0.0429 | 0.0321 | 0.0680 | 0.0542 | 0.0407 | 0.0386 |
| | | 3D | 0.0475 | 0.0615 | 0.0486 | 0.0760 | 0.0638 | 0.1240 | 0.0993 | 0.0751 | 0.0745 |
| | | 5D | 0.0692 | 0.0965 | 0.0737 | 0.1075 | 0.0892 | 0.1632 | 0.1315 | 0.1056 | 0.1045 |
| | | 7D | 0.0921 | 0.1218 | 0.1038 | 0.1304 | 0.1083 | 0.1900 | 0.1573 | 0.1285 | 0.1290 |
| | | Avg. | 0.0579 | 0.0770 | **0.0615** | 0.0892 | 0.0734 | 0.1363 | 0.1106 | 0.0875 | 0.0867 |

Table 30: Benchmark(MSE) on part 3(Fort Myers area) stations(MLP,CNN,Transformer)

| | Method | T | S0 | S1 | S2 | S3 | S4 | S5 | S6 | S7 | Avg. |
|---|---|---|---|---|---|---|---|---|---|---|---|
| **MLP** | MLP | 1D | 0.0036 | 0.0046 | 0.0052 | 0.0165 | 0.0137 | 0.0368 | 0.0139 | 0.0095 | 0.0130 |
| | | 3D | 0.0156 | 0.0159 | 0.0199 | 0.0359 | 0.0318 | 0.0888 | 0.0374 | 0.0252 | 0.0338 |
| | | 5D | 0.0295 | 0.0300 | 0.0345 | 0.0538 | 0.0486 | 0.1253 | 0.0552 | 0.0426 | 0.0524 |
| | | 7D | 0.0461 | 0.0442 | 0.0494 | 0.0729 | 0.0635 | 0.1537 | 0.0701 | 0.0562 | 0.0695 |
| | | Avg. | **0.0237** | **0.0237** | 0.0272 | **0.0448** | **0.0394** | **0.1011** | **0.0441** | **0.0334** | **0.0422** |
| | TSMixer | 1D | 0.0047 | 0.0056 | 0.0070 | 0.0182 | 0.0163 | 0.0420 | 0.0159 | 0.0116 | 0.0152 |
| | | 3D | 0.0176 | 0.0174 | 0.0217 | 0.0381 | 0.0353 | 0.0983 | 0.0403 | 0.0280 | 0.0371 |
| | | 5D | 0.0348 | 0.0315 | 0.0372 | 0.0561 | 0.0522 | 0.1410 | 0.0605 | 0.0452 | 0.0573 |
| | | 7D | 0.0548 | 0.0465 | 0.0531 | 0.0740 | 0.0679 | 0.1735 | 0.0778 | 0.0624 | 0.0763 |
| | | Avg. | 0.0280 | 0.0253 | 0.0297 | 0.0466 | 0.0429 | 0.1137 | 0.0486 | 0.0368 | 0.0465 |
| | NLinear | 1D | 0.0031 | 0.0044 | 0.0050 | 0.0174 | 0.0158 | 0.0392 | 0.0138 | 0.0098 | 0.0136 |
| | | 3D | 0.0153 | 0.0159 | 0.0202 | 0.0378 | 0.0349 | 0.0970 | 0.0383 | 0.0263 | 0.0357 |
| | | 5D | 0.0321 | 0.0298 | 0.0362 | 0.0561 | 0.0518 | 0.1392 | 0.0581 | 0.0433 | 0.0558 |
| | | 7D | 0.0520 | 0.0449 | 0.0526 | 0.0742 | 0.0677 | 0.1724 | 0.0754 | 0.0607 | 0.0750 |
| | | Avg. | 0.0256 | 0.0238 | 0.0285 | 0.0464 | 0.0426 | 0.1120 | 0.0464 | 0.0350 | 0.0450 |
| **CNN** | TCN | 1D | 0.0925 | 0.0196 | 0.0078 | 0.0383 | 0.0743 | 0.0452 | 0.0900 | 0.1310 | 0.0623 |
| | | 3D | 0.1029 | 0.1063 | 0.0244 | 0.0440 | 0.0804 | 0.1231 | 0.0514 | 0.1998 | 0.0916 |
| | | 5D | 0.1186 | 0.0304 | 0.0379 | 0.0681 | 0.1249 | 0.1691 | 0.0767 | 0.2183 | 0.1055 |
| | | 7D | 0.1905 | 0.1158 | 0.0575 | 0.0808 | 0.1354 | 0.1655 | 0.1613 | 0.1934 | 0.1375 |
| | | Avg. | 0.1261 | 0.0680 | 0.0319 | 0.0578 | 0.1038 | 0.1257 | 0.0949 | 0.1856 | 0.0992 |
| | ModernTCN | 1D | 0.0048 | 0.0050 | 0.0045 | 0.0208 | 0.0142 | 0.0451 | 0.0148 | 0.0100 | 0.0149 |
| | | 3D | 0.0219 | 0.0210 | 0.0187 | 0.0451 | 0.0345 | 0.1297 | 0.0455 | 0.0276 | 0.0430 |
| | | 5D | 0.0466 | 0.0406 | 0.0363 | 0.0757 | 0.0548 | 0.1949 | 0.0722 | 0.0451 | 0.0708 |
| | | 7D | 0.0710 | 0.0626 | 0.0527 | 0.1014 | 0.0705 | 0.2231 | 0.0981 | 0.0619 | 0.0927 |
| | | Avg. | 0.0361 | 0.0323 | 0.0280 | 0.0607 | 0.0435 | 0.1482 | 0.0576 | 0.0362 | 0.0553 |
| | TimesNet | 1D | 0.0035 | 0.0052 | 0.0066 | 0.0215 | 0.0262 | 0.0586 | 0.0206 | 0.0163 | 0.0198 |
| | | 3D | 0.0159 | 0.0181 | 0.0214 | 0.0461 | 0.0445 | 0.1318 | 0.0553 | 0.0331 | 0.0458 |
| | | 5D | 0.0315 | 0.0351 | 0.0369 | 0.0654 | 0.0684 | 0.2087 | 0.0811 | 0.0509 | 0.0723 |
| | | 7D | 0.0514 | 0.0543 | 0.0537 | 0.0919 | 0.0871 | 0.2343 | 0.1012 | 0.0706 | 0.0931 |
| | | Avg. | 0.0256 | 0.0282 | 0.0297 | 0.0562 | 0.0565 | 0.1584 | 0.0645 | 0.0427 | 0.0577 |
| **Transformer** | iTransformer | 1D | 0.0035 | 0.0047 | 0.0050 | 0.0177 | 0.0157 | 0.0436 | 0.0140 | 0.0094 | 0.0142 |
| | | 3D | 0.0171 | 0.0171 | 0.0209 | 0.0403 | 0.0352 | 0.1103 | 0.0410 | 0.0255 | 0.0384 |
| | | 5D | 0.0343 | 0.0335 | 0.0356 | 0.0623 | 0.0527 | 0.1582 | 0.0674 | 0.0423 | 0.0608 |
| | | 7D | 0.0518 | 0.0522 | 0.0539 | 0.0807 | 0.0714 | 0.1897 | 0.0847 | 0.0590 | 0.0804 |
| | | Avg. | 0.0267 | 0.0269 | 0.0288 | 0.0503 | 0.0437 | 0.1254 | 0.0518 | 0.0341 | 0.0485 |
| | PatchTST | 1D | 0.0032 | 0.0048 | 0.0044 | 0.0179 | 0.0154 | 0.0444 | 0.0144 | 0.0096 | 0.0143 |
| | | 3D | 0.0161 | 0.0205 | 0.0212 | 0.0420 | 0.0353 | 0.1100 | 0.0432 | 0.0266 | 0.0393 |
| | | 5D | 0.0385 | 0.0364 | 0.0374 | 0.0608 | 0.0531 | 0.1632 | 0.0668 | 0.0438 | 0.0625 |
| | | 7D | 0.0564 | 0.0578 | 0.0514 | 0.0832 | 0.0700 | 0.2020 | 0.0825 | 0.0598 | 0.0829 |
| | | Avg. | 0.0285 | 0.0299 | 0.0286 | 0.0510 | 0.0435 | 0.1299 | 0.0517 | 0.0350 | 0.0498 |

Table 31: Benchmark(MSE) on part 3(Fort Myers area) stations(RNN,GNN,LLM)

| | Method | T | S0 | S1 | S2 | S3 | S4 | S5 | S6 | S7 | Avg. |
|---|---|---|---|---|---|---|---|---|---|---|---|
| **RNN** | LSTM | 1D | 0.0183 | 0.0087 | 0.0109 | 0.0287 | 0.0217 | 0.0748 | 0.0274 | 0.0181 | 0.0261 |
| | | 3D | 0.0420 | 0.0271 | 0.0312 | 0.0588 | 0.0444 | 0.1455 | 0.0540 | 0.0381 | 0.0551 |
| | | 5D | 0.0624 | 0.0427 | 0.0506 | 0.0774 | 0.0630 | 0.1761 | 0.0717 | 0.0552 | 0.0749 |
| | | 7D | 0.0949 | 0.0621 | 0.0773 | 0.0985 | 0.0797 | 0.2013 | 0.0880 | 0.0714 | 0.0967 |
| | | Avg. | 0.0544 | 0.0352 | 0.0425 | 0.0659 | 0.0522 | 0.1494 | 0.0603 | 0.0457 | 0.0632 |
| | DeepAR | 1D | 0.0408 | 0.0090 | 0.0108 | 0.0298 | 0.0224 | 0.0750 | 0.0266 | 0.0173 | 0.0290 |
| | | 3D | 0.0634 | 0.0251 | 0.0316 | 0.0568 | 0.0463 | 0.1353 | 0.0580 | 0.0370 | 0.0567 |
| | | 5D | 0.0682 | 0.0414 | 0.0506 | 0.0743 | 0.0642 | 0.1775 | 0.0754 | 0.0564 | 0.0760 |
| | | 7D | 0.0910 | 0.0569 | 0.0728 | 0.0969 | 0.0799 | 0.1947 | 0.0899 | 0.0703 | 0.0941 |
| | | Avg. | 0.0659 | 0.0331 | 0.0414 | 0.0644 | 0.0532 | 0.1457 | 0.0625 | 0.0453 | 0.0639 |
| | DilatedRNN | 1D | 0.0172 | 0.0048 | 0.0056 | 0.0194 | 0.0162 | 0.0408 | 0.0175 | 0.0104 | 0.0165 |
| | | 3D | 0.0395 | 0.0193 | 0.0225 | 0.0442 | 0.0356 | 0.1128 | 0.0489 | 0.0300 | 0.0441 |
| | | 5D | 0.0583 | 0.0347 | 0.0401 | 0.0675 | 0.0560 | 0.1787 | 0.0786 | 0.0509 | 0.0706 |
| | | 7D | 0.0808 | 0.0583 | 0.0578 | 0.0958 | 0.0747 | 0.2167 | 0.0990 | 0.0714 | 0.0943 |
| | | Avg. | 0.0489 | 0.0293 | 0.0315 | 0.0567 | 0.0457 | 0.1372 | 0.0610 | 0.0407 | 0.0564 |
| **GNN** | GCN | 1D | 0.0108 | 0.0094 | 0.0060 | 0.3682 | 0.0982 | 0.0392 | 0.0227 | 0.0923 | 0.0809 |
| | | 3D | 0.0267 | 0.0246 | 0.0196 | 0.3854 | 0.1184 | 0.0918 | 0.0472 | 0.1217 | 0.1044 |
| | | 5D | 0.0434 | 0.0359 | 0.0339 | 0.4006 | 0.1337 | 0.1285 | 0.0674 | 0.1440 | 0.1234 |
| | | 7D | 0.0569 | 0.0499 | 0.0486 | 0.4154 | 0.1489 | 0.1565 | 0.0868 | 0.1794 | 0.1428 |
| | | Avg. | 0.0344 | 0.0300 | **0.0270** | 0.3924 | 0.1248 | 0.1040 | 0.0560 | 0.1343 | 0.1129 |
| | FourierGNN | 1D | 0.0037 | 0.0049 | 0.0063 | 0.0178 | 0.0153 | 0.0430 | 0.0145 | 0.0108 | 0.0145 |
| | | 3D | 0.0154 | 0.0166 | 0.0207 | 0.0379 | 0.0364 | 0.0938 | 0.0399 | 0.0268 | 0.0359 |
| | | 5D | 0.0298 | 0.0302 | 0.0359 | 0.0555 | 0.0496 | 0.1312 | 0.0609 | 0.0432 | 0.0545 |
| | | 7D | 0.0477 | 0.0434 | 0.0542 | 0.0769 | 0.0691 | 0.1594 | 0.0787 | 0.0591 | 0.0736 |
| | | Avg. | 0.0241 | **0.0237** | 0.0293 | 0.0471 | 0.0426 | 0.1069 | 0.0485 | 0.0350 | 0.0446 |
| | StemGNN | 1D | 0.0390 | 0.0081 | 0.0070 | 0.0221 | 0.0281 | 0.0535 | 0.0164 | 0.0145 | 0.0236 |
| | | 3D | 0.0816 | 0.0405 | 0.0270 | 0.0609 | 0.0599 | 0.1179 | 0.0722 | 0.0861 | 0.0683 |
| | | 5D | 0.1100 | 0.0850 | 0.0433 | 0.1015 | 0.0809 | 0.1587 | 0.1540 | 0.1312 | 0.1081 |
| | | 7D | 0.1554 | 0.1173 | 0.1096 | 0.1232 | 0.1079 | 0.1982 | 0.1746 | 0.2072 | 0.1492 |
| | | Avg. | 0.0965 | 0.0628 | 0.0467 | 0.0769 | 0.0692 | 0.1321 | 0.1043 | 0.1098 | 0.0873 |
| **LLM** | GPT4TS | 1D | 0.0038 | 0.0050 | 0.0060 | 0.0202 | 0.0223 | 0.0542 | 0.0183 | 0.6054 | 0.0919 |
| | | 3D | 0.0151 | 0.0174 | 0.0205 | 0.0442 | 0.0474 | 0.1530 | 0.0496 | 0.0339 | 0.0476 |
| | | 5D | 0.0321 | 0.0319 | 0.0371 | 0.0648 | 0.0716 | 0.2071 | 0.0829 | 0.0526 | 0.0725 |
| | | 7D | 0.0494 | 0.0489 | 0.0528 | 0.0872 | 0.0841 | 0.2397 | 0.0996 | 0.0734 | 0.0919 |
| | | Avg. | 0.0251 | 0.0258 | 0.0291 | 0.0541 | 0.0564 | 0.1635 | 0.0626 | 0.1913 | 0.0760 |
| | AutoTimes | 1D | 0.0037 | 0.0052 | 0.0048 | 0.0183 | 0.0159 | 0.0447 | 0.0161 | 0.0106 | 0.0149 |
| | | 3D | 0.0159 | 0.0167 | 0.0189 | 0.0396 | 0.0342 | 0.1026 | 0.0409 | 0.0263 | 0.0369 |
| | | 5D | 0.0321 | 0.0314 | 0.0337 | 0.0564 | 0.0519 | 0.1485 | 0.0614 | 0.0429 | 0.0573 |
| | | 7D | 0.0522 | 0.0466 | 0.0508 | 0.0745 | 0.0661 | 0.1807 | 0.0781 | 0.0586 | 0.0760 |
| | | Avg. | 0.0260 | 0.0250 | 0.0271 | 0.0472 | 0.0420 | 0.1191 | 0.0491 | 0.0346 | 0.0463 |

Table 32: Benchmark(SEDI10) on part 3(Fort Myers area) stations(MLP,CNN,Transformer)

| | Method | T | S0 | S1 | S2 | S3 | S4 | S5 | S6 | S7 | Avg. |
|---|---|---|---|---|---|---|---|---|---|---|---|
| **MLP** | MLP | 1D | 0.6714 | 0.9580 | 0.6989 | 0.7088 | 0.4910 | 0.6202 | 0.5661 | 0.8442 | 0.6948 |
| | | 3D | 0.6107 | 0.8594 | 0.6242 | 0.7260 | 0.4937 | 0.4783 | 0.5238 | 0.8326 | 0.6436 |
| | | 5D | 0.5833 | 0.7752 | 0.6230 | 0.6262 | 0.4892 | 0.5590 | 0.5791 | 0.7968 | 0.6290 |
| | | 7D | 0.5649 | 0.7550 | 0.5619 | 0.5997 | 0.5046 | 0.4772 | 0.4864 | 0.6886 | 0.5798 |
| | | Avg. | 0.6076 | 0.8369 | 0.6270 | 0.6652 | **0.4946** | **0.5337** | 0.5389 | **0.7906** | **0.6368** |
| | TSMixer | 1D | 0.6675 | 0.9189 | 0.6900 | 0.7345 | 0.5080 | 0.6620 | 0.6030 | 0.8160 | 0.7000 |
| | | 3D | 0.5919 | 0.8285 | 0.6258 | 0.6505 | 0.4595 | 0.5203 | 0.4941 | 0.7342 | 0.6131 |
| | | 5D | 0.5567 | 0.7866 | 0.5754 | 0.6006 | 0.4316 | 0.4412 | 0.4199 | 0.6803 | 0.5615 |
| | | 7D | 0.5350 | 0.7498 | 0.5379 | 0.5651 | 0.4129 | 0.3812 | 0.3557 | 0.6375 | 0.5219 |
| | | Avg. | 0.5878 | 0.8209 | 0.6073 | 0.6377 | 0.4530 | 0.5012 | 0.4682 | 0.7170 | 0.5991 |
| | NLinear | 1D | 0.6653 | 0.9466 | 0.7012 | 0.7387 | 0.5234 | 0.6204 | 0.6081 | 0.8296 | 0.7041 |
| | | 3D | 0.5963 | 0.8470 | 0.6314 | 0.6528 | 0.4532 | 0.5048 | 0.4892 | 0.7448 | 0.6149 |
| | | 5D | 0.5604 | 0.7925 | 0.5800 | 0.6023 | 0.4297 | 0.4342 | 0.4069 | 0.6914 | 0.5622 |
| | | 7D | 0.5393 | 0.7539 | 0.5417 | 0.5660 | 0.4113 | 0.3787 | 0.3420 | 0.6494 | 0.5228 |
| | | Avg. | 0.5903 | 0.8350 | 0.6136 | 0.6399 | 0.4544 | 0.4845 | 0.4615 | 0.7288 | 0.6010 |
| **CNN** | TCN | 1D | 0.6607 | 0.9349 | 0.7126 | 0.6214 | 0.3251 | 0.6057 | 0.3817 | 0.4229 | 0.5831 |
| | | 3D | 0.5849 | 0.7254 | 0.6607 | 0.6150 | 0.3200 | 0.3803 | 0.4778 | 0.3677 | 0.5165 |
| | | 5D | 0.4883 | 0.8096 | 0.5672 | 0.5006 | 0.2824 | 0.3337 | 0.3708 | 0.4034 | 0.4695 |
| | | 7D | 0.2490 | 0.6355 | 0.5764 | 0.6034 | 0.2939 | 0.3018 | 0.2916 | 0.3836 | 0.4169 |
| | | Avg. | 0.4957 | 0.7763 | 0.6292 | 0.5851 | 0.3054 | 0.4054 | 0.3805 | 0.3944 | 0.4965 |
| | ModernTCN | 1D | 0.6643 | 0.9457 | 0.6975 | 0.7506 | 0.5325 | 0.6309 | 0.5645 | 0.8346 | 0.7026 |
| | | 3D | 0.5833 | 0.8013 | 0.6528 | 0.6401 | 0.4782 | 0.4488 | 0.4296 | 0.7353 | 0.5962 |
| | | 5D | 0.5274 | 0.7259 | 0.5842 | 0.6140 | 0.4395 | 0.3732 | 0.3519 | 0.6687 | 0.5356 |
| | | 7D | 0.4967 | 0.6819 | 0.5463 | 0.5468 | 0.4399 | 0.3177 | 0.2592 | 0.6290 | 0.4897 |
| | | Avg. | 0.5679 | 0.7887 | 0.6202 | 0.6379 | 0.4725 | 0.4426 | 0.4013 | 0.7169 | 0.5810 |
| | TimesNet | 1D | 0.6628 | 0.8907 | 0.6824 | 0.7392 | 0.4722 | 0.5926 | 0.5199 | 0.7606 | 0.6651 |
| | | 3D | 0.5922 | 0.8103 | 0.6122 | 0.6461 | 0.4452 | 0.4391 | 0.3654 | 0.6902 | 0.5751 |
| | | 5D | 0.5666 | 0.7517 | 0.5491 | 0.6090 | 0.4134 | 0.3682 | 0.3255 | 0.6374 | 0.5276 |
| | | 7D | 0.5199 | 0.7043 | 0.5105 | 0.5579 | 0.3969 | 0.2931 | 0.2576 | 0.5950 | 0.4794 |
| | | Avg. | 0.5854 | 0.7892 | 0.5885 | 0.6380 | 0.4319 | 0.4233 | 0.3671 | 0.6708 | 0.5618 |
| **Transformer** | iTransformer | 1D | 0.6753 | 0.9308 | 0.6948 | 0.7630 | 0.5149 | 0.6342 | 0.5745 | 0.8270 | 0.7018 |
| | | 3D | 0.5920 | 0.8284 | 0.6263 | 0.6781 | 0.4784 | 0.4697 | 0.4621 | 0.7383 | 0.6092 |
| | | 5D | 0.5688 | 0.7619 | 0.5898 | 0.6217 | 0.4517 | 0.4084 | 0.3802 | 0.6824 | 0.5581 |
| | | 7D | 0.5440 | 0.7203 | 0.5514 | 0.5852 | 0.4287 | 0.3532 | 0.3130 | 0.6412 | 0.5171 |
| | | Avg. | 0.5950 | 0.8103 | 0.6156 | 0.6620 | 0.4684 | 0.4664 | 0.4325 | 0.7222 | 0.5965 |
| | PatchTST | 1D | 0.6779 | 0.9328 | 0.7007 | 0.7626 | 0.5236 | 0.6109 | 0.5828 | 0.8365 | 0.7035 |
| | | 3D | 0.6072 | 0.8306 | 0.6327 | 0.6674 | 0.4839 | 0.4845 | 0.4406 | 0.7434 | 0.6113 |
| | | 5D | 0.5537 | 0.7707 | 0.5892 | 0.6232 | 0.4570 | 0.4017 | 0.3710 | 0.6887 | 0.5569 |
| | | 7D | 0.5374 | 0.7078 | 0.5492 | 0.5853 | 0.4358 | 0.3455 | 0.3136 | 0.6471 | 0.5152 |
| | | Avg. | 0.5941 | 0.8105 | 0.6180 | 0.6596 | 0.4751 | 0.4606 | 0.4270 | 0.7289 | 0.5967 |

Table 33: Benchmark(SEDI10) on part 3(Fort Myers area) stations(RNN,GNN,LLM)

| | Method | T | S0 | S1 | S2 | S3 | S4 | S5 | S6 | S7 | Avg. |
|---|---|---|---|---|---|---|---|---|---|---|---|
| **RNN** | LSTM | 1D | 0.6375 | 0.8919 | 0.6771 | 0.7127 | 0.5130 | 0.6141 | 0.4802 | 0.6979 | 0.6530 |
| | | 3D | 0.5603 | 0.7366 | 0.5969 | 0.7031 | 0.4666 | 0.4219 | 0.4553 | 0.6716 | 0.5765 |
| | | 5D | 0.5230 | 0.7399 | 0.5582 | 0.6459 | 0.4201 | 0.3505 | 0.4925 | 0.6755 | 0.5507 |
| | | 7D | 0.5015 | 0.7195 | 0.5452 | 0.5964 | 0.4224 | 0.3320 | 0.4663 | 0.6533 | 0.5296 |
| | | Avg. | 0.5556 | 0.7720 | 0.5943 | 0.6645 | 0.4555 | 0.4296 | 0.4736 | 0.6746 | 0.5775 |
| | DeepAR | 1D | 0.6113 | 0.9191 | 0.5886 | 0.7494 | 0.5485 | 0.4949 | 0.4766 | 0.7039 | 0.6365 |
| | | 3D | 0.5456 | 0.7545 | 0.6393 | 0.6426 | 0.4689 | 0.3235 | 0.3871 | 0.7007 | 0.5578 |
| | | 5D | 0.5199 | 0.7361 | 0.5829 | 0.6406 | 0.3841 | 0.2802 | 0.2172 | 0.5790 | 0.4925 |
| | | 7D | 0.5267 | 0.7149 | 0.4846 | 0.5765 | 0.4471 | 0.2646 | 0.3546 | 0.5960 | 0.4956 |
| | | Avg. | 0.5509 | 0.7811 | 0.5738 | 0.6523 | 0.4621 | 0.3408 | 0.3589 | 0.6449 | 0.5456 |
| | DilatedRNN | 1D | 0.6157 | 0.9270 | 0.7114 | 0.8285 | 0.4265 | 0.5347 | 0.4866 | 0.7628 | 0.6616 |
| | | 3D | 0.5809 | 0.7952 | 0.6612 | 0.5798 | 0.4546 | 0.3865 | 0.3726 | 0.7252 | 0.5695 |
| | | 5D | 0.5291 | 0.8040 | 0.5665 | 0.6098 | 0.4765 | 0.2496 | 0.2638 | 0.6925 | 0.5239 |
| | | 7D | 0.5392 | 0.6853 | 0.5523 | 0.5684 | 0.4365 | 0.2247 | 0.3043 | 0.5970 | 0.4885 |
| | | Avg. | 0.5662 | 0.8029 | 0.6228 | 0.6466 | 0.4485 | 0.3489 | 0.3568 | 0.6944 | 0.5609 |
| **GNN** | GCN | 1D | 0.6792 | 0.8309 | 0.7292 | 0.4914 | 0.4288 | 0.5120 | 0.4721 | 0.7129 | 0.6071 |
| | | 3D | 0.6144 | 0.8262 | 0.6910 | 0.4325 | 0.3749 | 0.4319 | 0.5107 | 0.6594 | 0.5676 |
| | | 5D | 0.5949 | 0.7921 | 0.6016 | 0.4126 | 0.3364 | 0.3577 | 0.3270 | 0.6403 | 0.5078 |
| | | 7D | 0.5728 | 0.7445 | 0.6076 | 0.3891 | 0.3311 | 0.2809 | 0.3782 | 0.6840 | 0.4985 |
| | | Avg. | **0.6153** | 0.7984 | **0.6574** | 0.4314 | 0.3678 | 0.3956 | 0.4220 | 0.6741 | 0.5453 |
| | FourierGNN | 1D | 0.6606 | 0.9310 | 0.7011 | 0.7420 | 0.5204 | 0.5268 | 0.6103 | 0.7723 | 0.6831 |
| | | 3D | 0.6044 | 0.8487 | 0.6062 | 0.6820 | 0.4898 | 0.5028 | 0.5337 | 0.7566 | 0.6281 |
| | | 5D | 0.5545 | 0.8082 | 0.6223 | 0.6375 | 0.4579 | 0.3712 | 0.5406 | 0.7085 | 0.5876 |
| | | 7D | 0.5737 | 0.7776 | 0.6044 | 0.6699 | 0.4793 | 0.4239 | 0.5228 | 0.6609 | 0.5891 |
| | | Avg. | 0.5983 | **0.8414** | 0.6335 | **0.6828** | 0.4869 | 0.4562 | **0.5519** | **0.7246** | 0.6219 |
| | StemGNN | 1D | 0.6429 | 0.9027 | 0.6334 | 0.6619 | 0.4429 | 0.4493 | 0.5562 | 0.7738 | 0.6329 |
| | | 3D | 0.5120 | 0.8540 | 0.6633 | 0.6193 | 0.4460 | 0.3921 | 0.3833 | 0.6762 | 0.5683 |
| | | 5D | 0.4908 | 0.7262 | 0.6228 | 0.5595 | 0.3974 | 0.3235 | 0.3157 | 0.6449 | 0.5101 |
| | | 7D | 0.4009 | 0.6838 | 0.4941 | 0.5233 | 0.3831 | 0.2672 | 0.2345 | 0.5653 | 0.4440 |
| | | Avg. | 0.5116 | 0.7917 | 0.6034 | 0.5910 | 0.4173 | 0.3580 | 0.3724 | 0.6650 | 0.5388 |
| **LLM** | GPT4TS | 1D | 0.6635 | 0.9232 | 0.6942 | 0.7230 | 0.4875 | 0.6188 | 0.5335 | 0.7614 | 0.6757 |
| | | 3D | 0.6057 | 0.8277 | 0.6177 | 0.6298 | 0.4394 | 0.4481 | 0.4085 | 0.6951 | 0.5840 |
| | | 5D | 0.5684 | 0.7776 | 0.5751 | 0.5932 | 0.4164 | 0.3858 | 0.3212 | 0.6346 | 0.5340 |
| | | 7D | 0.5243 | 0.7301 | 0.5333 | 0.5432 | 0.3953 | 0.3318 | 0.2623 | 0.5767 | 0.4871 |
| | | Avg. | 0.5905 | 0.8147 | 0.6051 | 0.6223 | 0.4347 | 0.4462 | 0.3814 | 0.6670 | 0.5702 |
| | AutoTimes | 1D | 0.6599 | 0.9250 | 0.6921 | 0.7510 | 0.5256 | 0.6094 | 0.5451 | 0.8114 | 0.6899 |
| | | 3D | 0.6089 | 0.8376 | 0.6380 | 0.6568 | 0.4840 | 0.4850 | 0.4504 | 0.7400 | 0.6126 |
| | | 5D | 0.5746 | 0.7828 | 0.5837 | 0.6040 | 0.4538 | 0.4243 | 0.3857 | 0.6793 | 0.5610 |
| | | 7D | 0.5422 | 0.7350 | 0.5560 | 0.5717 | 0.4415 | 0.3649 | 0.3277 | 0.6459 | 0.5231 |
| | | Avg. | 0.5964 | 0.8201 | 0.6175 | 0.6459 | 0.4762 | 0.4709 | 0.4272 | 0.7191 | 0.5967 |

Table 34: Benchmark(SEDI5) on part 3(Fort Myers area) stations(MLP,CNN,Transformer)

| | Method | T | S0 | S1 | S2 | S3 | S4 | S5 | S6 | S7 | Avg. |
|---|---|---|---|---|---|---|---|---|---|---|---|
| **MLP** | MLP | 1D | 0.4894 | 0.8078 | 0.7149 | 0.4916 | 0.3294 | 0.5618 | 0.3728 | 0.6667 | 0.5543 |
| | | 3D | 0.4773 | 0.7190 | 0.6372 | 0.5178 | 0.3379 | 0.4137 | 0.3164 | 0.6467 | 0.5083 |
| | | 5D | 0.4737 | 0.6523 | 0.5924 | 0.4466 | 0.3030 | 0.4697 | 0.3772 | 0.6306 | 0.4932 |
| | | 7D | 0.4645 | 0.6408 | 0.5297 | 0.4251 | 0.3127 | 0.3881 | 0.2769 | 0.5457 | 0.4479 |
| | | Avg. | 0.4762 | 0.7050 | 0.6186 | 0.4703 | 0.3208 | **0.4583** | 0.3358 | **0.6224** | **0.5009** |
| | TSMixer | 1D | 0.4885 | 0.7318 | 0.7059 | 0.5263 | 0.3722 | 0.5500 | 0.3410 | 0.6631 | 0.5474 |
| | | 3D | 0.4713 | 0.6731 | 0.6384 | 0.4570 | 0.3260 | 0.3770 | 0.2068 | 0.5727 | 0.4653 |
| | | 5D | 0.4577 | 0.6479 | 0.5783 | 0.4103 | 0.3053 | 0.2971 | 0.1542 | 0.5068 | 0.4197 |
| | | 7D | 0.4474 | 0.6274 | 0.5311 | 0.3795 | 0.2879 | 0.2336 | 0.1456 | 0.4626 | 0.3894 |
| | | Avg. | 0.4662 | 0.6700 | 0.6134 | 0.4433 | 0.3229 | 0.3644 | 0.2119 | 0.5513 | 0.4554 |
| | NLinear | 1D | 0.4908 | 0.8016 | 0.7131 | 0.5444 | 0.3801 | 0.5330 | 0.3531 | 0.6808 | 0.5621 |
| | | 3D | 0.4729 | 0.6957 | 0.6459 | 0.4658 | 0.3114 | 0.3720 | 0.2127 | 0.5928 | 0.4711 |
| | | 5D | 0.4581 | 0.6564 | 0.5865 | 0.4188 | 0.2919 | 0.2901 | 0.1535 | 0.5293 | 0.4231 |
| | | 7D | 0.4475 | 0.6307 | 0.5373 | 0.3836 | 0.2755 | 0.2271 | 0.1539 | 0.4864 | 0.3927 |
| | | Avg. | 0.4673 | 0.6961 | 0.6207 | 0.4532 | 0.3147 | 0.3555 | 0.2183 | 0.5724 | 0.4623 |
| **CNN** | TCN | 1D | 0.4885 | 0.5418 | 0.6754 | 0.4250 | 0.0903 | 0.5215 | 0.1972 | 0.1387 | 0.3848 |
| | | 3D | 0.2287 | 0.4575 | 0.4992 | 0.3864 | 0.0903 | 0.2572 | 0.2046 | 0.1727 | 0.2871 |
| | | 5D | 0.2222 | 0.6698 | 0.3806 | 0.3737 | 0.1045 | 0.2205 | 0.1504 | 0.1179 | 0.2799 |
| | | 7D | 0.2372 | 0.4134 | 0.4001 | 0.3923 | 0.1494 | 0.2020 | 0.1318 | 0.1100 | 0.2545 |
| | | Avg. | 0.2941 | 0.5206 | 0.4888 | 0.3943 | 0.1086 | 0.3003 | 0.1710 | 0.1348 | 0.3016 |
| | ModernTCN | 1D | 0.4916 | 0.8000 | 0.7168 | 0.5206 | 0.3806 | 0.5178 | 0.3296 | 0.6500 | 0.5509 |
| | | 3D | 0.4736 | 0.6654 | 0.6446 | 0.4359 | 0.3385 | 0.2951 | 0.1874 | 0.5540 | 0.4493 |
| | | 5D | 0.4546 | 0.6087 | 0.5675 | 0.3828 | 0.3163 | 0.2291 | 0.1094 | 0.4870 | 0.3944 |
| | | 7D | 0.4437 | 0.5747 | 0.5238 | 0.3442 | 0.2964 | 0.1796 | 0.0746 | 0.4491 | 0.3607 |
| | | Avg. | 0.4659 | 0.6622 | 0.6132 | 0.4209 | 0.3329 | 0.3054 | 0.1752 | 0.5350 | 0.4388 |
| | TimesNet | 1D | 0.4887 | 0.7914 | 0.6931 | 0.5047 | 0.3416 | 0.4730 | 0.2862 | 0.5783 | 0.5196 |
| | | 3D | 0.4738 | 0.6845 | 0.6056 | 0.4265 | 0.3145 | 0.3008 | 0.1060 | 0.4963 | 0.4260 |
| | | 5D | 0.4572 | 0.6307 | 0.5365 | 0.3854 | 0.2857 | 0.2354 | 0.0782 | 0.4423 | 0.3814 |
| | | 7D | 0.4477 | 0.6095 | 0.4801 | 0.3425 | 0.2637 | 0.1745 | 0.0602 | 0.3952 | 0.3467 |
| | | Avg. | 0.4669 | 0.6790 | 0.5788 | 0.4148 | 0.3013 | 0.2959 | 0.1327 | 0.4780 | 0.4184 |
| **Transformer** | iTransformer | 1D | 0.4861 | 0.7835 | 0.7049 | 0.5526 | 0.3745 | 0.5381 | 0.3514 | 0.6485 | 0.5549 |
| | | 3D | 0.4722 | 0.6740 | 0.6211 | 0.4677 | 0.3346 | 0.3229 | 0.2136 | 0.5557 | 0.4577 |
| | | 5D | 0.4566 | 0.6520 | 0.5766 | 0.4145 | 0.3157 | 0.2654 | 0.1486 | 0.5018 | 0.4164 |
| | | 7D | 0.4463 | 0.6171 | 0.5237 | 0.3832 | 0.2963 | 0.2088 | 0.1013 | 0.4623 | 0.3799 |
| | | Avg. | 0.4653 | 0.6817 | 0.6066 | 0.4545 | 0.3303 | 0.3338 | 0.2037 | 0.5421 | 0.4522 |
| | PatchTST | 1D | 0.4924 | 0.8199 | 0.7104 | 0.5501 | 0.3796 | 0.4983 | 0.3188 | 0.6643 | 0.5542 |
| | | 3D | 0.4747 | 0.6933 | 0.6454 | 0.4688 | 0.3416 | 0.3528 | 0.1760 | 0.5698 | 0.4653 |
| | | 5D | 0.4567 | 0.6503 | 0.5773 | 0.4337 | 0.3168 | 0.2647 | 0.1300 | 0.5138 | 0.4179 |
| | | 7D | 0.4544 | 0.6030 | 0.5288 | 0.3971 | 0.2896 | 0.2039 | 0.1030 | 0.4685 | 0.3810 |
| | | Avg. | 0.4695 | 0.6916 | 0.6154 | 0.4624 | 0.3319 | 0.3299 | 0.1820 | 0.5541 | 0.4546 |

Table 35: Benchmark(SEDI5) on part 3(Fort Myers area) stations(RNN,GNN,LLM)

| | Method | T | S0 | S1 | S2 | S3 | S4 | S5 | S6 | S7 | Avg. |
|---|---|---|---|---|---|---|---|---|---|---|---|
| **RNN** | LSTM | 1D | 0.4820 | 0.6970 | 0.6832 | 0.5265 | 0.3526 | 0.4093 | 0.2297 | 0.5224 | 0.4878 |
| | | 3D | 0.4696 | 0.6260 | 0.5874 | 0.4875 | 0.2963 | 0.1749 | 0.2572 | 0.4429 | 0.4177 |
| | | 5D | 0.4635 | 0.6474 | 0.3289 | 0.4312 | 0.2716 | 0.1569 | 0.0442 | 0.4783 | 0.3528 |
| | | 7D | 0.4340 | 0.6733 | 0.3075 | 0.3608 | 0.2654 | 0.1364 | 0.0542 | 0.4455 | 0.3347 |
| | | Avg. | 0.4623 | 0.6609 | 0.4768 | 0.4515 | 0.2965 | 0.2194 | 0.1463 | 0.4723 | 0.3982 |
| | DeepAR | 1D | 0.4587 | 0.7348 | 0.5771 | 0.5112 | 0.3805 | 0.3081 | 0.1743 | 0.4682 | 0.4516 |
| | | 3D | 0.4104 | 0.6217 | 0.3898 | 0.4468 | 0.3524 | 0.1630 | 0.1703 | 0.4901 | 0.3806 |
| | | 5D | 0.3463 | 0.6626 | 0.3252 | 0.4434 | 0.2483 | 0.1350 | 0.0699 | 0.3937 | 0.3280 |
| | | 7D | 0.4636 | 0.6772 | 0.2954 | 0.4170 | 0.2511 | 0.1049 | 0.1605 | 0.4080 | 0.3472 |
| | | Avg. | 0.4197 | 0.6741 | 0.3969 | 0.4546 | 0.3081 | 0.1778 | 0.1437 | 0.4400 | 0.3769 |
| | DilatedRNN | 1D | 0.4427 | 0.7950 | 0.7256 | 0.6408 | 0.2791 | 0.4482 | 0.2522 | 0.5895 | 0.5216 |
| | | 3D | 0.4861 | 0.6492 | 0.6471 | 0.3783 | 0.3198 | 0.2445 | 0.1764 | 0.5450 | 0.4308 |
| | | 5D | 0.3852 | 0.6652 | 0.5548 | 0.4024 | 0.3185 | 0.1632 | 0.1136 | 0.5060 | 0.3886 |
| | | 7D | 0.4685 | 0.5954 | 0.5216 | 0.3734 | 0.2855 | 0.1483 | 0.1444 | 0.4271 | 0.3705 |
| | | Avg. | 0.4456 | 0.6762 | 0.6123 | 0.4487 | 0.3007 | 0.2510 | 0.1716 | 0.5169 | 0.4279 |
| **GNN** | GCN | 1D | 0.4893 | 0.7551 | 0.7258 | 0.3366 | 0.3070 | 0.3908 | 0.2399 | 0.5583 | 0.4754 |
| | | 3D | 0.4817 | 0.7295 | 0.6715 | 0.2813 | 0.2968 | 0.2954 | 0.2091 | 0.5118 | 0.4346 |
| | | 5D | 0.4785 | 0.7154 | 0.5946 | 0.2677 | 0.2595 | 0.2635 | 0.0498 | 0.4793 | 0.3885 |
| | | 7D | 0.4697 | 0.6812 | 0.5678 | 0.2610 | 0.2722 | 0.2174 | 0.1484 | 0.5102 | 0.3910 |
| | | Avg. | **0.4798** | **0.7203** | **0.6399** | 0.2867 | 0.2839 | 0.2918 | 0.1618 | 0.5149 | 0.4224 |
| | FourierGNN | 1D | 0.4873 | 0.7843 | 0.7085 | 0.5126 | 0.3541 | 0.4430 | 0.4340 | 0.6063 | 0.5412 |
| | | 3D | 0.4800 | 0.7305 | 0.6125 | 0.4674 | 0.3419 | 0.3558 | 0.3325 | 0.5709 | 0.4864 |
| | | 5D | 0.4773 | 0.6881 | 0.5911 | 0.4507 | 0.3064 | 0.2782 | 0.3179 | 0.5243 | 0.4543 |
| | | 7D | 0.4705 | 0.6634 | 0.5482 | 0.4566 | 0.3343 | 0.3072 | 0.3234 | 0.4842 | 0.4485 |
| | | Avg. | 0.4788 | 0.7166 | 0.6151 | **0.4718** | **0.3342** | 0.3460 | **0.3519** | 0.5464 | 0.4826 |
| | StemGNN | 1D | 0.4950 | 0.7993 | 0.6503 | 0.4581 | 0.3070 | 0.3161 | 0.3138 | 0.5819 | 0.4902 |
| | | 3D | 0.4878 | 0.6894 | 0.6336 | 0.4333 | 0.2762 | 0.2694 | 0.1468 | 0.4612 | 0.4247 |
| | | 5D | 0.4762 | 0.6203 | 0.6074 | 0.4065 | 0.2529 | 0.2048 | 0.1256 | 0.4329 | 0.3908 |
| | | 7D | 0.2821 | 0.5487 | 0.4545 | 0.3398 | 0.2205 | 0.1658 | 0.0749 | 0.3691 | 0.3069 |
| | | Avg. | 0.4353 | 0.6644 | 0.5865 | 0.4094 | 0.2642 | 0.2390 | 0.1653 | 0.4613 | 0.4032 |
| **LLM** | GPT4TS | 1D | 0.4909 | 0.7803 | 0.6876 | 0.4807 | 0.3624 | 0.4659 | 0.2754 | 0.6054 | 0.5186 |
| | | 3D | 0.4731 | 0.6916 | 0.6056 | 0.4087 | 0.3166 | 0.3103 | 0.1481 | 0.4977 | 0.4315 |
| | | 5D | 0.4595 | 0.6523 | 0.5701 | 0.3712 | 0.2935 | 0.2494 | 0.0887 | 0.4326 | 0.3896 |
| | | 7D | 0.4498 | 0.6217 | 0.5153 | 0.3324 | 0.2688 | 0.2017 | 0.0738 | 0.3889 | 0.3566 |
| | | Avg. | 0.4683 | 0.6865 | 0.5947 | 0.3983 | 0.3103 | 0.3068 | 0.1465 | 0.4811 | 0.4241 |
| | AutoTimes | 1D | 0.4910 | 0.7669 | 0.6932 | 0.5324 | 0.3777 | 0.5069 | 0.3207 | 0.6462 | 0.5419 |
| | | 3D | 0.4693 | 0.6760 | 0.6316 | 0.4674 | 0.3322 | 0.3641 | 0.2025 | 0.5692 | 0.4640 |
| | | 5D | 0.4547 | 0.6498 | 0.5702 | 0.4229 | 0.3073 | 0.2926 | 0.1335 | 0.5071 | 0.4173 |
| | | 7D | 0.4421 | 0.6195 | 0.5349 | 0.3920 | 0.2933 | 0.2368 | 0.1115 | 0.4639 | 0.3868 |
| | | Avg. | 0.4643 | 0.6781 | 0.6075 | 0.4537 | 0.3277 | 0.3501 | 0.1920 | 0.5466 | 0.4525 |

Table 36: Benchmark(SEDI1) on part 3(Fort Myers area) stations(MLP,CNN,Transformer)

| | Method | T | S0 | S1 | S2 | S3 | S4 | S5 | S6 | S7 | Avg. |
|---|---|---|---|---|---|---|---|---|---|---|---|
| **MLP** | MLP | 1D | 0.4747 | 0.4387 | 0.5051 | 0.2137 | 0.1295 | 0.3272 | 0.0017 | 0.2836 | 0.2968 |
| | | 3D | 0.4444 | 0.4520 | 0.2868 | 0.2392 | 0.1628 | 0.2860 | 0.0006 | 0.3115 | 0.2729 |
| | | 5D | 0.4302 | 0.2950 | 0.1017 | 0.1766 | 0.1696 | 0.3612 | 0.0005 | 0.3690 | 0.2380 |
| | | 7D | 0.4116 | 0.2920 | 0.2236 | 0.1575 | 0.1613 | 0.2435 | 0.0004 | 0.2357 | 0.2157 |
| | | Avg. | **0.4402** | 0.3694 | 0.2793 | **0.1968** | 0.1558 | **0.3045** | **0.0008** | **0.3000** | **0.2558** |
| | TSMixer | 1D | 0.4450 | 0.5180 | 0.4305 | 0.2132 | 0.1929 | 0.2698 | 0.0000 | 0.3516 | 0.3026 |
| | | 3D | 0.3848 | 0.3874 | 0.2667 | 0.1432 | 0.1651 | 0.1587 | 0.0000 | 0.2378 | 0.2180 |
| | | 5D | 0.3283 | 0.3044 | 0.2146 | 0.1247 | 0.1417 | 0.1347 | 0.0000 | 0.1959 | 0.1805 |
| | | 7D | 0.2849 | 0.2479 | 0.1902 | 0.1069 | 0.1197 | 0.1177 | 0.0000 | 0.1482 | 0.1519 |
| | | Avg. | 0.3607 | 0.3644 | 0.2755 | 0.1470 | 0.1548 | 0.1702 | 0.0000 | 0.2334 | 0.2133 |
| | NLinear | 1D | 0.4657 | 0.5401 | 0.4711 | 0.2360 | 0.2096 | 0.3693 | 0.0017 | 0.3968 | 0.3363 |
| | | 3D | 0.4022 | 0.3957 | 0.2858 | 0.1465 | 0.1677 | 0.2077 | 0.0006 | 0.2691 | 0.2344 |
| | | 5D | 0.3454 | 0.3299 | 0.2238 | 0.1311 | 0.1556 | 0.1897 | 0.0003 | 0.2083 | 0.1980 |
| | | 7D | 0.3003 | 0.2835 | 0.1990 | 0.1112 | 0.1377 | 0.1638 | 0.0002 | 0.1777 | 0.1717 |
| | | Avg. | 0.3784 | **0.3873** | 0.2949 | 0.1562 | 0.1676 | 0.2326 | 0.0007 | 0.2630 | 0.2351 |
| **CNN** | TCN | 1D | 0.3817 | 0.3979 | 0.5681 | 0.1620 | 0.1078 | 0.2254 | 0.0001 | 0.2565 | 0.2624 |
| | | 3D | 0.3269 | 0.2261 | 0.2620 | 0.1106 | 0.1482 | 0.1507 | 0.0000 | 0.1577 | 0.1728 |
| | | 5D | 0.3281 | 0.3077 | 0.2576 | 0.1217 | 0.1144 | 0.1199 | 0.0000 | 0.1687 | 0.1773 |
| | | 7D | 0.2895 | 0.2433 | 0.2785 | 0.0881 | 0.0582 | 0.1289 | 0.0000 | 0.1664 | 0.1566 |
| | | Avg. | 0.3315 | 0.2938 | **0.3415** | 0.1206 | 0.1072 | 0.1562 | 0.0000 | 0.1873 | 0.1923 |
| | ModernTCN | 1D | 0.4700 | 0.3796 | 0.4606 | 0.1984 | 0.1834 | 0.2864 | 0.0003 | 0.3521 | 0.2913 |
| | | 3D | 0.4134 | 0.2972 | 0.3154 | 0.1244 | 0.1610 | 0.1089 | 0.0000 | 0.2540 | 0.2093 |
| | | 5D | 0.3802 | 0.2393 | 0.2666 | 0.0989 | 0.1319 | 0.0585 | 0.0000 | 0.1811 | 0.1696 |
| | | 7D | 0.3497 | 0.2258 | 0.2511 | 0.0857 | 0.1173 | 0.0431 | 0.0000 | 0.1418 | 0.1518 |
| | | Avg. | 0.4033 | 0.2855 | 0.3234 | 0.1269 | 0.1484 | 0.1242 | 0.0001 | 0.2322 | 0.2055 |
| | TimesNet | 1D | 0.4591 | 0.3666 | 0.4525 | 0.1548 | 0.1686 | 0.1891 | 0.0000 | 0.2756 | 0.2583 |
| | | 3D | 0.4283 | 0.2930 | 0.2820 | 0.1182 | 0.1470 | 0.0856 | 0.0000 | 0.1680 | 0.1902 |
| | | 5D | 0.3777 | 0.2554 | 0.2337 | 0.1004 | 0.1092 | 0.0487 | 0.0000 | 0.1362 | 0.1577 |
| | | 7D | 0.3632 | 0.2254 | 0.1776 | 0.0801 | 0.0995 | 0.0378 | 0.0000 | 0.0923 | 0.1345 |
| | | Avg. | 0.4071 | 0.2851 | 0.2865 | 0.1134 | 0.1311 | 0.0903 | 0.0000 | 0.1680 | 0.1852 |
| **Transformer** | iTransformer | 1D | 0.4584 | 0.4340 | 0.4950 | 0.2389 | 0.1775 | 0.3983 | 0.0005 | 0.3726 | 0.3219 |
| | | 3D | 0.4305 | 0.2699 | 0.2939 | 0.1542 | 0.1563 | 0.1485 | 0.0001 | 0.2423 | 0.2120 |
| | | 5D | 0.3914 | 0.2541 | 0.2244 | 0.1336 | 0.1363 | 0.1145 | 0.0001 | 0.1856 | 0.1800 |
| | | 7D | 0.3506 | 0.2224 | 0.1918 | 0.1115 | 0.1161 | 0.0841 | 0.0000 | 0.1842 | 0.1576 |
| | | Avg. | 0.4077 | 0.2951 | 0.3013 | 0.1595 | 0.1465 | 0.1863 | 0.0002 | 0.2462 | 0.2179 |
| | PatchTST | 1D | 0.4734 | 0.4614 | 0.4548 | 0.2418 | 0.1918 | 0.3446 | 0.0003 | 0.3847 | 0.3191 |
| | | 3D | 0.4278 | 0.3069 | 0.2981 | 0.1757 | 0.1645 | 0.1653 | 0.0001 | 0.2403 | 0.2223 |
| | | 5D | 0.3990 | 0.2060 | 0.2292 | 0.1506 | 0.1415 | 0.0919 | 0.0000 | 0.1697 | 0.1735 |
| | | 7D | 0.3603 | 0.1898 | 0.1968 | 0.1058 | 0.1200 | 0.0701 | 0.0000 | 0.1344 | 0.1472 |
| | | Avg. | 0.4151 | 0.2910 | 0.2947 | 0.1685 | 0.1544 | 0.1680 | 0.0001 | 0.2323 | 0.2155 |

Table 37: Benchmark(SEDI1) on part 3(Fort Myers area) stations(RNN,GNN,LLM)

| | Method | T | S0 | S1 | S2 | S3 | S4 | S5 | S6 | S7 | Avg. |
|---|---|---|---|---|---|---|---|---|---|---|---|
| RNN | LSTM | 1D | 0.2500 | 0.3899 | 0.0246 | 0.1762 | 0.1810 | 0.1240 | 0.0000 | 0.2035 | 0.1686 |
| | | 3D | 0.2375 | 0.0599 | 0.0035 | 0.1100 | 0.1246 | 0.0055 | 0.0000 | 0.1008 | 0.0802 |
| | | 5D | 0.0000 | 0.0617 | 0.0000 | 0.0920 | 0.0927 | 0.0081 | 0.0000 | 0.0967 | 0.0439 |
| | | 7D | 0.2466 | 0.0486 | 0.0230 | 0.0938 | 0.0791 | 0.0256 | 0.0000 | 0.0555 | 0.0715 |
| | | Avg. | 0.1835 | 0.1400 | 0.0128 | 0.1180 | 0.1194 | 0.0408 | 0.0000 | 0.1141 | 0.0911 |
| | DeepAR | 1D | 0.1603 | 0.0000 | 0.0998 | 0.1648 | 0.1890 | 0.1125 | 0.0000 | 0.1389 | 0.1082 |
| | | 3D | 0.2332 | 0.1744 | 0.0000 | 0.0798 | 0.0822 | 0.0015 | 0.0000 | 0.0887 | 0.0825 |
| | | 5D | 0.2488 | 0.0519 | 0.0000 | 0.1098 | 0.0928 | 0.0190 | 0.0000 | 0.0000 | 0.0653 |
| | | 7D | 0.0000 | 0.0091 | 0.0056 | 0.0657 | 0.0243 | 0.0088 | 0.0000 | 0.0961 | 0.0262 |
| | | Avg. | 0.1606 | 0.0589 | 0.0263 | 0.1050 | 0.0971 | 0.0355 | 0.0000 | 0.0809 | 0.0705 |
| | DilatedRNN | 1D | 0.1049 | 0.3690 | 0.3560 | 0.3224 | 0.0969 | 0.3043 | 0.0003 | 0.2794 | 0.2291 |
| | | 3D | 0.2490 | 0.2811 | 0.0161 | 0.1159 | 0.1277 | 0.0744 | 0.0001 | 0.1527 | 0.1271 |
| | | 5D | 0.0949 | 0.2827 | 0.2367 | 0.1348 | 0.1246 | 0.0415 | 0.0000 | 0.2265 | 0.1427 |
| | | 7D | 0.2413 | 0.1618 | 0.2236 | 0.1230 | 0.1192 | 0.0497 | 0.0000 | 0.0805 | 0.1249 |
| | | Avg. | 0.1725 | 0.2736 | 0.2081 | 0.1740 | 0.1171 | 0.1175 | 0.0001 | 0.1848 | 0.1560 |
| GNN | GCN | 1D | 0.3817 | 0.3979 | 0.5681 | 0.1620 | 0.1078 | 0.2254 | 0.0001 | 0.2565 | 0.2624 |
| | | 3D | 0.3269 | 0.2261 | 0.2620 | 0.1106 | 0.1482 | 0.1507 | 0.0000 | 0.1577 | 0.1728 |
| | | 5D | 0.3281 | 0.3077 | 0.2576 | 0.1217 | 0.1144 | 0.1199 | 0.0000 | 0.1687 | 0.1773 |
| | | 7D | 0.2895 | 0.2433 | 0.2785 | 0.0881 | 0.0582 | 0.1289 | 0.0000 | 0.1664 | 0.1566 |
| | | Avg. | 0.3315 | 0.2938 | **0.3415** | 0.1206 | 0.1072 | 0.1562 | 0.0000 | 0.1873 | 0.1923 |
| | FourierGNN | 1D | 0.4363 | 0.3351 | 0.3967 | 0.2117 | 0.1928 | 0.2551 | 0.0018 | 0.2885 | 0.2647 |
| | | 3D | 0.4293 | 0.2745 | 0.2772 | 0.1733 | 0.2145 | 0.1698 | 0.0000 | 0.2810 | 0.2274 |
| | | 5D | 0.4124 | 0.3231 | 0.2661 | 0.1731 | 0.1441 | 0.2090 | 0.0000 | 0.2253 | 0.2191 |
| | | 7D | 0.3564 | 0.2544 | 0.2314 | 0.1561 | 0.2023 | 0.1808 | 0.0000 | 0.1790 | 0.1951 |
| | | Avg. | 0.4086 | 0.2968 | 0.2928 | 0.1785 | **0.1884** | 0.2037 | 0.0005 | 0.2434 | 0.2266 |
| | StemGNN | 1D | 0.1616 | 0.4190 | 0.1862 | 0.1781 | 0.1094 | 0.1578 | 0.0002 | 0.2462 | 0.1823 |
| | | 3D | 0.2372 | 0.2514 | 0.1446 | 0.1443 | 0.0443 | 0.1068 | 0.0000 | 0.1309 | 0.1324 |
| | | 5D | 0.0956 | 0.1130 | 0.0178 | 0.1026 | 0.0790 | 0.0844 | 0.0000 | 0.1162 | 0.0761 |
| | | 7D | 0.0360 | 0.0911 | 0.0384 | 0.1245 | 0.0247 | 0.1175 | 0.0000 | 0.0726 | 0.0631 |
| | | Avg. | 0.1326 | 0.2186 | 0.0968 | 0.1374 | 0.0644 | 0.1167 | 0.0001 | 0.1415 | 0.1135 |
| LLM | GPT4TS | 1D | 0.4586 | 0.4502 | 0.4421 | 0.1424 | 0.1876 | 0.2122 | 0.0000 | 0.2792 | 0.2715 |
| | | 3D | 0.4088 | 0.3268 | 0.2713 | 0.1127 | 0.1530 | 0.0738 | 0.0000 | 0.1810 | 0.1909 |
| | | 5D | 0.3916 | 0.2650 | 0.2210 | 0.1011 | 0.1201 | 0.0412 | 0.0000 | 0.1263 | 0.1583 |
| | | 7D | 0.3606 | 0.2190 | 0.1939 | 0.0847 | 0.1091 | 0.0466 | 0.0000 | 0.1000 | 0.1392 |
| | | Avg. | 0.4049 | 0.3153 | 0.2821 | 0.1102 | 0.1424 | 0.0935 | 0.0000 | 0.1716 | 0.1900 |
| | AutoTimes | 1D | 0.4710 | 0.3713 | 0.4458 | 0.2398 | 0.1955 | 0.3082 | 0.0000 | 0.3205 | 0.2940 |
| | | 3D | 0.4297 | 0.2009 | 0.2890 | 0.1551 | 0.1653 | 0.1697 | 0.0000 | 0.2165 | 0.2033 |
| | | 5D | 0.3880 | 0.1895 | 0.2212 | 0.1363 | 0.1375 | 0.1271 | 0.0000 | 0.1561 | 0.1695 |
| | | 7D | 0.3494 | 0.1505 | 0.1984 | 0.1012 | 0.1175 | 0.0965 | 0.0000 | 0.1250 | 0.1423 |
| | | Avg. | 0.4095 | 0.2280 | 0.2886 | 0.1581 | 0.1539 | 0.1753 | 0.0000 | 0.2045 | 0.2023 |

Table 38: Benchmark(MAE) on all stations

| Method | T | S0 | S1 | S2 | S3 | S4 | S5 | S6 | S7 | Avg. |
|---|---|---|---|---|---|---|---|---|---|---|
| MLP | 1D | 0.1324 | 0.1424 | 0.1441 | 0.1043 | 0.1098 | 0.1192 | 0.1423 | 0.1310 | 0.1282 |
| | 3D | 0.2158 | 0.2352 | 0.2471 | 0.1838 | 0.1875 | 0.1903 | 0.2516 | 0.2082 | 0.2150 |
| | 5D | 0.2810 | 0.2902 | 0.3178 | 0.2277 | 0.2437 | 0.2354 | 0.3228 | 0.2565 | 0.2719 |
| | 7D | 0.3329 | 0.3374 | 0.3666 | 0.2661 | 0.2837 | 0.2719 | 0.3793 | 0.2915 | 0.3162 |
| | Avg. | **0.2405** | **0.2513** | **0.2689** | **0.1955** | **0.2062** | **0.2042** | **0.2740** | **0.2218** | **0.2328** |
| TCN | 1D | 0.3079 | 0.2613 | 0.2634 | 0.2414 | 0.2079 | 0.1885 | 0.2957 | 0.2740 | 0.2550 |
| | 3D | 0.4332 | 0.3215 | 0.3490 | 0.3061 | 0.2900 | 0.2554 | 0.3944 | 0.3465 | 0.3370 |
| | 5D | 0.4324 | 0.3788 | 0.4040 | 0.3563 | 0.3457 | 0.3070 | 0.4547 | 0.3992 | 0.3848 |
| | 7D | 0.4842 | 0.3951 | 0.4429 | 0.3915 | 0.3803 | 0.3397 | 0.4979 | 0.4242 | 0.4195 |
| | Avg. | 0.4144 | 0.3392 | 0.3648 | 0.3238 | 0.3060 | 0.2727 | 0.4107 | 0.3610 | 0.3491 |
| LSTM | 1D | 0.1867 | 0.2613 | 0.2736 | 0.1595 | 0.2496 | 0.1697 | 0.3087 | 0.1836 | 0.2241 |
| | 3D | 0.2862 | 0.3516 | 0.3751 | 0.2391 | 0.3291 | 0.2446 | 0.4267 | 0.2704 | 0.3153 |
| | 5D | 0.3475 | 0.4012 | 0.4475 | 0.2892 | 0.3753 | 0.2944 | 0.4867 | 0.3172 | 0.3699 |
| | 7D | 0.4174 | 0.4312 | 0.5125 | 0.3261 | 0.4193 | 0.3261 | 0.5173 | 0.3429 | 0.4116 |
| | Avg. | 0.3094 | 0.3613 | 0.4022 | 0.2535 | 0.3433 | 0.2587 | 0.4348 | 0.2785 | 0.3302 |
| GCN | 1D | 0.1850 | 0.2317 | 0.2365 | 0.2061 | 0.1953 | 0.1681 | 0.1944 | 0.2152 | 0.2040 |
| | 3D | 0.2668 | 0.3177 | 0.3426 | 0.2735 | 0.2782 | 0.2385 | 0.3119 | 0.2914 | 0.2901 |
| | 5D | 0.3329 | 0.3673 | 0.4137 | 0.3184 | 0.3256 | 0.2796 | 0.3977 | 0.3336 | 0.3461 |
| | 7D | 0.3863 | 0.3860 | 0.4624 | 0.3484 | 0.3646 | 0.3102 | 0.4265 | 0.3620 | 0.3808 |
| | Avg. | 0.2928 | 0.3257 | 0.3638 | 0.2866 | 0.2909 | 0.2491 | 0.3326 | 0.3005 | 0.3053 |

Table 39: Benchmark(MSE) on all stations

| Method | T | S0 | S1 | S2 | S3 | S4 | S5 | S6 | S7 | Avg. |
|---|---|---|---|---|---|---|---|---|---|---|
| MLP | 1D | 0.1144 | 0.1360 | 0.1284 | 0.0684 | 0.1124 | 0.0747 | 0.1674 | 0.1256 | 0.1159 |
| | 3D | 0.2226 | 0.3080 | 0.2875 | 0.1478 | 0.3256 | 0.1424 | 0.5486 | 0.2405 | 0.2779 |
| | 5D | 0.3117 | 0.4687 | 0.4318 | 0.2081 | 0.6883 | 0.1928 | 1.2099 | 0.3317 | 0.4804 |
| | 7D | 0.3808 | 0.6967 | 0.6094 | 0.2648 | 1.0917 | 0.2369 | 1.8233 | 0.3899 | 0.6867 |
| | Avg. | **0.2574** | **0.4024** | **0.3643** | **0.1723** | **0.5545** | **0.1617** | **0.9373** | **0.2719** | **0.3902** |
| TCN | 1D | 0.3964 | 0.5800 | 0.4714 | 0.2575 | 0.3783 | 0.2180 | 1.3093 | 0.5552 | 0.5208 |
| | 3D | 0.6306 | 0.6849 | 0.6180 | 0.3337 | 0.5102 | 0.2900 | 1.5333 | 0.6517 | 0.6566 |
| | 5D | 0.5709 | 0.7997 | 0.7216 | 0.3930 | 0.6323 | 0.3534 | 1.6885 | 0.7298 | 0.7362 |
| | 7D | 0.6880 | 0.8404 | 0.7966 | 0.4639 | 0.7027 | 0.3941 | 1.8442 | 0.7434 | 0.8092 |
| | Avg. | 0.5715 | 0.7263 | 0.6519 | 0.3620 | 0.5559 | 0.3139 | 1.5938 | 0.6700 | 0.6807 |
| LSTM | 1D | 0.2550 | 1.4732 | 1.0367 | 0.1620 | 5.2575 | 0.1497 | 6.8509 | 0.4084 | 1.9492 |
| | 3D | 0.3871 | 1.3765 | 1.0735 | 0.2642 | 5.7927 | 0.2476 | 8.5420 | 0.5052 | 2.2736 |
| | 5D | 0.4911 | 1.5590 | 1.4324 | 0.3373 | 5.8335 | 0.3032 | 8.3886 | 0.5652 | 2.3638 |
| | 7D | 0.6198 | 1.6812 | 1.8663 | 0.3982 | 6.0814 | 0.3494 | 8.5648 | 0.6176 | 2.5224 |
| | Avg. | 0.4383 | 1.5225 | 1.3522 | 0.2904 | 5.7413 | 0.2625 | 8.0866 | 0.5241 | 2.2772 |
| GCN | 1D | 0.1994 | 0.2886 | 0.7728 | 0.3590 | 0.2534 | 0.1165 | 0.2171 | 6.2444 | 1.0564 |
| | 3D | 0.3102 | 0.4645 | 1.0667 | 0.4061 | 0.5848 | 0.1883 | 0.9433 | 3.7576 | 0.9652 |
| | 5D | 0.4273 | 0.5908 | 1.3546 | 0.4490 | 0.8592 | 0.2408 | 2.4139 | 2.7351 | 1.1338 |
| | 7D | 0.5166 | 0.5968 | 1.5264 | 0.4728 | 1.3609 | 0.2752 | 1.8411 | 2.3212 | 1.1139 |
| | Avg. | 0.3634 | 0.4852 | 1.1801 | 0.4217 | 0.7646 | 0.2052 | 1.3539 | 3.7646 | 1.0673 |

Table 40: Benchmark(SEDI10) on all stations

| Method | T | S0 | S1 | S2 | S3 | S4 | S5 | S6 | S7 | Avg. |
|---|---|---|---|---|---|---|---|---|---|---|
| MLP | 1D | 0.7865 | 0.8159 | 0.8152 | 0.7883 | 0.7313 | 0.7192 | 0.7116 | 0.7581 | 0.7658 |
| | 3D | 0.7108 | 0.7527 | 0.7714 | 0.7639 | 0.6709 | 0.5924 | 0.6336 | 0.6414 | 0.6921 |
| | 5D | 0.6823 | 0.6972 | 0.7482 | 0.7092 | 0.6185 | 0.5809 | 0.5830 | 0.6343 | 0.6567 |
| | 7D | 0.6455 | 0.6726 | 0.7134 | 0.6753 | 0.6142 | 0.5620 | 0.5496 | 0.5815 | 0.6268 |
| | Avg. | **0.7063** | **0.7346** | **0.7621** | **0.7342** | **0.6587** | **0.6136** | **0.6195** | **0.6538** | **0.6853** |
| TCN | 1D | 0.6211 | 0.7314 | 0.7264 | 0.6545 | 0.5807 | 0.6232 | 0.5594 | 0.4628 | 0.6199 |
| | 3D | 0.4990 | 0.6284 | 0.6505 | 0.5697 | 0.4678 | 0.5020 | 0.4852 | 0.3738 | 0.5221 |
| | 5D | 0.4997 | 0.6025 | 0.6164 | 0.5319 | 0.4059 | 0.4549 | 0.4289 | 0.3187 | 0.4824 |
| | 7D | 0.4366 | 0.5680 | 0.5829 | 0.4747 | 0.3637 | 0.4181 | 0.3906 | 0.3059 | 0.4426 |
| | Avg. | 0.5141 | 0.6326 | 0.6440 | 0.5577 | 0.4545 | 0.4996 | 0.4660 | 0.3653 | 0.5167 |
| LSTM | 1D | 0.7478 | 0.7266 | 0.7410 | 0.7615 | 0.6781 | 0.6422 | 0.6255 | 0.6716 | 0.6993 |
| | 3D | 0.6290 | 0.6424 | 0.6606 | 0.6671 | 0.5615 | 0.5070 | 0.5091 | 0.5369 | 0.5892 |
| | 5D | 0.5632 | 0.5683 | 0.6139 | 0.6053 | 0.5035 | 0.4434 | 0.4307 | 0.4694 | 0.5247 |
| | 7D | 0.5249 | 0.5386 | 0.5845 | 0.5758 | 0.4605 | 0.3974 | 0.3886 | 0.4256 | 0.4870 |
| | Avg. | 0.6162 | 0.6190 | 0.6500 | 0.6524 | 0.5509 | 0.4975 | 0.4885 | 0.5259 | 0.5751 |
| GCN | 1D | 0.7461 | 0.7112 | 0.7543 | 0.7031 | 0.6442 | 0.6466 | 0.6565 | 0.6686 | 0.6913 |
| | 3D | 0.6725 | 0.6669 | 0.6912 | 0.6308 | 0.5846 | 0.5725 | 0.5800 | 0.6044 | 0.6254 |
| | 5D | 0.6331 | 0.6218 | 0.6668 | 0.5908 | 0.5515 | 0.5247 | 0.5314 | 0.5519 | 0.5840 |
| | 7D | 0.5989 | 0.5811 | 0.6531 | 0.5765 | 0.5183 | 0.4903 | 0.5025 | 0.5109 | 0.5539 |
| | Avg. | 0.6626 | 0.6452 | 0.6913 | 0.6253 | 0.5746 | 0.5585 | 0.5676 | 0.5840 | 0.6137 |

Table 41: Benchmark(SEDI5) on all stations

| Method | T | S0 | S1 | S2 | S3 | S4 | S5 | S6 | S7 | Avg. |
|---|---|---|---|---|---|---|---|---|---|---|
| MLP | 1D | 0.7136 | 0.7660 | 0.7338 | 0.7105 | 0.6322 | 0.6125 | 0.6060 | 0.6216 | 0.6745 |
| | 3D | 0.6163 | 0.7050 | 0.6930 | 0.6926 | 0.5791 | 0.4788 | 0.5176 | 0.5137 | 0.5995 |
| | 5D | 0.5987 | 0.6437 | 0.6679 | 0.6416 | 0.5329 | 0.4810 | 0.4710 | 0.5052 | 0.5678 |
| | 7D | 0.5702 | 0.6306 | 0.6431 | 0.6119 | 0.5267 | 0.4753 | 0.4457 | 0.4662 | 0.5462 |
| | Avg. | **0.6247** | **0.6863** | **0.6845** | **0.6642** | **0.5678** | **0.5119** | **0.5101** | **0.5267** | **0.5970** |
| TCN | 1D | 0.4196 | 0.6603 | 0.6184 | 0.5291 | 0.4095 | 0.4960 | 0.4138 | 0.3313 | 0.4848 |
| | 3D | 0.3450 | 0.5523 | 0.5307 | 0.4549 | 0.2944 | 0.3870 | 0.3365 | 0.2404 | 0.3926 |
| | 5D | 0.3236 | 0.5302 | 0.5064 | 0.4307 | 0.2343 | 0.3585 | 0.2869 | 0.2074 | 0.3597 |
| | 7D | 0.2981 | 0.4912 | 0.4486 | 0.3805 | 0.1940 | 0.3273 | 0.2666 | 0.1886 | 0.3243 |
| | Avg. | 0.3466 | 0.5585 | 0.5260 | 0.4488 | 0.2831 | 0.3922 | 0.3259 | 0.2419 | 0.3904 |
| LSTM | 1D | 0.6546 | 0.6720 | 0.6402 | 0.6737 | 0.5568 | 0.5009 | 0.4883 | 0.5126 | 0.5874 |
| | 3D | 0.5112 | 0.5662 | 0.5342 | 0.5557 | 0.4398 | 0.3529 | 0.3555 | 0.3751 | 0.4613 |
| | 5D | 0.4395 | 0.4974 | 0.4877 | 0.4993 | 0.3820 | 0.3008 | 0.2920 | 0.2929 | 0.3990 |
| | 7D | 0.3879 | 0.4701 | 0.4458 | 0.4642 | 0.3109 | 0.2694 | 0.2556 | 0.2664 | 0.3588 |
| | Avg. | 0.4983 | 0.5514 | 0.5270 | 0.5482 | 0.4224 | 0.3560 | 0.3479 | 0.3618 | 0.4516 |
| GCN | 1D | 0.6660 | 0.6406 | 0.6734 | 0.6192 | 0.5498 | 0.5445 | 0.5430 | 0.5278 | 0.5955 |
| | 3D | 0.5885 | 0.6013 | 0.6062 | 0.5427 | 0.4961 | 0.4673 | 0.4629 | 0.4690 | 0.5292 |
| | 5D | 0.5593 | 0.5528 | 0.5793 | 0.5059 | 0.4684 | 0.4301 | 0.4192 | 0.4291 | 0.4930 |
| | 7D | 0.5471 | 0.5177 | 0.5610 | 0.4934 | 0.4320 | 0.4081 | 0.3855 | 0.3945 | 0.4674 |
| | Avg. | 0.5902 | 0.5781 | 0.6050 | 0.5403 | 0.4866 | 0.4625 | 0.4526 | 0.4551 | 0.5213 |

Table 42: Benchmark(SEDI1) on all stations

| Method | T | S0 | S1 | S2 | S3 | S4 | S5 | S6 | S7 | Avg. |
|---|---|---|---|---|---|---|---|---|---|---|
| MLP | 1D | 0.5452 | 0.6313 | 0.5586 | 0.5222 | 0.3956 | 0.3222 | 0.3557 | 0.3511 | 0.4602 |
| | 3D | 0.4265 | 0.5790 | 0.5332 | 0.5326 | 0.3359 | 0.2318 | 0.2916 | 0.2890 | 0.4025 |
| | 5D | 0.4535 | 0.5114 | 0.5228 | 0.4968 | 0.3349 | 0.2440 | 0.2813 | 0.3036 | 0.3935 |
| | 7D | 0.4738 | 0.5377 | 0.5065 | 0.4550 | 0.3195 | 0.2545 | 0.2705 | 0.2740 | 0.3864 |
| | Avg. | **0.4748** | **0.5649** | **0.5303** | **0.5017** | **0.3465** | **0.2631** | **0.2998** | **0.3044** | **0.4107** |
| TCN | 1D | 0.1333 | 0.4546 | 0.4028 | 0.3001 | 0.0736 | 0.2119 | 0.1582 | 0.1412 | 0.2345 |
| | 3D | 0.1118 | 0.3684 | 0.3182 | 0.2284 | 0.0440 | 0.1646 | 0.1282 | 0.0833 | 0.1809 |
| | 5D | 0.1033 | 0.3531 | 0.3050 | 0.2328 | 0.0305 | 0.1499 | 0.1147 | 0.0757 | 0.1706 |
| | 7D | 0.0802 | 0.3176 | 0.2611 | 0.2063 | 0.0249 | 0.1433 | 0.0978 | 0.0746 | 0.1507 |
| | Avg. | 0.1071 | 0.3734 | 0.3218 | 0.2419 | 0.0432 | 0.1674 | 0.1247 | 0.0937 | 0.1842 |
| LSTM | 1D | 0.4140 | 0.4712 | 0.4004 | 0.4026 | 0.2813 | 0.1812 | 0.2226 | 0.2254 | 0.3249 |
| | 3D | 0.2247 | 0.2719 | 0.2113 | 0.2167 | 0.1419 | 0.0936 | 0.1123 | 0.1012 | 0.1717 |
| | 5D | 0.1839 | 0.1920 | 0.1746 | 0.1489 | 0.1184 | 0.0703 | 0.0868 | 0.0744 | 0.1312 |
| | 7D | 0.1173 | 0.1752 | 0.1429 | 0.1642 | 0.0394 | 0.0573 | 0.0715 | 0.0608 | 0.1036 |
| | Avg. | 0.2350 | 0.2776 | 0.2323 | 0.2331 | 0.1453 | 0.1006 | 0.1233 | 0.1155 | 0.1828 |
| GCN | 1D | 0.4770 | 0.4553 | 0.4873 | 0.4167 | 0.3108 | 0.2708 | 0.3098 | 0.2742 | 0.3752 |
| | 3D | 0.4046 | 0.4419 | 0.3886 | 0.3418 | 0.2585 | 0.2237 | 0.2572 | 0.2331 | 0.3187 |
| | 5D | 0.3840 | 0.3823 | 0.3712 | 0.3168 | 0.2323 | 0.1991 | 0.2144 | 0.2147 | 0.2894 |
| | 7D | 0.3641 | 0.3583 | 0.3773 | 0.2930 | 0.2013 | 0.1843 | 0.1942 | 0.1889 | 0.2702 |
| | Avg. | 0.4074 | 0.4095 | 0.4061 | 0.3421 | 0.2507 | 0.2195 | 0.2439 | 0.2277 | 0.3134 |

Table 43: MAE results of input factors ablation study on S6. All, G, R, and C represent all factors, groundwater, rainfall, and human control(pump and gate).

| Method | T | w/ All | w/o G | w/o R | w/o C | w/o GR | w/o RC | w/o WC | w/o WRC |
|---|---|---|---|---|---|---|---|---|---|
| iTransformer | 1D | 0.0738 | 0.0735 | 0.0729 | 0.0733 | 0.0739 | 0.0737 | 0.0739 | 0.0742 |
| | 3D | 0.1275 | 0.1279 | 0.1278 | 0.1275 | 0.1290 | 0.1280 | 0.1265 | 0.1276 |
| | 5D | 0.1660 | 0.1662 | 0.1664 | 0.1662 | 0.1660 | 0.1660 | 0.1658 | 0.1671 |
| | 7D | 0.1953 | 0.1953 | 0.1953 | 0.1948 | 0.1954 | 0.1961 | 0.1947 | 0.1955 |
| | Avg. | 0.1406 | 0.1407 | 0.1406 | 0.1405 | 0.1411 | 0.1410 | **0.1402** | 0.1411 |
| PatchTST | 1D | 0.0726 | 0.0731 | 0.0713 | 0.0719 | 0.0732 | 0.0722 | 0.0727 | 0.0772 |
| | 3D | 0.1252 | 0.1262 | 0.1252 | 0.1255 | 0.1273 | 0.1273 | 0.1254 | 0.1295 |
| | 5D | 0.1615 | 0.1639 | 0.1624 | 0.1618 | 0.1649 | 0.1649 | 0.1634 | 0.1667 |
| | 7D | 0.1910 | 0.1933 | 0.1923 | 0.1916 | 0.1938 | 0.1941 | 0.1927 | 0.1951 |
| | Avg. | **0.1376** | 0.1391 | 0.1378 | 0.1377 | 0.1398 | 0.1396 | 0.1385 | 0.1421 |
| TSMixer | 1D | 0.1004 | 0.0781 | 0.1599 | 0.1443 | 0.0778 | 0.1789 | 0.0782 | 0.0782 |
| | 3D | 0.1471 | 0.1296 | 0.2611 | 0.2216 | 0.1297 | 0.2506 | 0.1299 | 0.1299 |
| | 5D | 0.1802 | 0.1658 | 0.3104 | 0.2350 | 0.1657 | 0.3018 | 0.1664 | 0.1663 |
| | 7D | 0.2107 | 0.1941 | 0.2776 | 0.2435 | 0.1939 | 0.3914 | 0.1947 | 0.1944 |
| | Avg. | 0.1596 | 0.1419 | 0.2523 | 0.2111 | **0.1418** | 0.2807 | 0.1423 | 0.1422 |
| NLinear | 1D | 0.0937 | 0.0977 | 0.0854 | 0.0933 | 0.0850 | 0.0804 | 0.0983 | 0.0780 |
| | 3D | 0.1429 | 0.1461 | 0.1363 | 0.1422 | 0.1366 | 0.1328 | 0.1462 | 0.1314 |
| | 5D | 0.1769 | 0.1796 | 0.1713 | 0.1764 | 0.1717 | 0.1689 | 0.1797 | 0.1678 |
| | 7D | 0.2050 | 0.2074 | 0.2003 | 0.2043 | 0.2008 | 0.1978 | 0.2072 | 0.1970 |
| | Avg. | 0.1546 | 0.1577 | 0.1483 | 0.1540 | 0.1485 | 0.1450 | 0.1579 | **0.1435** |
| TimesNet | 1D | 0.0986 | 0.0936 | 0.0986 | 0.0986 | 0.0928 | 0.0970 | 0.0924 | 0.0930 |
| | 3D | 0.1513 | 0.1446 | 0.1511 | 0.1525 | 0.1465 | 0.1529 | 0.1448 | 0.1468 |
| | 5D | 0.1883 | 0.1838 | 0.1889 | 0.1890 | 0.1843 | 0.1908 | 0.1830 | 0.1814 |
| | 7D | 0.2187 | 0.2177 | 0.2262 | 0.2200 | 0.2131 | 0.2219 | 0.2111 | 0.2109 |
| | Avg. | 0.1642 | 0.1599 | 0.1662 | 0.1650 | 0.1592 | 0.1656 | **0.1578** | 0.1580 |

Table 44: MSE results of input factors ablation study on S6. All, G, R, and C represent all factors, groundwater, rainfall, and human control(pump and gate).

| Method | T | w/ All | w/o G | w/o R | w/o C | w/o GR | w/o RC | w/o WC | w/o WRC |
|---|---|---|---|---|---|---|---|---|---|
| iTransformer | 1D | 0.0400 | 0.0407 | 0.0398 | 0.0402 | 0.0405 | 0.0400 | 0.0405 | 0.0409 |
| | 3D | 0.0817 | 0.0828 | 0.0821 | 0.0818 | 0.0827 | 0.0825 | 0.0818 | 0.0832 |
| | 5D | 0.1175 | 0.1172 | 0.1170 | 0.1167 | 0.1166 | 0.1169 | 0.1164 | 0.1184 |
| | 7D | 0.1419 | 0.1430 | 0.1407 | 0.1430 | 0.1429 | 0.1431 | 0.1407 | 0.1424 |
| | Avg. | 0.0953 | 0.0960 | **0.0949** | 0.0954 | 0.0957 | 0.0956 | **0.0949** | 0.0962 |
| PatchTST | 1D | 0.0395 | 0.0393 | 0.0392 | 0.0393 | 0.0405 | 0.0399 | 0.0389 | 0.0432 |
| | 3D | 0.0798 | 0.0807 | 0.0806 | 0.0797 | 0.0831 | 0.0819 | 0.0797 | 0.0850 |
| | 5D | 0.1107 | 0.1128 | 0.1122 | 0.1105 | 0.1154 | 0.1140 | 0.1123 | 0.1173 |
| | 7D | 0.1368 | 0.1400 | 0.1388 | 0.1367 | 0.1412 | 0.1393 | 0.1385 | 0.1428 |
| | Avg. | 0.0917 | 0.0932 | 0.0927 | **0.0916** | 0.0950 | 0.0938 | **0.0923** | 0.0971 |
| TSMixer | 1D | 0.0562 | 0.0415 | 0.1891 | 0.2590 | 0.0414 | 0.4396 | 0.0419 | 0.0416 |
| | 3D | 0.0957 | 0.0827 | 0.4979 | 0.2580 | 0.0830 | 0.4833 | 0.0825 | 0.0826 |
| | 5D | 0.1243 | 0.1145 | 0.5722 | 0.3232 | 0.1145 | 0.4855 | 0.1139 | 0.1141 |
| | 7D | 0.1557 | 0.1396 | 0.3653 | 0.2389 | 0.1394 | 1.2701 | 0.1389 | 0.1382 |
| | Avg. | 0.1080 | 0.0946 | 0.4061 | 0.2698 | 0.0946 | 0.6697 | 0.0943 | **0.0941** |
| NLinear | 1D | 0.0477 | 0.0497 | 0.0450 | 0.0471 | 0.0459 | 0.0434 | 0.0494 | 0.0426 |
| | 3D | 0.0878 | 0.0898 | 0.0848 | 0.0871 | 0.0856 | 0.0837 | 0.0895 | 0.0829 |
| | 5D | 0.1178 | 0.1196 | 0.1154 | 0.1171 | 0.1160 | 0.1145 | 0.1191 | 0.1139 |
| | 7D | 0.1434 | 0.1451 | 0.1411 | 0.1425 | 0.1417 | 0.1400 | 0.1444 | 0.1395 |
| | Avg. | 0.0992 | 0.1011 | 0.0966 | 0.0984 | 0.0973 | 0.0954 | 0.1006 | 0.0947 |
| TimesNet | 1D | 0.0564 | 0.0518 | 0.0580 | 0.0573 | 0.0521 | 0.0550 | 0.0515 | 0.0517 |
| | 3D | 0.1049 | 0.0970 | 0.1043 | 0.1049 | 0.0985 | 0.1086 | 0.0966 | 0.1015 |
| | 5D | 0.1409 | 0.1357 | 0.1419 | 0.1416 | 0.1397 | 0.1485 | 0.1338 | 0.1338 |
| | 7D | 0.1729 | 0.1770 | 0.1863 | 0.1770 | 0.1677 | 0.1826 | 0.1624 | 0.1624 |
| | Avg. | 0.1188 | 0.1154 | 0.1226 | 0.1202 | 0.1145 | 0.1237 | **0.1111** | 0.1123 |

Table 45: SEDI(10%) results of input factors ablation study on S6. All, G, R, and C represent all factors, groundwater, rainfall, and human control(pump and gate).

| Method | T | w/ All | w/o G | w/o R | w/o C | w/o GR | w/o RC | w/o WC | w/o WRC |
|---|---|---|---|---|---|---|---|---|---|
| iTransformer | 1D | 0.6508 | 0.6479 | 0.6510 | 0.6551 | 0.6525 | 0.6548 | 0.6512 | 0.6520 |
| | 3D | 0.5526 | 0.5429 | 0.5485 | 0.5510 | 0.5393 | 0.5485 | 0.5513 | 0.5533 |
| | 5D | 0.4856 | 0.4793 | 0.4831 | 0.4877 | 0.4786 | 0.4864 | 0.4824 | 0.4779 |
| | 7D | 0.4384 | 0.4323 | 0.4408 | 0.4399 | 0.4301 | 0.4380 | 0.4320 | 0.4326 |
| | Avg. | 0.5318 | 0.5256 | 0.5308 | **0.5334** | 0.5251 | 0.5319 | 0.5292 | 0.5289 |
| PatchTST | 1D | 0.6632 | 0.6629 | 0.6665 | 0.6644 | 0.6686 | 0.6618 | 0.6621 | 0.6561 |
| | 3D | 0.5696 | 0.5614 | 0.5704 | 0.5717 | 0.5616 | 0.5586 | 0.5658 | 0.5455 |
| | 5D | 0.5028 | 0.5013 | 0.4997 | 0.5043 | 0.4926 | 0.4945 | 0.4993 | 0.4857 |
| | 7D | 0.4562 | 0.4518 | 0.4498 | 0.4544 | 0.4442 | 0.4465 | 0.4476 | 0.4364 |
| | Avg. | 0.5480 | 0.5443 | 0.5466 | **0.5487** | 0.5418 | 0.5404 | 0.5437 | 0.5309 |
| TSMixer | 1D | 0.6052 | 0.6538 | 0.5202 | 0.5722 | 0.6475 | 0.5493 | 0.6499 | 0.6525 |
| | 3D | 0.5267 | 0.5571 | 0.4251 | 0.4290 | 0.5580 | 0.4402 | 0.5581 | 0.5596 |
| | 5D | 0.4732 | 0.4959 | 0.3523 | 0.4067 | 0.4963 | 0.3709 | 0.4932 | 0.4973 |
| | 7D | 0.4245 | 0.4487 | 0.3516 | 0.3703 | 0.4497 | 0.3206 | 0.4454 | 0.4474 |
| | Avg. | 0.5074 | 0.5389 | 0.4123 | 0.4446 | 0.5379 | 0.4202 | 0.5367 | **0.5392** |
| NLinear | 1D | 0.6477 | 0.6485 | 0.6485 | 0.6481 | 0.6506 | 0.6529 | 0.6602 | 0.6561 |
| | 3D | 0.5476 | 0.5556 | 0.5521 | 0.5472 | 0.5521 | 0.5566 | 0.5517 | 0.5582 |
| | 5D | 0.4868 | 0.4928 | 0.4905 | 0.4865 | 0.4904 | 0.4936 | 0.4960 | 0.4950 |
| | 7D | 0.4356 | 0.4355 | 0.4383 | 0.4345 | 0.4381 | 0.4440 | 0.4345 | 0.4450 |
| | Avg. | 0.5294 | 0.5331 | 0.5324 | 0.5291 | 0.5328 | 0.5368 | 0.5356 | **0.5386** |
| TimesNet | 1D | 0.5936 | 0.6093 | 0.5855 | 0.5884 | 0.6067 | 0.5939 | 0.6086 | 0.6089 |
| | 3D | 0.4772 | 0.5059 | 0.4986 | 0.4925 | 0.5117 | 0.4854 | 0.5093 | 0.5046 |
| | 5D | 0.4304 | 0.4489 | 0.4264 | 0.4305 | 0.4460 | 0.4202 | 0.4456 | 0.4396 |
| | 7D | 0.3878 | 0.3959 | 0.3696 | 0.3837 | 0.4095 | 0.3839 | 0.4091 | 0.4082 |
| | Avg. | 0.4723 | 0.4900 | 0.4700 | 0.4738 | **0.4935** | 0.4709 | 0.4931 | 0.4903 |

Table 46: MAE results of temporal information ablation study on S6.

| Method | T | 6H | 12H | 1D | 2D | 3D | 4D | 5D | 6D |
|---|---|---|---|---|---|---|---|---|---|
| iTransformer | 1D | 0.1136 | 0.1123 | 0.1097 | 0.1123 | 0.1133 | 0.1198 | 0.1214 | 0.1209 |
| | 3D | 0.1999 | 0.1978 | 0.1980 | 0.2025 | 0.2071 | 0.2121 | 0.2168 | 0.2137 |
| | 5D | 0.2571 | 0.2537 | 0.2547 | 0.2586 | 0.2725 | 0.2776 | 0.2815 | 0.2779 |
| | 7D | 0.3036 | 0.3007 | 0.3015 | 0.3059 | 0.3179 | 0.3242 | 0.3221 | 0.3261 |
| | Avg. | 0.2185 | _0.2161_ | **0.2160** | 0.2198 | 0.2277 | 0.2334 | 0.2354 | 0.2346 |
| PatchTST | 1D | 0.1266 | 0.1164 | 0.1135 | 0.1127 | 0.1094 | 0.1099 | 0.1106 | 0.1159 |
| | 3D | 0.2123 | 0.2067 | 0.1988 | 0.1979 | 0.1963 | 0.1940 | 0.1979 | 0.2011 |
| | 5D | 0.2693 | 0.2650 | 0.2566 | 0.2530 | 0.2554 | 0.2518 | 0.2522 | 0.2559 |
| | 7D | 0.3143 | 0.3116 | 0.3039 | 0.2988 | 0.2984 | 0.2983 | 0.3011 | 0.3069 |
| | Avg. | 0.2306 | 0.2249 | 0.2182 | 0.2156 | _0.2149_ | **0.2135** | 0.2155 | 0.2199 |
| TSMixer | 1D | 0.1197 | 0.1216 | 0.1237 | 0.1269 | 0.1258 | 0.1269 | 0.1289 | 0.1266 |
| | 3D | 0.2021 | 0.2033 | 0.2058 | 0.2087 | 0.2051 | 0.2073 | 0.2074 | 0.2052 |
| | 5D | 0.2586 | 0.2592 | 0.2618 | 0.2639 | 0.2576 | 0.2604 | 0.2585 | 0.2582 |
| | 7D | 0.3046 | 0.3049 | 0.3070 | 0.3092 | 0.3007 | 0.3029 | 0.2997 | 0.3009 |
| | Avg. | **0.2213** | _0.2222_ | 0.2246 | 0.2272 | 0.2223 | 0.2244 | 0.2236 | 0.2227 |
| NLinear | 1D | 0.1159 | 0.1180 | 0.1211 | 0.1462 | 0.1505 | 0.1508 | 0.1587 | 0.1586 |
| | 3D | 0.1998 | 0.2011 | 0.2047 | 0.2220 | 0.2289 | 0.2305 | 0.2388 | 0.2382 |
| | 5D | 0.2568 | 0.2577 | 0.2605 | 0.2743 | 0.2796 | 0.2870 | 0.2871 | 0.2925 |
| | 7D | 0.3031 | 0.3036 | 0.3064 | 0.3186 | 0.3234 | 0.3293 | 0.3323 | 0.3355 |
| | Avg. | **0.2189** | _0.2201_ | 0.2232 | 0.2403 | 0.2456 | 0.2494 | 0.2542 | 0.2562 |
| TimesNet | 1D | 0.1175 | 0.1234 | 0.1357 | 0.1565 | 0.1750 | 0.1903 | 0.2066 | 0.2184 |
| | 3D | 0.2017 | 0.2084 | 0.2235 | 0.2423 | 0.2618 | 0.2775 | 0.2880 | 0.3012 |
| | 5D | 0.2602 | 0.2635 | 0.2757 | 0.2887 | 0.2989 | 0.3235 | 0.3386 | 0.3657 |
| | 7D | 0.3086 | 0.3097 | 0.3254 | 0.3272 | 0.3570 | 0.3636 | 0.3804 | 0.4016 |
| | Avg. | **0.2220** | _0.2262_ | 0.2401 | 0.2537 | 0.2732 | 0.2887 | 0.3034 | 0.3217 |

Table 47: MSE results of temporal information ablation study on S6.

| Method | T | 6H | 12H | 1D | 2D | 3D | 4D | 5D | 6D |
|---|---|---|---|---|---|---|---|---|---|
| iTransformer | 1D | 0.0829 | 0.0815 | 0.0793 | 0.0825 | 0.0844 | 0.0878 | 0.0899 | 0.0899 |
| | 3D | 0.2093 | 0.2068 | 0.2082 | 0.2152 | 0.2218 | 0.2281 | 0.2379 | 0.2306 |
| | 5D | 0.3186 | 0.3162 | 0.3174 | 0.3219 | 0.3472 | 0.3551 | 0.3556 | 0.3594 |
| | 7D | 0.4170 | 0.4149 | 0.4139 | 0.4220 | 0.4499 | 0.4614 | 0.4614 | 0.4757 |
| | Avg. | 0.2569 | _0.2549_ | **0.2547** | 0.2604 | 0.2758 | 0.2831 | 0.2862 | 0.2889 |
| PatchTST | 1D | 0.0888 | 0.0860 | 0.0851 | 0.0827 | 0.0795 | 0.0787 | 0.0788 | 0.0835 |
| | 3D | 0.2151 | 0.2104 | 0.2107 | 0.2078 | 0.2064 | 0.2050 | 0.2072 | 0.2089 |
| | 5D | 0.3243 | 0.3214 | 0.3210 | 0.3156 | 0.3130 | 0.3256 | 0.3408 | 0.3187 |
| | 7D | 0.4236 | 0.4187 | 0.4270 | 0.4193 | 0.4275 | 0.4548 | 0.4255 | 0.4348 |
| | Avg. | 0.2630 | 0.2592 | 0.2609 | 0.2563 | 0.2566 | 0.2660 | _0.2631_ | **0.2615** |
| TSMixer | 1D | 0.0899 | 0.0899 | 0.0888 | 0.0887 | 0.0860 | 0.0853 | 0.0855 | 0.0837 |
| | 3D | 0.2152 | 0.2145 | 0.2141 | 0.2121 | 0.2047 | 0.2042 | 0.2034 | 0.1992 |
| | 5D | 0.3241 | 0.3228 | 0.3226 | 0.3170 | 0.3060 | 0.3050 | 0.3035 | 0.3024 |
| | 7D | 0.4235 | 0.4222 | 0.4214 | 0.4142 | 0.3999 | 0.3982 | 0.3968 | 0.3985 |
| | Avg. | 0.2632 | 0.2623 | 0.2617 | 0.2580 | 0.2491 | 0.2482 | _0.2473_ | **0.2460** |
| NLinear | 1D | 0.0852 | 0.0858 | 0.0873 | 0.1111 | 0.1135 | 0.1122 | 0.1196 | 0.1176 |
| | 3D | 0.2111 | 0.2115 | 0.2148 | 0.2370 | 0.2444 | 0.2436 | 0.2535 | 0.2509 |
| | 5D | 0.3201 | 0.3206 | 0.3236 | 0.3439 | 0.3490 | 0.3587 | 0.3563 | 0.3629 |
| | 7D | 0.4199 | 0.4200 | 0.4235 | 0.4427 | 0.4480 | 0.4555 | 0.4586 | 0.4623 |
| | Avg. | **0.2591** | _0.2595_ | 0.2623 | 0.2836 | 0.2887 | 0.2925 | 0.2970 | 0.2984 |
| TimesNet | 1D | 0.0866 | 0.0937 | 0.1104 | 0.1357 | 0.1603 | 0.1750 | 0.2039 | 0.2297 |
| | 3D | 0.2177 | 0.2333 | 0.2594 | 0.2953 | 0.3156 | 0.3607 | 0.3811 | 0.4114 |
| | 5D | 0.3316 | 0.3372 | 0.3781 | 0.3997 | 0.4169 | 0.4597 | 0.5154 | 0.5691 |
| | 7D | 0.4363 | 0.4407 | 0.4918 | 0.4891 | 0.5910 | 0.5837 | 0.5953 | 0.6566 |
| | Avg. | **0.2681** | _0.2762_ | 0.3100 | 0.3299 | 0.3710 | 0.3948 | 0.4239 | 0.4667 |

Table 48: SEDI(10%) results of temporal information ablation study on S6.

| Method | T | 6H | 12H | 1D | 2D | 3D | 4D | 5D | 6D |
|---|---|---|---|---|---|---|---|---|---|
| iTransformer | 1D | 0.7825 | 0.7945 | 0.7941 | 0.7898 | 0.7703 | 0.7857 | 0.7728 | 0.7632 |
| | 3D | 0.6792 | 0.6945 | 0.6987 | 0.6957 | 0.6847 | 0.6801 | 0.6691 | 0.6627 |
| | 5D | 0.6220 | 0.6392 | 0.6420 | 0.6318 | 0.6201 | 0.6189 | 0.6178 | 0.6022 |
| | 7D | 0.5797 | 0.5984 | 0.6006 | 0.5992 | 0.5867 | 0.5841 | 0.5738 | 0.5665 |
| | Avg. | 0.6659 | _0.6816_ | **0.6838** | 0.6791 | 0.6654 | 0.6672 | 0.6584 | 0.6487 |
| PatchTST | 1D | 0.7522 | 0.7946 | 0.8004 | 0.8033 | 0.8100 | 0.8118 | 0.8058 | 0.8099 |
| | 3D | 0.6324 | 0.6425 | 0.6977 | 0.7010 | 0.7046 | 0.7030 | 0.7033 | 0.7032 |
| | 5D | 0.5820 | 0.5796 | 0.6389 | 0.6427 | 0.6404 | 0.6467 | 0.6362 | 0.6326 |
| | 7D | 0.5415 | 0.5314 | 0.5959 | 0.6024 | 0.5986 | 0.5957 | 0.5929 | 0.5982 |
| | Avg. | 0.6270 | 0.6370 | 0.6832 | 0.6874 | _0.6884_ | **0.6893** | 0.6846 | 0.6860 |
| TSMixer | 1D | 0.7812 | 0.7886 | 0.7954 | 0.7912 | 0.7931 | 0.7965 | 0.7905 | 0.7952 |
| | 3D | 0.6861 | 0.6906 | 0.6971 | 0.7005 | 0.7016 | 0.7067 | 0.6946 | 0.6983 |
| | 5D | 0.6307 | 0.6351 | 0.6397 | 0.6463 | 0.6445 | 0.6503 | 0.6400 | 0.6385 |
| | 7D | 0.5917 | 0.5942 | 0.5984 | 0.6054 | 0.6003 | 0.6038 | 0.5959 | 0.5900 |
| | Avg. | 0.6724 | 0.6771 | 0.6827 | _0.6859_ | 0.6849 | **0.6893** | 0.6803 | 0.6805 |
| NLinear | 1D | 0.7915 | 0.8002 | 0.8029 | 0.7843 | 0.7891 | 0.7933 | 0.7929 | 0.7904 |
| | 3D | 0.6956 | 0.6989 | 0.7019 | 0.6857 | 0.6958 | 0.6974 | 0.6925 | 0.6940 |
| | 5D | 0.6359 | 0.6383 | 0.6447 | 0.6320 | 0.6329 | 0.6373 | 0.6422 | 0.6373 |
| | 7D | 0.5944 | 0.5995 | 0.6016 | 0.5875 | 0.5889 | 0.5951 | 0.5973 | 0.5994 |
| | Avg. | 0.6794 | _0.6842_ | **0.6878** | 0.6724 | 0.6767 | 0.6808 | 0.6812 | 0.6803 |
| TimesNet | 1D | 0.7590 | 0.7610 | 0.7482 | 0.7130 | 0.6908 | 0.6854 | 0.6548 | 0.6392 |
| | 3D | 0.6582 | 0.6606 | 0.6499 | 0.6148 | 0.5993 | 0.5889 | 0.5790 | 0.5731 |
| | 5D | 0.5976 | 0.6096 | 0.5988 | 0.5779 | 0.5631 | 0.5461 | 0.5458 | 0.5246 |
| | 7D | 0.5647 | 0.5692 | 0.5451 | 0.5447 | 0.5269 | 0.5263 | 0.5062 | 0.5060 |
| | Avg. | _0.6449_ | **0.6501** | 0.6355 | 0.6126 | 0.5950 | 0.5867 | 0.5714 | 0.5607 |

Table 49: MAE results of spatial information ablation study on S6.

| Method | T | MAE | | | | | MSE | | | | | SEDI(10%) | | | | |
|---|---|---|---|---|---|---|---|---|---|---|---|---|---|---|---|---|
| | | 1.0R | 1.2R | 1.4R | 1.6R | 1.8R | 1.0R | 1.2R | 1.4R | 1.6R | 1.8R | 1.0R | 1.2R | 1.4R | 1.6R | 1.8R |
| iTransformer | 1D | 0.1123 | 0.1135 | 0.1125 | 0.1099 | 0.1119 | 0.0825 | 0.0825 | 0.0818 | 0.0822 | 0.0807 | 0.7898 | 0.7810 | 0.7894 | 0.7863 | 0.7939 |
| | 3D | 0.2025 | 0.2038 | 0.2020 | 0.2024 | 0.2021 | 0.2152 | 0.2144 | 0.2117 | 0.2136 | 0.2119 | 0.6957 | 0.7011 | 0.6952 | 0.7022 | 0.7036 |
| | 5D | 0.2586 | 0.2650 | 0.2584 | 0.2538 | 0.2513 | 0.3219 | 0.3314 | 0.3241 | 0.3257 | 0.3186 | 0.6318 | 0.6294 | 0.6329 | 0.6353 | 0.6361 |
| | 7D | 0.3059 | 0.3040 | 0.3017 | 0.3006 | 0.2995 | 0.4220 | 0.4239 | 0.4181 | 0.4176 | 0.4197 | 0.5992 | 0.5973 | 0.5989 | 0.5972 | 0.5965 |
| | Avg. | 0.2198 | 0.2216 | 0.2187 | _0.2167_ | **0.2162** | 0.2604 | 0.2631 | _0.2589_ | 0.2598 | **0.2577** | 0.6791 | 0.6772 | 0.6791 | _0.6803_ | **0.6825** |
| PatchTST | 1D | 0.1127 | 0.1131 | 0.1139 | 0.1113 | 0.1118 | 0.0827 | 0.0833 | 0.0829 | 0.0825 | 0.0821 | 0.8033 | 0.8030 | 0.8066 | 0.8068 | 0.8095 |
| | 3D | 0.1979 | 0.1967 | 0.1990 | 0.1976 | 0.1976 | 0.2078 | 0.2060 | 0.2087 | 0.2082 | 0.2083 | 0.7010 | 0.7036 | 0.7048 | 0.7040 | 0.7073 |
| | 5D | 0.2530 | 0.2531 | 0.2548 | 0.2556 | 0.2550 | 0.3156 | 0.3133 | 0.3133 | 0.3182 | 0.3160 | 0.6427 | 0.6440 | 0.6433 | 0.6437 | 0.6483 |
| | 7D | 0.2988 | 0.2971 | 0.2994 | 0.3000 | 0.2991 | 0.4193 | 0.4102 | 0.4102 | 0.4115 | 0.4097 | 0.6024 | 0.6035 | 0.6049 | 0.6056 | 0.6065 |
| | Avg. | _0.2156_ | **0.2150** | 0.2168 | 0.2161 | 0.2159 | 0.2563 | **0.2532** | 0.2542 | 0.2551 | _0.2540_ | 0.6874 | 0.6885 | 0.6899 | _0.6900_ | **0.6929** |
| TSMixer | 1D | 0.1269 | 0.1259 | 0.1255 | 0.1271 | 0.1256 | 0.0887 | 0.0882 | 0.0876 | 0.0888 | 0.0879 | 0.7912 | 0.7896 | 0.7918 | 0.7895 | 0.7891 |
| | 3D | 0.2087 | 0.2082 | 0.2072 | 0.2075 | 0.2064 | 0.2121 | 0.2125 | 0.2096 | 0.2093 | 0.2096 | 0.7005 | 0.6996 | 0.7026 | 0.7011 | 0.7001 |
| | 5D | 0.2639 | 0.2638 | 0.2621 | 0.2632 | 0.2625 | 0.3170 | 0.3182 | 0.3136 | 0.3147 | 0.3146 | 0.6463 | 0.6451 | 0.6503 | 0.6484 | 0.6492 |
| | 7D | 0.3092 | 0.3086 | 0.3074 | 0.3082 | 0.3062 | 0.4142 | 0.4151 | 0.4117 | 0.4114 | 0.4097 | 0.6054 | 0.6042 | 0.6078 | 0.6066 | 0.6055 |
| | Avg. | 0.2272 | 0.2266 | _0.2255_ | 0.2265 | **0.2252** | 0.2580 | 0.2585 | _0.2556_ | 0.2560 | **0.2554** | 0.6859 | 0.6846 | **0.6881** | _0.6864_ | 0.6860 |
| NLinear | 1D | 0.1462 | 0.1482 | 0.1473 | 0.1464 | 0.1449 | 0.1111 | 0.1133 | 0.1123 | 0.1113 | 0.1097 | 0.7843 | 0.7816 | 0.7807 | 0.7811 | 0.7821 |
| | 3D | 0.2220 | 0.2238 | 0.2227 | 0.2213 | 0.2205 | 0.2370 | 0.2397 | 0.2381 | 0.2358 | 0.2347 | 0.6857 | 0.6836 | 0.6850 | 0.6865 | 0.6876 |
| | 5D | 0.2743 | 0.2756 | 0.2746 | 0.2744 | 0.2735 | 0.3439 | 0.3461 | 0.3443 | 0.3437 | 0.3423 | 0.6320 | 0.6300 | 0.6315 | 0.6317 | 0.6310 |
| | 7D | 0.3186 | 0.3199 | 0.3191 | 0.3181 | 0.3173 | 0.4427 | 0.4450 | 0.4435 | 0.4415 | 0.4402 | 0.5875 | 0.5888 | 0.5904 | 0.5930 | 0.5932 |
| | Avg. | 0.2403 | 0.2419 | 0.2409 | _0.2400_ | **0.2390** | 0.2836 | 0.2860 | 0.2845 | _0.2831_ | **0.2817** | 0.6724 | 0.6710 | 0.6719 | _0.6731_ | **0.6735** |
| TimesNet | 1D | 0.1565 | 0.1582 | 0.1597 | 0.1655 | 0.1661 | 0.1357 | 0.1423 | 0.1392 | 0.1458 | 0.1492 | 0.7130 | 0.7099 | 0.7155 | 0.7093 | 0.7007 |
| | 3D | 0.2423 | 0.2451 | 0.2408 | 0.2424 | 0.2454 | 0.2953 | 0.3075 | 0.2847 | 0.2893 | 0.3104 | 0.6148 | 0.6168 | 0.6174 | 0.6202 | 0.6131 |
| | 5D | 0.2887 | 0.2935 | 0.3038 | 0.2924 | 0.2985 | 0.3997 | 0.4087 | 0.4531 | 0.4028 | 0.4239 | 0.5779 | 0.5812 | 0.5651 | 0.5824 | 0.5674 |
| | 7D | 0.3272 | 0.3399 | 0.3380 | 0.3422 | 0.3413 | 0.4891 | 0.5223 | 0.5136 | 0.5219 | 0.5451 | 0.5447 | 0.5377 | 0.5359 | 0.5426 | 0.5410 |
| | Avg. | **0.2537** | _0.2592_ | 0.2606 | 0.2606 | 0.2628 | **0.3299** | 0.3452 | 0.3476 | _0.3400_ | 0.3571 | _0.6126_ | 0.6114 | 0.6085 | **0.6136** | 0.6056 |

