# OpenReview forum: "Exploring Data-Driven Models for Compound Flood Forecasting: A comprehensive benchmark"
_ICLR.cc/2026/Conference — Submitted to ICLR 2026_

### Official Review · Reviewer_rJ8Q · 2025-10-29

**Soundness:** 1
**Presentation:** 2
**Contribution:** 1
**Rating:** 2
**Confidence:** 4

**Summary:**

This manuscript introduces SF${^2}$Bench, a new benchmark related to compound flood forecasting. The dataset covers South Florid and includes features like water level, sea level, groundwater level, rainfall, and human management activities on hydraulic structures (gates and pumps). It is argued that these key features are missing from current benchmarks for analyzing compound floods. The manuscript further benchmarks different types of ML models for timeseries forecasting like MLP, RNN, CNN, GNN, and LLM.

**Strengths:**

- It is beneficial to provide a dataset which not only includes water level but also different target features related to compound flood.
- Baselines are described in detail.
- The paper is clear and concise.

**Weaknesses:**

- **Contribution**. The contribution is very limited to be published at ICLR and not much interesting for people from ML community. Usually people evaluate on GRDC, CARAVAN and CAMEL datasets and adding a new dataset is beneficial but I think it would not close a critical gap for people developing models for flood forecasting.
- Missing standard evaluation metrics like NSE and F1. MAE or MSE are not metrics for compound floods.
- L69-73: the manuscript argues that prior works focus on limited regions. The introduced dataset does not overcome this limitation and it is also limited to a local region.
- Data Processing. For a benchmark, I would not do interpolation to replace missing data. Usually, we missing data is flagged rather than filled with assumed values especially if the frequency is daily. Moreover, why assuming a linear interpolation for streamflow?
- The paper does not report confidence intervals or standard deviations. The results are based on single runs, making it difficult to assess the statistical significance or stability of the benchmark baselines.

**Questions:**

- L250-252: where do we see this influence? It is not clear from the ablation study and it shows the opposite.
- Tables 3 and 4: isn’t it surprising to see MLP outperforming LSTM in water level forecasting? This is against a plethora of studies which rely on LSTM for temporal modeling especially for streamflow forecasting.
- Table 6: it looks like most features are not important at all
- L1072: loss function like MSE is not a good loss for tasks related to extremes forecasting e.g., compound flood.
- Flood forecasting is an active field of research. There are some key related works from different categories are missing e.g., to name a few [[1](https://www.nature.com/articles/s41586-024-07145-1), [2](https://hess.copernicus.org/articles/27/1/2023/), [3](https://arxiv.org/abs/2505.22535), [4](https://agupubs.onlinelibrary.wiley.com/doi/full/10.1029/2023WR036170), [5](https://agupubs.onlinelibrary.wiley.com/doi/10.1029/2023WR035337)]. Dataset like [CARAVAN](https://www.nature.com/articles/s41597-023-01975-w).

**Minor**:
- L264-266: but do you still use all information as input? Please clarify this in the manuscript.
- Figure 3 (b): I can’t see anything from this figure. All curves are plotted above each other.
- Page 7: I would remove content from the main paper.

---

### Official Review · Reviewer_LNYo · 2025-10-30

**Soundness:** 3
**Presentation:** 2
**Contribution:** 2
**Rating:** 4
**Confidence:** 3

**Summary:**

This paper presents a large-scale benchmark dataset for compound flood forecasting, called $SF^{2}Bench$. Its primary contribution is the novel integration of four key drivers often treated in isolation. This paper provides a comprehensive benchmark of six classes of deep learning models, evaluating them on both standard (MAE/MSE) and crucial extreme-event (SEDI) metrics.

**Strengths:**

1. The paper provides a high-quality and public benchmark is essential for advancing data-driven forecasting methods, which tackles a critical and intensifying real-world problem.
2. The benchmark is rigorous, evaluating a wide array of models, from simple linear ones to SOTA Transformers and LLMs.
3. The results provide novel insights, which include the observed disconnect between MAE/MSE and SEDI performance. This highlights the need for specialized models for extreme events, and the strong performance of simple MLP models against larger ones.

**Weaknesses:**

1. The description of the factor ablation study in Section 4.5, paragraphs 3, is confusing and contains conclusions that contradict Table 6. For example, this paper claimed that TimesNet sees "performance improvements" with all factors (line 434). However, its MAE worsens from 0.1580 ("w/o WRC") to 0.1642 ("w/ All"). Moreover, this paragraph also claims that "providing only groundwater information... leads to the best performance" for iTransformer, which is also contradicted by Table 6, where "w/ All" (0.1406) is better than "w/o RC" (0.1410). This section needs a thorough revision to ensure the text accurately reflects the data.
2. The paper critiques traditional physics-based models (e.g., HEC-RAS) as "time-inefficient"  but provides no quantitative comparison (in accuracy or efficiency). Including even a simplified physics-based or hybrid model baseline would significantly strengthen the claims of data-driven models' superiority.

Minor issue:
- The caption of Figure 16 incorrectly states "Dilated. means DilatedRNN", which seems copied from Figure 14.

**Questions:**

See weaknesses.

---

### Official Review · Reviewer_Ur8Y · 2025-10-31

**Soundness:** 1
**Presentation:** 2
**Contribution:** 1
**Rating:** 2
**Confidence:** 3

**Summary:**

This work introduces a dataset for flood prediction in South Florida.
A set of basic deep learning models is used to provide baseline performance
for the task of predicting 1-7 days ahead given two days worth of data.

**Strengths:**

- Flood forecasting is an increasingly relevant problem.
 - The inclusion of dynamic gate and pump inputs is novel to the best of my knowledge.

**Weaknesses:**

- The metrics, terminology and units are non-standard for the hydrology community.
   E.g. the "water level" is normally referred to as "streamflow",
   performance of streamflow prediciton is typically reported in terms of Nash-Sutcliffe Efficiency (NSE)
   with additional metrics for peak and low flow performance (see e.g. Kratzert et al., 2019),
   and precipitation ("rainfall" in this manuscript) is provided in mm / day (instead of inches) (see e.g. Addor et al., 2017).
   This renders this work practically incomparable with most existing work in the field of hydrology.
 - The performance of baselines is reported without error bars, making this work practically irreproducible.
   Most deep learning methods involve sources of randomness and therefore it is necessary to aggregate results from multiple re-runs.
 - The hydrology community has converged to using LSTMs for streamflow prediction (e.g., Kratzert et al., 2018; Nearing et al., 2024; Tursun et al., 2024).
   Therefore, it seems a bit odd that they do not seem to perform well on this particular benchmark.
   This suggests that the baselines might not have been properly configured or tuned.
   Furthermore, there are no details on how the hyper-parameters were obtained (not even in the appendix).
 - The context of two days of data proposed in this manuscript is not well motivated.
   Especially, considering that the goal is to make predictions up to seven days in the future.
   Typical streamflow prediction tasks rely on one year of data (e.g. Nearing et al., 2024).
   Having one year of data should make it significantly easier to predict seasonal events.
 - It is not entirely clear what the key novelty of the proposed benchmark is supposed to be.
   There are plenty of public datasets that enable streamflow prediction on a global scale
   as well as efforts to aggregate the information that is provided in these different datasets into one single dataset (Kratzert et al., 2023).
   These also provide additional features, mostly through meteorological forcing data,
   and addition of the dynamic gate and pump inputs appears to be too specific for the South Florida region to enable benefits elsewhere in the world.
 - The usefulness of this particular dataset seems to be limited.
   First of all, the gauge network of South Florida is likely way more dense than in most regions in the world (see references in Nearing et al., 2024).
   Therefore, it seems unlikely that improvements on this benchmark would translate to other regions around the world.
   Secondly, it can be beneficial to include more geographical diversity in hydrological datasets (Kratzert et al., 2024).
   Events that are rare in one catchment could be quite common in another catchment somewhere else.
   Therefore, I would suspect flood prediction to benefit from a more diverse dataset.
 - There has been a recent surge in foundation models for time-series forecasting (e.g. Auer et al., 2025; Ansari et al., 2025)
   A modern hydrology benchmark should provide baseline performance of at least one of these models.
 - It is unclear why the data needs to be explicilty split in eight parts.
   Normally, it should be sufficient to sample random sub-sequences from a single time-series.
   Especially given that sub-sequences have a fixed length, I can not think of a reason why it would be necessary or useful to artificially split the data like that.

### Additional References
 - Ansari et al. (2025). [Chronos-2: From Univariate to Universal Forecasting](https://arxiv.org/abs/2510.15821). arXiv preprint arXiv:2510.15821.
 - Auer et al. (2025). [TiRex: Zero-Shot Forecasting Across Long and Short Horizons with Enhanced In-Context Learning](https://arxiv.org/abs/2505.23719). arXiv preprint arXiv:2505.23719.
 - Kratzert et al. (2019). [Towards learning universal, regional, and local hydrological behaviors via machine learning applied to large-sample datasets](https://hess.copernicus.org/articles/23/5089/2019/hess-23-5089-2019.html). Hydrology and Earth System Sciences, 23(12), 5089-5110.
 - Kratzert et al. (2018). [Rainfall-runoff modelling using long short-term memory (LSTM) networks](https://hess.copernicus.org/articles/22/6005/2018/). Hydrology and Earth System Sciences, 22(11), 6005-6022.
 - Kratzert et al. (2023). [Caravan-A global community dataset for large-sample hydrology](https://www.nature.com/articles/s41597-023-01975-w). Scientific Data, 10(1), 61.
 - Kratzert et al. (2024). [HESS Opinions: Never train a Long Short-Term Memory (LSTM) network on a single basin](https://hess.copernicus.org/articles/28/4187/2024/hess-28-4187-2024.html). Hydrology and Earth System Sciences, 28(17), 4187-4201.
 - Nearing et al. (2024). [Global prediction of extreme floods in ungauged watersheds](https://www.nature.com/articles/s41586-024-07145-1). Nature, 627(8004), 559-563.
 - Tursun et al. (2024). [Streamflow prediction in human-regulated catchments using multiscale deep learning modeling with anthropogenic similarities](https://agupubs.onlinelibrary.wiley.com/doi/full/10.1029/2023WR036853). Water Resources Research, 60(9), e2023WR036853.

**Questions:**

1. Is there any motivation for using such a non-standard terminology, metrics and units?
 2. How do the baseline results compare to established results in the hydrology literature?
 3. What is the average performance (with confidence intervals) of the baseline models?
 4. How can it be that the best baseline model is an MLP when all modern results rely on LSTMs?
 5. Why are only two days of data used for prediction, instead of one year?
 6. What does the proposed dataset bring compared to the Caravan dataset (Kratzert et al., 2023)?
 7. To what extent could existing flood warning systems be improved with ML when there are so many available gauges?
 8. How many regios exist with such a dense gauge network?
 9. How do modern foundation models for time-series perform on this benchmark?
 10. Why is the data split in eight distinct parts?
 11. How is the data split in parts? What criteria were used to decide where to split?

---

### Official Review · Reviewer_vjiA · 2025-10-31

**Soundness:** 3
**Presentation:** 3
**Contribution:** 2
**Rating:** 4
**Confidence:** 4

**Summary:**

This paper introduces SF^2Bench, which is a benchmark for compound (rainfall, sea, ground water, etc.) flood forecasting. The dataset is built from data from South Florida and is composed of time series from 2,452 monitoring stations in the period from 1985 to 2024. Compared with other datasets available, SF^2Bench focuses on a particular region and is more comprehensive for compound flooding as it includes human control and groundwater data. The dataset is applied in the comparison of six deep learning models (MLP, CNN, Transformer, RNN, GNN, and LLM) in terms of Mean Squared Error (MSE), Mean Average Error (MAE), and Symmetric Extremal Dependence Index (SEDI). The results show that different models excel at different metrics, with MLP being best in terms of MSE and SEDI and Transformer being best at MAE.

**Strengths:**

- The paper is well-written and easy to follow

- Compound flood forecasting is a relevant problem

- A comprehensive set of models is considered in the experiments

**Weaknesses:**

- It is unclear whether ML conference papers will take advantage of SF^2Bench: The paper does not make a strong case for why the proposed dataset will advance machine learning research. Based on the experiments, it seems that the dataset is yet another time series dataset, and there are plenty of time series benchmarks. None of the methods considered were designed for flood forecasting, and most of the flood forecasting papers cited in the related work are not published in ML conferences or journals.

- The benefits of SF^2Bench for the evaluation of ML models compared to existing datasets are unclear: The paper discusses several existing datasets, especially CAMELS, that do not provide the same information as SF^2Bench, but that itself is not a great reason for a novel dataset. It would be more relevant to emphasize what findings can be supported by SF^2Bench and not by the existing datasets. For instance, is the performance of models on SF^2Bench necessarily different from what is found using other datasets? Would such differences make a difference in real applications?

- The results do not provide any novel insights into how to address the problem: The results presented in the paper do not do a great job at motivating the dataset. The authors describe how the results found are expected based on existing work. It is not clear how the analysis of the spatial and temporal information is particularly novel as well. Ideally, a benchmark paper should bring new insights that might lead to research directions towards novel models for the problem.

**Questions:**

1) How will the proposed dataset contribute to ML research beyond the application domain considered (flooding)?

2) How is the proposed dataset fundamentally different from existing ones in terms of potential research findings it can support?

---

### Meta-Review · Area_Chair_33EE · 2026-01-06

**Summary:**

This paper introduces a new benchmark dataset for compound flood forecasting. However, it is unclear to what extent this dataset will advance machine learning research. Furthermore, the reviewers have identified issues with the dataset itself, including deviations from common practices in hydrology, making it questionable to what extent this benchmark will be able to bridge between the ML and hydrology research communities.

**Reviewer Scores:**

N/A

---

### Decision · Program_Chairs · 2026-01-26

Reject